# SPREADING OUT-OF-DISTRIBUTION DETECTION ON GRAPHS

**Daeho Um**[*]
AI Center, Samsung Electronics
daeho.um@samsung.com

**Jongin Lim**
AI Center, Samsung Electronics
jonny.lim@samsung.com

**Sunoh Kim**
Computer Engineering
Dankook University
suno8386@dankook.ac.kr

**Yuneil Yeo**
Department of Civil and Environmental Engineering
UC Berkeley
yuneily@berkeley.edu

**Yoonho Jung**
Department of Electrical and Computer Engineering
Seoul National University
jungyh19@snu.ac.kr

## ABSTRACT

Node-level out-of-distribution (OOD) detection on graphs has received significant attention from the machine learning community. However, previous approaches are evaluated using unrealistic benchmarks that consider only randomly selected OOD nodes, failing to reflect the interactions among nodes. In this paper, we introduce a new challenging task to model the interactions of OOD nodes in a graph, termed spreading OOD detection, where a newly emerged OOD node spreads its property to neighboring nodes. We curate realistic benchmarks by employing the epidemic spreading models that simulate the spreading of OOD nodes on the graph. We also showcase a "Spreading COVID-19" dataset to demonstrate the applicability of spreading OOD detection in real-world scenarios. Furthermore, to effectively detect spreading OOD samples under the proposed benchmark setup, we present a new approach called energy distribution-based detector (EDBD), which includes a novel energy-aggregation scheme. EDBD is designed to mitigate undesired mixing of OOD scores between in-distribution (ID) and OOD nodes. Our extensive experimental results demonstrate the superiority of our approach over state-of-the-art methods in both spreading OOD detection and conventional node-level OOD detection tasks across seven benchmark datasets. The source code is available at https://github.com/daehoum1/edbd.

## 1 INTRODUCTION

While neural networks have demonstrated surpassing performance in various machine learning tasks (He et al., 2016; Kipf & Welling, 2016), they often yield overconfident outputs for unseen data during training, leading to crucial problems in safety-critical applications. To address this issue, considerable research has been conducted on detecting new samples in out-of-distribution (OOD) (Yang et al., 2021). However, the majority of current OOD approaches assume *i.i.d.* (independently and identically distributed) samples (Hendrycks & Gimpel, 2016; Lee et al., 2018; Liu et al., 2020), and research taking into account inter-dependency among samples has been under-explored.

Recently, several graph-based OOD detection methods (Zhao et al., 2020b; Stadler et al., 2021; Wu et al., 2023; Song & Wang, 2022) have emerged to exploit inter-dependency among samples. These methods leverage the graph structure, where a node represents an individual sample and an edge models the connection between two nodes. Then, they perform node-level OOD detection tasks on the given graph. Despite their potential, existing graph-based OOD methods exhibit a critical limitation. Since there is no graph benchmark dataset designed for node-level OOD detection, these methods are commonly evaluated on synthetic graph datasets where OOD samples are assigned to

---

[*]Corresponding author

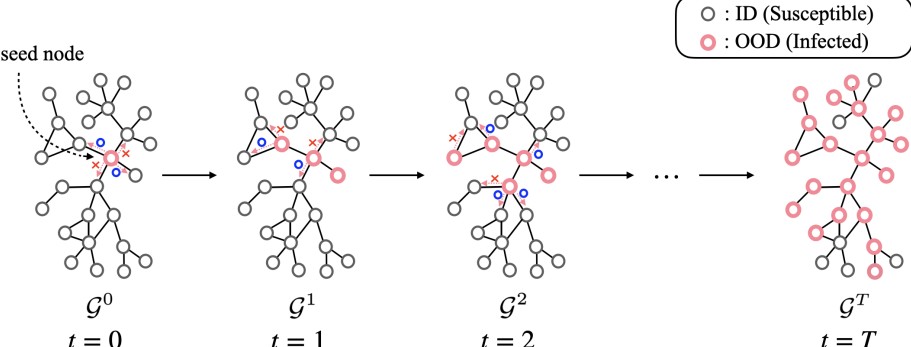

Figure 1: Simulation of the spreading of OOD using an epidemic spreading model. Each infected OOD node attempts to infect a susceptible neighbor with a specific probability. A blue ○ represents a successful infection case along an edge while a red × signifies a failure case. Note that the spread can be initiated from multiple seed nodes in practice. The objective of spreading OOD detection is to discriminate OOD nodes well in every $t \in \{0, \dots, T\}$.

randomly selected nodes.[1] The previous evaluation scenarios, characterized by the random node selection, fail to reflect interactions with the various nodes associated with OOD samples.

In real-world scenarios, newly introduced samples (*i.e.*, OOD samples) typically engage in interactions with other connected nodes, spreading out through edges in a graph. For example, if a person is suddenly infected with a virus, the virus spreads out to the person's neighboring individuals via interaction or contact. Computer viruses (Kephart & White, 1993), brand-new products (Amini & Li, 2011), and contaminants (Mei & Gong, 2018) are other examples of OOD sample spreading. Consequently, for enhanced applicability in real-world scenarios, it is imperative to incorporate a new benchmark setup for graph-based OOD detection that encompasses the spreading of OOD samples.

In this paper, we formulate a new benchmark setup for graph-based OOD methods termed spreading OOD detection. Figure 1 illustrates the scenario of the proposed spreading OOD detection. First, we randomly select nodes to locate OOD samples on a graph with in-distribution (ID) nodes, as in previous graph-based OOD methods (Zhao et al., 2020b; Stadler et al., 2021; Song & Wang, 2022; Wu et al., 2023). These selected nodes serve as sources (*e.g.*, the first infected person) for the spread of OOD. We model the spreading of OOD samples by a graph diffusion process where a new sample in OOD spreads from the source nodes to its neighboring nodes. Specifically, we utilize the epidemic spreading models (Allen, 1994; Peng et al., 2013) that treat a spreading process as a Markov chain. Then, the goal of spreading OOD detection is to discriminate OOD nodes well across all time stamps on the Markov chain, without prior information about the spreading process. The proposed benchmark setup enables the evaluation of graph-based OOD methods on various graph datasets under realistic settings. To show the applicability of spreading OOD detection in real-world situations, we further present Spreading COVID-19, a new benchmark dataset simulating the spread of COVID-19.

Furthermore, we propose Energy Distribution-Based Detector (EDBD), a novel graph-based OOD method that utilizes an energy (Liu et al., 2020) as an OOD score. Unlike the previous graph-based OOD method (Wu et al., 2023) that updates initial energies via simple neighborhood aggregation solely based on the graph structure, we allow energies to directly control their aggregation process. Specifically, we exploit the initial energies as a temporary OOD indicator, and regulate aggregation on each node to alleviate the mixing of energies between OOD nodes and ID nodes. This deliberate aggregation of EDBD enhances accurate OOD discrimination, ensuring that the influence of OOD nodes on neighboring ID nodes is appropriately tempered. Through extensive experiments, we demonstrate the effectiveness of EDBD not only in spreading OOD detection but also the existing OOD detection settings.

Our key contributions are summarized as follows:

- We formulate spreading OOD detection, a new benchmark setup for node-level OOD detection that incorporates interactions among OOD samples. We further establish a new

---

[1]Nodes belonging to randomly chosen classes (Zhao et al., 2020b; Stadler et al., 2021; Wu et al., 2023; Song & Wang, 2022) or randomly chosen nodes (Stadler et al., 2021) are designated as OOD.

dataset called Spreading COVID-19 to benchmark the spreading OOD detection scenario in real-world data. We believe that our new benchmark setup and dataset offer a robust basis for comparing node-level OOD detection methods, considering their practical applicability to real-world scenarios.

- We propose EDBD, a novel node-level OOD detector, which employs an attentive energy-aggregation scheme that prevents the mixing of energies between ID nodes and OOD nodes. Our EDBD framework demonstrates superior performance over existing state-of-the-art methods in both spreading OOD detection and conventional OOD detection tasks on graphs.

## 2 RELATED WORK

**OOD Detection.** OOD detection and OOD generalization are two major problems related to OOD. This paper focuses on OOD detection (Yang et al., 2021; Yoon et al., 2024; Um et al., 2024), where the goal is to identify unfamiliar samples not drawn from in-distribution (ID), which is the distribution of training data. A popular line of OOD detection research designs scoring functions, including maximum softmax probability (Hendrycks & Gimpel, 2016), ODIN score (Liang et al., 2018), Mahalanobis distance-based score (Lee et al., 2018), and energy-based score (Liu et al., 2020). Besides, various approaches, such as outlier exposure introducing auxiliary OOD dataset (Hendrycks et al., 2019; Liu et al., 2020), have been proposed to tackle OOD detection. Meanwhile, OOD generalization (Shen et al., 2021) aims to develop a machine learning model to perform well on OOD data. However, many OOD detection studies assume that inputs are *i.i.d.*, thus predictions are based on a single input.

**OOD Detection on Graphs.** In contrast, graph-structured data contain dependencies between nodes, which are represented through the graph structure. Therefore, several methods have been developed to consider the graph structure in detecting OOD graphs (Li et al., 2022; Liu et al., 2023). However, while these methods target graph-level OOD detection tasks, we focus on node-level OOD detection tasks. To address node-level OOD detection tasks on graphs, various techniques have been developed, such as Graph-based Kernel Dirichlet distribution Estimation (GKDE) (Zhao et al., 2020b), Graph Posterior Network (GPN) (Stadler et al., 2021), and OOD Graph Attention Network (OODGAT) (Song & Wang, 2022). Recently, GNNSAFE (Wu et al., 2023) shows its effectiveness in node-level OOD detection, which produces a final OOD score for a node by aggregating the initial OOD scores of neighboring nodes. However, in GNNSAFE, a node aggregates OOD scores from neighboring nodes, assigning equal importance to each, regardless of the OOD score distribution.

**Machine Learning for COVID-19.** The COVID-19 pandemic has driven the machine learning community to develop solutions and proactively prepare for future epidemics. In response, many datasets related to COVID-19 have been released, including those with medical images, textual data, and speech data (Zhao et al., 2020a; Shuja et al., 2021). Additionally, various studies have focused on modeling the spread of COVID-19 (Panagopoulos et al., 2021; Alguliyev et al., 2021) and applying machine learning techniques for its classification (Barstugan et al., 2020; Song et al., 2023). Nonetheless, our work stands apart due to our novel approach, which does not rely on specific information or features related to COVID-19 during training. This aspect makes our approach adaptable to new epidemics where initial information may be unavailable.

## 3 WHAT IS "SPREADING OOD DETECTION"

In this section, we formulate the problem of spreading OOD detection and describe the procedure of the spreading OOD benchmark setup in detail.

### 3.1 NOTATION

Let $\mathcal{G} = (\mathcal{V}, \mathcal{E}, \mathbf{X})$ be a graph where $\mathcal{V} = \{v_1, \ldots, v_N\}$ represents a set of $N$ nodes, $\mathcal{E}$ represents a set of edges, and $\mathbf{X} \in \mathbb{R}^{N \times F}$ represents a feature matrix of $F$-dimensional node features. $\mathbf{A} \in \{0, 1\}^{N \times N}$ denotes an adjacency matrix, where $\mathbf{A}_{i,j} = 0$ only if $v_i$ and $v_j$ are not connected to each other. $\mathbf{x}_i$ denotes the node feature of $v_i$, *i.e.*, $i$-th row vector of $\mathbf{X}$. $\mathcal{N}(v_i)$ denotes the set of neighbors of $v_i$.

### 3.2 PROBLEM STATEMENT

Figure 4 illustrates an overview of the proposed spreading OOD detection. Formally, let $\mathcal{D}_{in}$ and $\mathcal{D}_{out}$ represent in-distribution (ID) and out-of-distribution (OOD) data, respectively. For $\mathcal{D}_{in}$, each sample in $\mathcal{D}_{in}$ is associated with a feature vector $\mathbf{x}_{in} \in \mathbb{R}^F$ and a class label $y_{in} \in \{1, \ldots, C\}$.

For $\mathcal{D}_{out}$, is associated only with a feature vector $\mathbf{x}_{out} \in \mathbb{R}^F$, and $\mathbf{x}_{out}$ and $\mathbf{x}_{in}$ are sampled from distinct distributions. Following the conventional OOD benchmarks (Hendrycks & Gimpel, 2016), we split $\mathcal{D}_{in}$ into $\mathcal{D}_{in}^{train}$ and $\mathcal{D}_{in}^{test}$, and only $\mathcal{D}_{in}^{train}$ is used for training. During training, we train a neural classifier with features and labels in $\mathcal{D}_{in}^{train}$, without a graph structure, to account for real-world spreading scenarios where classification often occurs at the sample-level (Perdisci et al., 2008; Kannan et al., 2011; Ward et al., 2020).

For evaluation, we assume a graph $\mathcal{G} = (\mathcal{V}, \mathcal{E}, \mathbf{X})$ is given, and formulate a graph-based OOD problem. Specifically, we define $\mathcal{G}$ to be a graph consisting of $N$ ID samples in $\mathcal{D}_{in}^{test}$ as nodes of the graph. The graph structure of $\mathcal{G}$ represents inter-dependency among the nodes such as social relations among people. At $t = 0$, we randomly select nodes in the given graph $\mathcal{G}$ and set them as OOD nodes. These selected nodes serve as sources (*e.g.*, the first infected person) for the spread of OOD. After the initialization, each ID node is probabilistically infected by neighboring OOD nodes for each time stamp $t \geq 1$, spreading OOD samples through the graph as $t$ increases.[2] When an ID node becomes an OOD node, its node feature is replaced by that of an OOD sample randomly chosen in $\mathcal{D}_{out}$. Concretely, we denote the resulting graph at each time stamp $t \in \{0, 1, \ldots, T\}$ as $\mathcal{G}^t = (\mathcal{V}, \mathcal{E}, \mathbf{X}^t)$ where $T$ is the end point of the spread and $\mathbf{X}^t$ denotes a feature matrix containing OOD features for infected nodes at $t$.

The goal of spreading OOD detection is to discriminate OOD samples (infected nodes) from ID samples (uninfected nodes) when $\mathcal{G}^t$ is given. Note that during evaluation, only $\mathcal{G}^t$ is provided without any additional information such as the position of the seed node or the current time stamp $t$. Thus, spreading OOD detection requires an OOD detector that performs well at any time stamp $t$. This benchmark setup does not necessarily require a dedicated dataset; it can be easily implemented using existing graph datasets (see Appendix B.2.1). Spreading OOD detection provides a realistic setup for node-level OOD detection, which lack a standard evaluation setup except for random class split.

### 3.3 OOD Spreading Scheme

Let $\mathbf{M} \in \{0, 1\}^{N \times (T+1)}$ be a binary mask representing which nodes are OOD at $t$ where $\mathbf{M}_{i,t} = 1$ represents that $v_i$ is OOD at $t$. Note that some nodes are randomly designated as OOD nodes at $t = 0$, and then the OOD samples spread along edges as $t$ increases. To model the spread of OOD samples, we employ Susceptible-Infected (SI) and Susceptible-Infected-Susceptible (SIS) models (Allen, 1994; Peng et al., 2013), which are widely used in epidemic-spreading studies. In this section, we primarily introduce the procedure using SIS model, since SI model can be considered as a special case of SIS model.

SIS model can be formulated as a Markov process. Specifically, at each time stamp $t$, every node in $\mathcal{G}^t$ can be in one of two states: susceptible or infected. During each time interval between consecutive time stamps, an infected node attempts to infect its susceptible neighbor with a probability of $\gamma$. Simultaneously, an infected node may be cured with a probability of $\delta$. The probability that $v_i$ will not be infected by its neighbors during the interval $[t-1, t]$ is denoted as $\zeta_{i,t}$. $v_i$ will remain uninfected at $t$ if all of $v_i$'s neighbors were either uninfected at $t-1$ or, if they were infected, they failed to infect $v_i$ during $[t-1, t]$. Thus, considering multiple neighbors of $v_i$ simultaneously, $\zeta_{i,t}$ can be expressed as follows:

$$\begin{aligned}
\zeta_{i,t} &= \prod_{j \in \mathcal{N}(i)} (\mathbf{M}_{j,t-1}(1 - \gamma) + (1 - \mathbf{M}_{j,t-1})) \\
&= \prod_{j \in \mathcal{N}(i)} (1 - \gamma \mathbf{M}_{j,t-1}).
\end{aligned} \tag{1}$$

$v_i$ is uninfected at $t$ if it was uninfected at $t-1$ and did not receive infections from its neighbors at $t$, or was infected at $t-1$ but was cured at $t$. That is, the probability that $v_i$ is uninfected can be expressed by $(1 - \mathbf{M}_{i,t-1})\zeta_{i,t} + \delta \mathbf{M}_{i,t-1}\zeta_{i,t}$. Hence, we determine the OOD configuration at $t$ depending on that at $t-1$ as follows:

$$\mathbf{M}_{i,t} \sim \text{Ber}(1 - ((1 - \mathbf{M}_{i,t-1})\zeta_{i,t} + \delta \mathbf{M}_{i,t-1}\zeta_{i,t})), \tag{2}$$

for $i \in \{1, \ldots, N\}$. When a node is infected, the original features of the node is replaced with features sampled from $\mathcal{D}_{out}$. As $t$ increases by 1, we iteratively spread OOD using Equation (2). At each time stamp $t$, as features change by infection, $\mathcal{G}^t = (\mathcal{V}, \mathcal{E}, \mathbf{X}^t)$ is generated.

---

[2]Details of the spreading scheme (*i.e.* how to infect a node?) is described in Sec. 3.3.

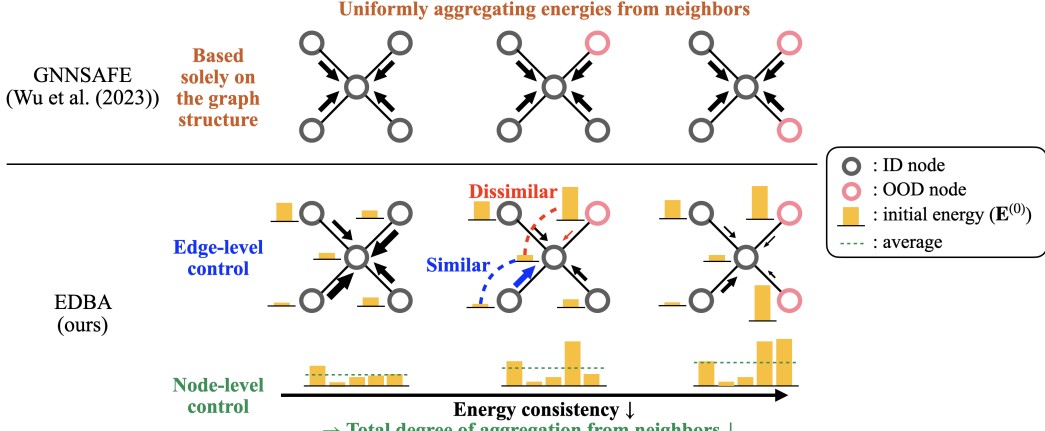

Figure 2: The concept of energy distribution-based aggregation (EDBA) that prevents undesired energy mixing between ID and OOD nodes. EDBA allows the energies themselves to control the entire aggregation process. As shown in the three graphs below, high energy similarity between two ID nodes (or OOD nodes) strengthens energy propagation at edge level, whereas low energy similarity between ID and OOD nodes weakens energy propagation. The energy histograms at the bottom show energy variance among nodes, representing the energy distribution consistency among nodes. Moving from left to right, the histogram exhibits larger variance and subsequently lower consistency, resulting in a decrease in the total degree of aggregation from neighboring nodes.

SI models are SIS models with $\delta = 0$, assuming that once a node is infected, it cannot be cured. Thus, the spread in an SI model is also simulated through Eq 2 by substituting zero for $\delta$. Figure 1 illustrates the simulation process using an SI model.

## 3.4 SPREADING COVID-19

We create Spreading COVID-19 dataset to demonstrate the applicability of spreading OOD detection in real-world problems. Spreading COVID-19 simulates the propagation of COVID-19 on a human network. This dataset enables the detection of individuals infected with COVID-19 by identifying OOD nodes on the graph. In this dataset, ID data consists of four classes including normal (no illness), allergies, cold, and flu. An OOD class is COVID-19, which is a disease that has suddenly emerged and significantly impacted the world. For each class, we generate samples with 23-dimensional features, created based on information about symptoms of respiratory illnesses (AAFA; Clinic), such as chest tightness and rapid breathing. We utilize a graph structure from the LastFM Asia graph (Rozemberczki & Sarkar, 2020). We use SI and SIS models to simulate the spread of COVID-19 and create episodes for both models, where an episode represents $\{\mathcal{G}^t\}_{t \in [0,T]}$. Figure 4 in Appendix A illustrates the process of spreading OOD detection on a mini version of the Spreading COVID-19 dataset under the single-seed setting. Further details on the Spreading COVID-19 dataset, including justification regarding dataset construction, are included in Appendix A, and we provide the entire dataset in the supplementary material.

## 4 PROPOSED METHOD

In this section, we describe our method called Energy Distribution-Based Detector (EDBD) for spreading OOD detection. We first briefly overview the proposed EDBD in Sec 4.1. We then present an aggregation scheme of EDBD and two main components of the aggregation scheme in Sec. 4.2, Sec. 4.3, and Sec. 4.4, respectively.

### 4.1 OVERVIEW

The key challenge in our method is how to design an energy-aggregation scheme to enhance the discriminative ability between ID nodes and OOD nodes. We utilize an energy (LeCun et al., 2006) as an OOD score and aim to refine energies through a novel aggregation scheme. To obtain these energies, we first train a neural classifier on $\mathcal{D}_{in}^{train}$ with cross-entropy loss. After training the neural classifier, we define an energy function of the neural classifier $f(\mathbf{x}) : \mathbb{R}^F \to \mathbb{R}^C$, which maps an $F$-dimensional input vector $\mathbf{x}$ to $C$-dimensional logits. The energy function $E(f(\mathbf{x})) : \mathbb{R}^C \to \mathbb{R}$ (LeCun

et al., 2006) is defined by

$$E(f(\mathbf{x})) = -T \cdot \log \sum_{c=1}^{C} e^{f_c(\mathbf{x})/T}, \tag{3}$$

where a hyperparameter $T$ is set to 1 in most cases and $f_y(\mathbf{x})$ denotes the $y$-th index of $f(\mathbf{x})$, *i.e.*, the logit corresponding to the $y$-th class. The energy $E(f(\mathbf{x}))$ is shown to be an efficient indicator for OOD detection (Liu et al., 2020), where a higher $E(f(\mathbf{x}))$ implies that $\mathbf{x}$ is more likely to be OOD. Hence, we identify a node as an OOD node when its energy is high. Utilizing the graph structure, neighborhood aggregation can refine the initial energies to be more helpful in OOD detection (Wu et al., 2023). Here, we aim to design an aggregation scheme to enhance the discriminative ability of energies between ID and OOD nodes.

Unlike the previous method (Wu et al., 2023) that uniformly aggregates energies of neighboring nodes, our EDBD deliberately aggregates energies based on the energy distribution of neighboring nodes (see Figure 2), which avoids the undesired mixing of energies between ID and OOD nodes. In EDBA, the initial energies serve not only as targets for aggregation but also as controllers of the aggregation process (see Sec. 4.2). Specifically, EDBA consists of edge-level and node-level controllers. For the edge-level controller, EDBA employs the energy similarity matrix as the transition matrix used for aggregation (see Sec. 4.3). Namely, EDBA weakens the energy propagated from a node with high energy (potentially an OOD node) to a connected node with low energy (potentially an ID node) during aggregation, and vice versa. For the node-level controller, EDBA employs the energy consistency matrix as the degree of total aggregation from neighbors (see Sec. 4.4). When energies are inconsistent across the neighboring nodes, ID and OOD nodes may coexist among them. In such cases, aggregating energy from neighboring nodes may lead to undesirable outcomes. Hence, EDBA adjusts the degree of aggregation based on the energy consistency matrix.

## 4.2 ENERGY DISTRIBUTION-BASED AGGREGATION

EDBA controls energy aggregation at both the edge-level and node-level to mitigate the mixing of energies between ID nodes and OOD nodes. When $\mathcal{G}^t = (\mathcal{V}, \mathcal{E}, \mathbf{X}^t)$ is provided in testing phase, EDBD obtains the initial energies $\mathbf{E}^{(0)} = [\mathbf{E}_1^{(0)}, \ldots, \mathbf{E}_N^{(0)}]^\top$ by using Equation (3), where $\mathbf{E}_i^{(0)} = E(f(\mathbf{x}_i^t))$. Then, EDBA refines $\mathbf{E}^{(0)}$ through $K$-step aggregation. EDBA can be formulated by a convex combination of $\mathbf{E}^{(k-1)}$ and transitioned $\mathbf{E}^{(k-1)}$ (*i.e.*, $\mathbf{SE}^{(k-1)}$) as

$$\mathbf{E}^{(k)} = (\mathbf{I} - \alpha\mathbf{C})\mathbf{E}^{(k-1)} + \alpha\mathbf{CSE}^{(k-1)}, \tag{4}$$

where $\mathbf{S} \in \mathbb{R}^{N \times N}$ and $\mathbf{C} \in \mathbb{R}^{N \times N}$ denote energy similarity matrix and energy consistency matrix, respectively, while $\alpha$ is a hyperparameter between 0 and 1, and $\mathbf{I} \in \mathbb{R}^{N \times N}$ is an identity matrix. After completing $K$ aggregation steps with EDBA, the final energies $\mathbf{E}^{(K)}$ are obtained, which are used to detect OOD samples.

During aggregation, while $\mathbf{S}$ determines the degree of message passing that occurs on each edge (edge level), $\mathbf{C}$ determines the degree of reflecting transitioned energies from neighboring nodes for each node (node level). These $\mathbf{S}$ and $\mathbf{C}$ are constructed by utilizing $\mathbf{E}^{(0)}$.

## 4.3 ENERGY SIMILARITY MATRIX

In almost all cases, energy-based OOD detectors do not show perfect performance on test data. This implies that high energies are not guaranteed for OOD samples. To refine imperfect energies, GNNSAFE (Wu et al., 2023) demonstrates that aggregating energies from neighbors can be effective for OOD detection on graph-structured data. However, simple aggregation of energies from neighbors may impede OOD detection as an OOD node may have ID nodes in its neighbors, and vice versa. For example, from the central node's point of view in Figure 3(a), two nodes are ID among the three neighbors while the central node is OOD. Hence, aggregating energies from neighboring nodes with equal importance might lower the central node's energy, which can hamper its correct classification as OOD.

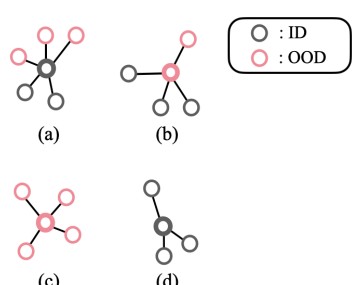

Figure 3: Examples of a node and its neighbors on $\mathcal{G}_t$.

Inspired by graph neural networks that use the attention mechanism (Veličković et al., 2018; Lim et al., 2021), we propose energy similarity-based aggregation for edge-level control of aggregation. Energy aggregation at the $i$-th node $v_i$ for updating its energy can be seen as a weighted sum of the $i$-th node's energy and the energies of its neighbors. In our energy similarity-based aggregation, the weight of a neighbor's energy is adjusted based on its similarity to the $v_i$'s energy. Energy similarity-based aggregation assigns the larger weight of a neighbor's energy as the energies between the neighbor and $v_i$ are similar. This ensures a more accurate energy assessment for OOD detection.

Formally, we construct a weighted adjacency matrix $\overline{\mathbf{S}} \in \mathbb{R}^{N \times N}$ as follows:

$$\overline{\mathbf{S}}_{i,j} = \begin{cases} \text{sim}(\mathbf{E}_i^{(0)}, \mathbf{E}_j^{(0)}) & \text{if } \mathbf{A}_{i,j} = 1 \\ 0 & \text{if } \mathbf{A}_{i,j} = 0, \end{cases} \tag{5}$$

where $\mathbf{A}$ is the adjacency matrix of $\mathcal{G}^t$ and $\text{sim}(\mathbf{E}_i^{(0)}, \mathbf{E}_j^{(0)})$ represents the similarity between $\mathbf{E}_i^{(0)}$ and $\mathbf{E}_j^{(0)}$. We define $\text{sim}(\mathbf{E}_i^{(0)}, \mathbf{E}_j^{(0)})$ as $(\epsilon \cdot (\max(\mathbf{E}^{(0)}) - \min(\mathbf{E}^{(0)})) + (1 - \epsilon) \cdot |\mathbf{E}_i^{(0)} - \mathbf{E}_j^{(0)}|)^{-1}$, where $\epsilon \in (0, 1)$ is a hyperparameter. We then normalize $\overline{\mathbf{S}}$ to $\mathbf{S} = \mathbf{D}^{-1}\overline{\mathbf{S}}$ where $\mathbf{D}$ is a diagonal matrix with diagonal entries $\mathbf{D}_{ii} = \sum_j \overline{\mathbf{S}}_{i,j}$. This row-stochastic $\mathbf{S}$ effectively aggregates neighboring energies based on the energy similarity between two connected nodes.

## 4.4 ENERGY CONSISTENCY MATRIX

Since OOD spreads by infecting nodes that are directly connected to them along edges, OOD nodes cluster together. Consequently, the remaining nodes, which are ID nodes, also form clusters. In the case where nodes are located within their cluster like Figure 3(c) and 3(d), energy aggregation can help refining each central node's energy to be closer to its own type since it is connected only with nodes of the same type (OOD/ID). In contrast, Figure 3(a) and 3(b) illustrate nodes located at the boundary between an ID cluster and an OOD cluster. In these cases, aggregating the energy of neighbors may impede accurate OOD detection. Hence, energy consistency-based aggregation is designed to vary the total amount of energies from neighbors for each node differently, depending on the node's location in relation to the clusters.

To control aggregation at node-level based on the location of a node, we analyze the energy distribution of the node and its neighboring nodes. If a node is located within a cluster, the energy variance among these nodes will be low (*i.e.*, the energies are consistent), as all the nodes belong to the same type (either OOD or ID). Conversely, for a node at the boundary, this variance will be high (*i.e.*, the energies are inconsistent). Thus, for each node, we construct a set denoted by $\mathcal{S}$ as follows,

$$\mathcal{S}(v_i) = \{\mathbf{E}_i^{(0)}\} \cup \{\mathbf{E}_j^{(0)}\}_{j \in \mathcal{N}(v_i)} \tag{6}$$

for $i \in \{1, \ldots, N\}$. We then calculate the standard deviation of the energies within $\mathcal{S}(v_i)$ for all $i \in \{1, \ldots, N\}$. Let $\sigma_i$ denote the standard deviation for the energies in $\mathcal{S}(v_i)$. Using $\{\sigma_i\}_{i=1,\ldots,N}$, we construct a normalized diagonal matrix $\mathbf{\Sigma}$, where diagonal entries $\mathbf{\Sigma}_{i,i} = \sigma_i/\max(\{\sigma_j\}_{j \in \{1,\ldots,N\}})$ $(0 \leq \mathbf{\Sigma}_{ii} \leq 1)$. Finally, we define $\mathbf{C} \in \mathbb{R}^{N \times N}$ by

$$\mathbf{C} = \mathbf{I} - \beta\mathbf{\Sigma}, \tag{7}$$

where $0 < \beta \leq 1$ is a hyperparamter that controls the influence of $\mathbf{\Sigma}_{ii}$.

We recall Equation (4) of EDBA expressed by $\mathbf{E}^{(k)} = (\mathbf{I} - \alpha\mathbf{C})\mathbf{E}^{(k-1)} + \alpha\mathbf{C}\mathbf{S}\mathbf{E}^{(k-1)}$. If $v_i$ has inconsistent energy distribution in its surroundings (*i.e.*, high $\mathbf{\Sigma}_{ii} \approx 1$), $\mathbf{C}_{ii}$ approaches zero as $\beta$ increases. This implies the $i$-th element of $\alpha\mathbf{C}\mathbf{S}\mathbf{E}^{(k-1)}$ becomes almost zero, that is, the transitioned energy of $v_i$ becomes almost zero during aggregation. Then, $\mathbf{E}_i^{(k)} \approx \mathbf{E}_i^{(k-1)}$, that is, $v_i$ preserves its energy value. In summary, if $v_i$ is located at the boundary, $\Sigma_{ii}$ tends to have a high value. Thus, by controlling $\beta$, we can make $v_i$ preserve its energy without energy mixing from its neighboring nodes.

## 5 EXPERIMENTS

The term *label leave-out* refers to the conventional node-level OOD detection task on graphs, which assumes nodes belonging to a subset of classes as ID and leaves out the other nodes for OOD. We conduct extensive experiments on 1) spreading OOD detection and 2) label leave-out.

Table 1: Performance on label leave-out in benchmark datasets. The average FPR95 (%), average AUROC (%), and average AUPR (%) across 10 independent runs are reported with standard deviation errors.

| Dataset | CORA | | | AMAZON-PHOTO | | |
|---|---|---|---|---|---|---|
| Method | FPR95($\downarrow$) | AUROC($\uparrow$) | AUPR($\uparrow$) | FPR95($\downarrow$) | AUROC($\uparrow$) | AUPR($\uparrow$) |
| MSP | $40.37 \pm 2.19$ | $91.13 \pm 0.22$ | $78.16 \pm 0.19$ | $28.87 \pm 1.65$ | $94.41 \pm 0.61$ | $92.44 \pm 0.75$ |
| ODIN | $100.00 \pm 0.00$ | $49.05 \pm 0.57$ | $24.18 \pm 0.08$ | $92.72 \pm 8.43$ | $63.30 \pm 7.61$ | $51.72 \pm 7.43$ |
| Mahalanobis | $86.11 \pm 6.19$ | $66.93 \pm 1.95$ | $40.56 \pm 3.77$ | $56.11 \pm 16.06$ | $82.51 \pm 4.83$ | $75.73 \pm 7.09$ |
| Energy | $38.36 \pm 3.46$ | $91.46 \pm 0.33$ | $78.10 \pm 0.29$ | $30.49 \pm 3.93$ | $93.96 \pm 0.68$ | $91.73 \pm 0.75$ |
| GKDE | $60.88 \pm 2.25$ | $87.15 \pm 0.60$ | $72.12 \pm 1.10$ | $91.60 \pm 8.81$ | $60.00 \pm 11.43$ | $56.61 \pm 12.77$ |
| GPN | $44.04 \pm 5.85$ | $87.48 \pm 6.38$ | $81.21 \pm 7.40$ | $35.54 \pm 11.48$ | $91.48 \pm 2.71$ | $88.04 \pm 3.41$ |
| OODGAT | $85.21 \pm 1.66$ | $64.81 \pm 0.87$ | $62.65 \pm 1.01$ | $13.33 \pm 0.46$ | $97.27 \pm 0.33$ | $95.01 \pm 0.60$ |
| GNNSAFE | $31.31 \pm 1.11$ | $92.84 \pm 0.38$ | $82.22 \pm 0.40$ | $6.57 \pm 0.38$ | $97.36 \pm 0.04$ | $97.13 \pm 0.10$ |
| EDBD | $\mathbf{30.48 \pm 1.11}$ | $\mathbf{92.95 \pm 0.38}$ | $\mathbf{82.31 \pm 0.38}$ | $\mathbf{5.82 \pm 0.66}$ | $\mathbf{97.48 \pm 0.07}$ | $\mathbf{97.60 \pm 0.06}$ |
| Dataset | AMAZON-COMPUTERS | | | COAUTHOR-CS | | |
| Method | FPR95($\downarrow$) | AUROC($\uparrow$) | AUPR($\uparrow$) | FPR95($\downarrow$) | AUROC($\uparrow$) | AUPR($\uparrow$) |
| MSP | $70.77 \pm 3.54$ | $76.81 \pm 2.31$ | $71.01 \pm 2.23$ | $29.07 \pm 3.53$ | $94.15 \pm 0.73$ | $97.73 \pm 0.28$ |
| ODIN | $98.72 \pm 2.56$ | $53.36 \pm 1.91$ | $45.93 \pm 2.83$ | $100.00 \pm 0.00$ | $52.35 \pm 4.36$ | $75.26 \pm 1.96$ |
| Mahalanobis | $71.09 \pm 1.90$ | $73.14 \pm 1.27$ | $62.63 \pm 2.43$ | $64.40 \pm 14.83$ | $81.73 \pm 3.53$ | $84.30 \pm 15.80$ |
| Energy | $58.40 \pm 3.41$ | $84.72 \pm 1.50$ | $79.36 \pm 1.66$ | $18.60 \pm 3.87$ | $95.98 \pm 0.71$ | $98.39 \pm 0.29$ |
| GKDE | $90.64 \pm 7.22$ | $58.59 \pm 10.46$ | $49.23 \pm 8.10$ | $59.70 \pm 7.83$ | $88.02 \pm 1.77$ | $95.50 \pm 0.65$ |
| GPN | $80.55 \pm 16.98$ | $74.08 \pm 15.09$ | $69.27 \pm 17.30$ | $26.68 \pm 11.56$ | $93.54 \pm 3.35$ | $97.40 \pm 1.47$ |
| OODGAT | $86.16 \pm 7.35$ | $73.55 \pm 5.48$ | $84.17 \pm 2.99$ | $13.16 \pm 1.13$ | $96.83 \pm 0.21$ | $96.58 \pm 0.10$ |
| GNNSAFE | $39.94 \pm 6.84$ | $89.75 \pm 1.79$ | $85.63 \pm 3.36$ | $11.31 \pm 1.69$ | $97.44 \pm 0.35$ | $99.06 \pm 0.13$ |
| EDBD | $\mathbf{35.59 \pm 6.94}$ | $\mathbf{92.45 \pm 0.98}$ | $\mathbf{90.34 \pm 1.45}$ | $\mathbf{10.19 \pm 1.63}$ | $\mathbf{97.68 \pm 0.35}$ | $\mathbf{99.14 \pm 0.13}$ |

## 5.1 EXPERIMENTAL SETUP

**Evaluation Metric.** For label leave-out, we measure three metrics: 1) the false positive rate of OOD samples when the true positive rate of ID samples is at $95\%$ (FPR95); 2) the area under the receiver operating characteristic curve (AUROC); 3) the area under the precision-recall curve (AUPR). In spreading OOD detection, the goal is to perform well under any spreading situations regardless of $t$. Hence we define new metrics calculated per episode. FPR95-T represents the averaged FPR95 across each graph in $\{\mathcal{G}^t\}_{t \in \{1,...,T\}}$, where $T$ is the length of the episode. Similarly, we define AUROC-T and AUPR-T, which are the averaged AUROC and AUPR values, respectively, calculated across each graph at different time stamps within the episode.

**Datasets.** For label leave-out, following the setup in Wu et al. (2023), we perform experiments on four benchmark datasets: Cora (Sen et al., 2008), Amazon-Photo (Shchur et al., 2018), Amazon-Computers (Shchur et al., 2018), and Coauthor-CS (Sinha et al., 2015). For spreading OOD detection, we utilize three benchmark datasets, including Cora, LastFM Asia (Rozemberczki & Sarkar, 2020), and the proposed Spreading COVID-19 dataset. We use SI and SIS models for epidemic spreading simulations. For each (dataset, epidemic model) pair, we create 15 episodes, each of which conclude when over $75\%$ of nodes are infected. We then allocate five episodes for the validation set and ten episodes for the test set. For the benchmark datasets used in spreading OOD detection, we employ a Bernoulli distribution (*i.e.,* $\mathbf{x} \sim \text{Ber}(0.1)$) to generate the features of OOD nodes.

**Baselines.** We compare EDBD with eight state-of-the-art methods, which can be divided into two categories: **(1)** OOD detection methods for *i.i.d.* sampled data and **(2)** OOD detection methods for graph-structured data. Category **(1)** includes MSP (Hendrycks & Gimpel, 2016), ODIN (Liang et al., 2018), Mahalanobis (Lee et al., 2018), and Energy (Liu et al., 2020), while Category **(2)** comprises GKDE (Zhao et al., 2020b), GPN (Stadler et al., 2021), OODGAT (Song & Wang, 2022) and GNNSAFE (Wu et al., 2023). GKDE and OODGAT are evaluated solely on label leave-out since they requires identical graph structures during both the training and testing phases.

**Backbone Encoder.** For fair comparisons, we use the same backbone for all baselines and the proposed method. For spreading OOD detection, we utilize a multi-layer perceptron (MLP) since the connectivity among samples is not provided during training. For label leave-out, where both training and testing are conducted on a given graph, we exploit GCN (Kipf & Welling, 2016) as a backbone. For all experiments, hyperparameters are tuned on validation sets. Further experimental details are provided in Appendix B.

Table 2: Performance on Spreading OOD detection in the Spreading COVID-19 dataset. The average FPR95-T (%), average AUROC-T (%), and average AUPR-T (%) across 10 episodes are reported with standard deviation errors.

### Single-seed setting

| Epidemic model | SI | | | SIS | | |
|---|---|---|---|---|---|---|
| Method | FPR95-T ($\downarrow$) | AUROC-T ($\uparrow$) | AUPR-T ($\uparrow$) | FPR95-T ($\downarrow$) | AUROC-T ($\uparrow$) | AUPR-T ($\uparrow$) |
| MSP | $94.25 \pm 4.56$ | $38.51 \pm 5.41$ | $70.17 \pm 4.55$ | $97.98 \pm 1.38$ | $34.45 \pm 5.87$ | $70.01 \pm 2.73$ |
| ODIN | $93.19 \pm 5.00$ | $61.82 \pm 11.50$ | $78.08 \pm 6.05$ | $89.87 \pm 5.76$ | $65.86 \pm 10.63$ | $80.26 \pm 5.54$ |
| Mahalanobis | $63.64 \pm 44.93$ | $47.29 \pm 42.24$ | $79.46 \pm 15.28$ | $72.71 \pm 36.33$ | $46.61 \pm 34.07$ | $78.95 \pm 12.00$ |
| Energy | $54.82 \pm 17.43$ | $80.46 \pm 8.54$ | $88.00 \pm 5.52$ | $67.94 \pm 18.98$ | $73.94 \pm 14.49$ | $85.51 \pm 5.70$ |
| GPN | $95.29 \pm 2.36$ | $50.35 \pm 13.69$ | $79.55 \pm 65.45$ | $87.07 \pm 14.50$ | $67.20 \pm 12.68$ | $83.03 \pm 4.40$ |
| GNNSAFE | $69.76 \pm 15.23$ | $80.65 \pm 6.07$ | $87.40 \pm 4.12$ | $77.76 \pm 14.64$ | $76.02 \pm 12.65$ | $85.53 \pm 5.30$ |
| EDBD | $\mathbf{54.67 \pm 17.58}$ | $\mathbf{81.60 \pm 8.35}$ | $\mathbf{88.22 \pm 5.59}$ | $\mathbf{67.42 \pm 19.75}$ | $\mathbf{76.22 \pm 14.58}$ | $\mathbf{85.94 \pm 5.94}$ |

### Multi-seed setting

| Epidemic model | SI | | | SIS | | |
|---|---|---|---|---|---|---|
| Method | FPR95-T ($\downarrow$) | AUROC-T ($\uparrow$) | AUPR-T ($\uparrow$) | FPR95-T ($\downarrow$) | AUROC-T ($\uparrow$) | AUPR-T ($\uparrow$) |
| MSP | $96.29 \pm 1.94$ | $36.41 \pm 5.86$ | $58.63 \pm 3.18$ | $96.14 \pm 1.87$ | $37.49 \pm 5.36$ | $55.23 \pm 2.98$ |
| ODIN | $92.61 \pm 3.42$ | $63.15 \pm 12.07$ | $72.47 \pm 6.61$ | $91.93 \pm 4.69$ | $64.59 \pm 10.37$ | $69.05 \pm 6.80$ |
| Mahalanobis | $90.44 \pm 0.53$ | $27.93 \pm 0.67$ | $53.78 \pm 0.95$ | $90.55 \pm 0.50$ | $31.09 \pm 0.63$ | $54.08 \pm 0.62$ |
| Energy | $49.78 \pm 14.73$ | $82.66 \pm 6.79$ | $85.94 \pm 4.92$ | $60.40 \pm 10.33$ | $80.18 \pm 6.24$ | $81.63 \pm 4.84$ |
| GPN | $87.72 \pm 6.17$ | $62.81 \pm 9.26$ | $75.13 \pm 5.38$ | $89.29 \pm 5.18$ | $61.85 \pm 8.69$ | $71.39 \pm 5.29$ |
| GNNSAFE | $63.42 \pm 14.09$ | $82.21 \pm 5.91$ | $\mathbf{86.11 \pm 4.33}$ | $63.32 \pm 1.14$ | $77.80 \pm 5.16$ | $80.40 \pm 3.96$ |
| EDBD | $\mathbf{49.62 \pm 15.22}$ | $\mathbf{83.29 \pm 6.68}$ | $\mathbf{86.11 \pm 5.01}$ | $\mathbf{59.93 \pm 10.47}$ | $\mathbf{80.68 \pm 6.16}$ | $\mathbf{81.70 \pm 4.92}$ |

Table 3: Performance on Spreading OOD detection in benchmark datasets. The average FPR95-T (%), average AUROC-T (%), and average AUPR-T (%) across 10 episodes are reported with standard deviation errors.

| Epidemic model | Dataset | CORA | | | LASTFM ASIA | | |
|---|---|---|---|---|---|---|---|
| | Method | FPR95-T ($\downarrow$) | AUROC-T ($\uparrow$) | AUPR-T ($\uparrow$) | FPR95-T ($\downarrow$) | AUROC-T ($\uparrow$) | AUPR-T ($\uparrow$) |
| SI | MSP | $24.80 \pm 16.38$ | $96.04 \pm 2.26$ | $97.73 \pm 1.13$ | $52.98 \pm 0.41$ | $93.71 \pm 1.80$ | $97.96 \pm 0.44$ |
| | ODIN | $100.00 \pm 0.00$ | $45.65 \pm 10.85$ | $74.03 \pm 4.70$ | $100.00 \pm 0.00$ | $17.35 \pm 6.96$ | $68.61 \pm 4.08$ |
| | Mahalanobis | $65.32 \pm 44.90$ | $65.50 \pm 32.93$ | $88.21 \pm 11.85$ | $59.29 \pm 48.43$ | $87.34 \pm 15.97$ | $96.66 \pm 3.92$ |
| | Energy | $22.79 \pm 18.21$ | $96.36 \pm 2.12$ | $98.03 \pm 1.07$ | $54.44 \pm 16.82$ | $93.56 \pm 1.71$ | $97.86 \pm 0.45$ |
| | GPN | $100.00 \pm 0.00$ | $53.71 \pm 4.38$ | $73.93 \pm 2.67$ | $100.00 \pm 0.00$ | $19.70 \pm 9.19$ | $75.05 \pm 18.16$ |
| | GNNSAFE | $31.15 \pm 10.74$ | $93.45 \pm 2.16$ | $97.59 \pm 0.71$ | $64.83 \pm 8.52$ | $83.73 \pm 4.22$ | $93.94 \pm 1.05$ |
| | EDBD | $\mathbf{14.99 \pm 14.53}$ | $\mathbf{97.54 \pm 1.81}$ | $\mathbf{98.68 \pm 0.85}$ | $\mathbf{46.68 \pm 14.83}$ | $\mathbf{94.10 \pm 1.71}$ | $\mathbf{98.11 \pm 0.41}$ |
| SIS | MSP | $40.47 \pm 8.78$ | $92.91 \pm 1.34$ | $93.22 \pm 1.59$ | $57.19 \pm 18.25$ | $91.48 \pm 2.88$ | $94.26 \pm 1.43$ |
| | ODIN | $95.52 \pm 8.97$ | $52.14 \pm 11.24$ | $69.25 \pm 6.79$ | $100.00 \pm 0.00$ | $20.71 \pm 9.38$ | $61.54 \pm 5.78$ |
| | Mahalanobis | $74.28 \pm 40.81$ | $68.97 \pm 25.99$ | $88.02 \pm 10.64$ | $86.86 \pm 28.32$ | $83.20 \pm 6.94$ | $\mathbf{94.87 \pm 1.96}$ |
| | Energy | $38.95 \pm 9.16$ | $93.27 \pm 1.34$ | $93.59 \pm 1.55$ | $58.12 \pm 18.99$ | $91.08 \pm 3.27$ | $94.07 \pm 1.57$ |
| | GPN | $100.00 \pm 0.00$ | $47.38 \pm 4.39$ | $69.93 \pm 6.55$ | $100.00 \pm 0.00$ | $22.41 \pm 13.41$ | $87.30 \pm 9.84$ |
| | GNNSAFE | $47.55 \pm 11.24$ | $90.91 \pm 2.78$ | $93.19 \pm 1.54$ | $73.94 \pm 8.07$ | $82.73 \pm 4.55$ | $90.17 \pm 1.64$ |
| | EDBD | $\mathbf{32.61 \pm 7.18}$ | $\mathbf{94.17 \pm 1.22}$ | $\mathbf{94.19 \pm 1.39}$ | $\mathbf{52.37 \pm 19.93}$ | $\mathbf{91.78 \pm 3.04}$ | $94.40 \pm 1.52$ |

## 5.2 LABEL LEAVE-OUT RESULTS

The results of label leave-out, the conventional OOD detection setting on graphs, are demonstrated in Table 1. In this setting, EDBD outperforms state-of-the-art methods across all metrics. Class homophily, which is the phenomenon where nodes within the same class tend to connect with each other, is inherent in many graph-structured datasets. Thus, since label leave-out treats a subset of classes as OOD, OOD nodes in this setting may also exhibit dense connections among themselves, similar to the pattern observed in spreading OOD detection. We confirm that EDBD achieves state-of-the-art performance across all the datasets.

## 5.3 SPREADING OOD DETECTION RESULTS

Table 2 shows the results of spreading OOD detection on the Spreading COVID-19 dataset. The results of spreading OOD detection on existing benchmark datasets are shown in Table 3. Our EDBD achieves state-of-the-art performance regardless of the epidemic model on the Spreading COVID-19, Cora, and LastFM Asia datasets. The proposed EDBD achieves the best performance across the evaluation metrics, regardless of the epidemic model, both under the single-seed setting and the multi-seed setting. GNNSAFE, which refines energies via neighborhood aggregation, suffers performance degradation compared to Energy, a method agnostic to graph structure, in terms of FPR95-T. This indicates that aggregating without consideration of energy distribution results in a performance drop. In contrast, EDBD consistently achieves superior performance across all experiments, demonstrating the effectiveness of the proposed energy-distribution-based aggregation scheme. Through the results

Table 4: Ablation study of EDBD on spreading OOD detection and label leave-out, conducted on Cora. **S** and **C** denote energy similarity matrix and energy consistency matrix, respectively.

*Label leave-out*

| Dataset | | Amazon-Photo | | | Coauthor-CS | | |
|---|---|---|---|---|---|---|---|
| **S** | **C** | FPR95($\downarrow$) | AUROC($\uparrow$) | AUPR($\uparrow$) | FPR95($\downarrow$) | AUROC($\uparrow$) | AUPR($\uparrow$) |
| ✗ | ✗ | $6.57 \pm 0.38$ | $97.36 \pm 0.04$ | $97.13 \pm 0.10$ | $11.31 \pm 1.69$ | $97.44 \pm 0.35$ | $99.06 \pm 0.13$ |
| ✗ | ✓ | $6.43 \pm 0.70$ | $97.47 \pm 0.07$ | $97.58 \pm 0.05$ | $10.98 \pm 1.85$ | $97.55 \pm 0.36$ | $99.11 \pm 0.12$ |
| ✓ | ✗ | $6.05 \pm 0.62$ | $97.41 \pm 0.06$ | $97.22 \pm 0.02$ | $10.66 \pm 1.65$ | $97.59 \pm 0.31$ | $99.10 \pm 0.09$ |
| ✓ | ✓ | $\mathbf{5.82 \pm 0.66}$ | $\mathbf{97.48 \pm 0.07}$ | $\mathbf{97.60 \pm 0.06}$ | $\mathbf{10.19 \pm 1.63}$ | $\mathbf{97.68 \pm 0.35}$ | $\mathbf{99.14 \pm 0.13}$ |

*Spreading OOD on COVID-19*

| Model | | SI | | | SIS | | |
|---|---|---|---|---|---|---|---|
| **S** | **C** | FPR95-T($\downarrow$) | AUROC-T($\uparrow$) | AUPR-T($\uparrow$) | FPR95-T($\downarrow$) | AUROC-T($\uparrow$) | AUPR-T($\uparrow$) |
| ✗ | ✗ | $69.76 \pm 15.23$ | $80.63 \pm 6.07$ | $87.40 \pm 4.12$ | $77.76 \pm 14.64$ | $76.02 \pm 12.65$ | $85.53 \pm 5.30$ |
| ✗ | ✓ | $56.25 \pm 18.02$ | $81.54 \pm 8.39$ | $88.10 \pm 5.57$ | $70.59 \pm 19.60$ | $76.11 \pm 14.31$ | $85.86 \pm 5.81$ |
| ✓ | ✗ | $54.86 \pm 17.60$ | $81.58 \pm 8.32$ | $88.16 \pm 5.59$ | $68.03 \pm 19.51$ | $76.16 \pm 13.97$ | $85.88 \pm 5.85$ |
| ✓ | ✓ | $\mathbf{54.67 \pm 17.58}$ | $\mathbf{81.60 \pm 8.35}$ | $\mathbf{88.22 \pm 5.59}$ | $\mathbf{67.42 \pm 19.75}$ | $\mathbf{76.22 \pm 14.58}$ | $\mathbf{85.94 \pm 5.94}$ |

*Spreading OOD on Cora*

| Model | | SI | | | SIS | | |
|---|---|---|---|---|---|---|---|
| **S** | **C** | FPR95-T($\downarrow$) | AUROC-T($\uparrow$) | AUPR-T($\uparrow$) | FPR95-T($\downarrow$) | AUROC-T($\uparrow$) | AUPR-T($\uparrow$) |
| ✗ | ✗ | $31.15 \pm 10.74$ | $93.45 \pm 2.16$ | $97.59 \pm 0.71$ | $47.55 \pm 11.24$ | $90.91 \pm 2.78$ | $93.19 \pm 1.54$ |
| ✗ | ✓ | $20.42 \pm 14.48$ | $96.77 \pm 1.80$ | $98.59 \pm 0.80$ | $38.92 \pm 10.04$ | $93.32 \pm 1.69$ | $94.07 \pm 1.41$ |
| ✓ | ✗ | $15.63 \pm 13.91$ | $97.51 \pm 1.78$ | $98.61 \pm 0.83$ | $32.84 \pm 7.31$ | $94.06 \pm 1.23$ | $94.10 \pm 1.39$ |
| ✓ | ✓ | $\mathbf{14.99 \pm 14.53}$ | $\mathbf{97.54 \pm 1.81}$ | $\mathbf{98.68 \pm 0.85}$ | $\mathbf{32.61 \pm 7.18}$ | $\mathbf{94.17 \pm 1.22}$ | $\mathbf{94.19 \pm 1.39}$ |

on the benchmark datasets, we confirm that EDBD is consistently effective in spreading OOD detection under various settings.

## 5.4 ABLATION STUDY

We conduct an extensive ablation study to analyze the effectiveness of the components in EDBD. We perform the ablation study under six experimental settings, including different tasks, datasets, and epidemic models. Table 4 demonstrates the results of the ablation study. We exclude **S** by replacing $\text{sim}(\mathbf{E}_i^{(0)}, \mathbf{E}_j^{(0)})$ in Equation (5) with 1. This implies that EDBD utilizes only graph structure during the energy aggregation process without considering energy similarity between two connected nodes. To exclude **C**, we simply set in Equation (4) to zero. The first rows of the tables in Table 4, where both **S** and **C**, correspond to the performance of GNNSAFE, as it relies solely on the graph structure during energy aggregation. We confirm that the two components significantly contribute to performance improvement. The combination of two components results in the best performance.

Appendix C provides a complexity analysis in terms of time and memory (in Appendix C.1), statistical analysis (in Appendix C.6), performance according to time stamps (in Appendix C.2), and extensive experiments that demonstrate the robustness of EDBD under realistic scenarios, including dynamic graph structures (in Appendix C.3) and missing features (in Appendix C.4).

## 6 CONCLUSION

In this paper, we introduce spreading OOD detection, a problem that facilitates the evaluation of node-level OOD detection under realistic settings. To highlight the significance of this problem, we then present the Spreading COVID-19 dataset, which allows node-level OOD detection methods to identify COVID-19 without prior knowledge of the virus. Moreover, we propose EDBD that leverages flexible aggregation depending on each node's circumstance. Our aggregation scheme is not only applicable to refining energy but also can be extended to the refinement of any scalar value with noise on a graph. However, our aggregation scheme is effective when connected nodes tend to have similar scalar values. In contrast to graph-level OOD detection, there is no dataset specifically designed for node-level OOD detection. We hope that our work will inspire the creation of datasets for practical node-level OOD detection research within the machine learning community. Since we employ SI and SIS models in this work, spreading OOD detection using complex epidemic models is left for future research.

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

## A   SPREADING COVID-19 DATASET

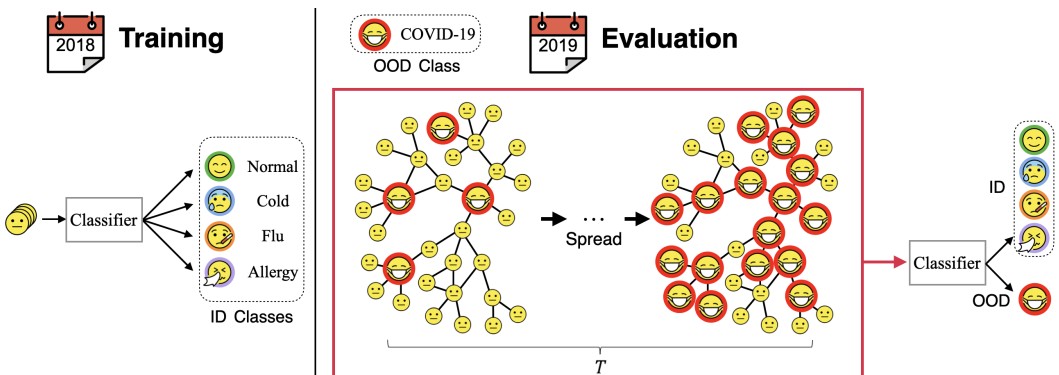

Figure 4: Illustration of spreading OOD detection on the Spreading COVID-19 dataset. A classifier can only be trained using ID data. During testing, OOD nodes emerge on a network and serve as seed nodes for spreading. OOD begins to spread from these seed nodes by infecting neighboring nodes. The objective of spreading OOD detection is to discriminate OOD nodes from ID nodes in a graph where the spread is occurring. Spreading OOD detection is a generalization of the existing OOD detection setting that tranforms randomly selected nodes into OOD nodes. (All emojis designed by OpenMoji.)

Spreading COVID-19 is a dataset for spreading OOD detection. In this dataset, ID classes are normal (no illness), allergies, cold, and flu. An OOD class is COVID-19. For each class, we generate samples with 23-dimensional features, created based on information about symptoms of respiratory illnesses (AAFA; Clinic). Features include the following symptoms: {sore throat, cough, muscle ache, rapid breathing, chest tightness, chill, runny nodes, stuffy nose, fever, nausea, vomiting, diarrhea, shortness of breath, difficulty breathing, loss of taste, loss of smell, itchy nose, itchy eyes, itchy mouth, itchy inner ear, sneezing, pink eye, tiredness}. These features are sampled based on the occurrence of each symptom in the respective diseases. For each illness class, we set the sampling probability for its corresponding symptoms (*i.e.*, features) to $0.5$. For normal class, we set the sampling probability equally to $0.01$ across all the symptoms. Through these processes, $\mathcal{D}_{in}^{train}$, $\mathcal{D}_{in}^{val}$, $\mathcal{D}_{in}^{test}$, and $\mathcal{D}_{out}$ are prepared. $\mathcal{D}_{in}^{train}$ and $\mathcal{D}_{in}^{val}$ contain $450$ and $50$ samples for each ID class, respectively.

We utilize the graph structure from the LastFM Asia graph (Rozemberczki & Sarkar, 2020), which has $7,624$ nodes. The LastFM Asia dataset is licensed under a Creative Commons Attribution 4.0 international (CC BY 4.0) license. This license allows for the sharing and adaptation of the dataset for any purpose, provided that the appropriate credit is given. We then assign samples randomly sampled from $\mathcal{D}_{in}^{test}$ to each node, representing an individual. We randomly assign $1,906$ samples to nodes for each ID class. To generate an episode of COVID-19 spreading, we randomly select a seed node and replace its features with those randomly sampled from $\mathcal{D}_{out}$ to simulate the node being infected with COVID-19. Using the SI and SIS models, the initial seed node attempts to infect neighboring nodes. We set $\gamma = 0.5$ for the SI model and $(\gamma, \delta) = (0.5, 0.1)$ for the SIS model. For each model, we create $15$ episodes. We conclude an episode when over $75\%$ of nodes are infected. As a result, we obtain a dataset for a single-seed setting. For a multi-seed setting, we repeat the aforementioned procedures by setting the number of initial seed nodes to $100$.

We provide the Spreading COVID-19 dataset in the supplementary material. This zip file contain four types of elements: (1)'graph_edge_index.npy': the graph structure; (2) 'x_allergy.npy', 'x_cold.npy', 'x_covid.npy'; 'x_flu.npy', 'x_normal.npy': generated symptom features for each class (3) 'graph_x.npy' ID-class features of the graph before COVID-19 spreading is applied; (4) npy format files in the 'episodes' folder: These files are matrices of shape $N$ by $T$, where the $t$-th column represents the infection status of the nodes at time stamp $t$. In this column, a value of 1 indicates infected, and a value of 0 indicates not infected. The license for the Spreading COVID-19 dataset is Creative Commons Attribution 4.0 international (CC BY 4.0).

**Justification on Using LastFM Asia for the Graph Structure.** Publicly available COVID-19 data with offline contact networks among individuals is lacking while there are many datasets containing infection populations over time and individual symptoms. To the best of our knowledge, except for a few small-sized networks, there are no publicly available datasets for offline social networks to meet a COVID-19 dataset. Dunbar et al. (2015) provides a rationale that online networks share very similar structural characteristics with offline networks. Based on this rationale, for the most realistic alternative to the offline social network, we utilize the online social network (LastFM Asia) to simulate the pandemic in our Spreading COVID-19 dataset. As demonstrated in Singh et al. (2021), our setting is more promising than using randomly generated networks as used in Alrasheed et al. (2020); Rafiq et al. (2023).

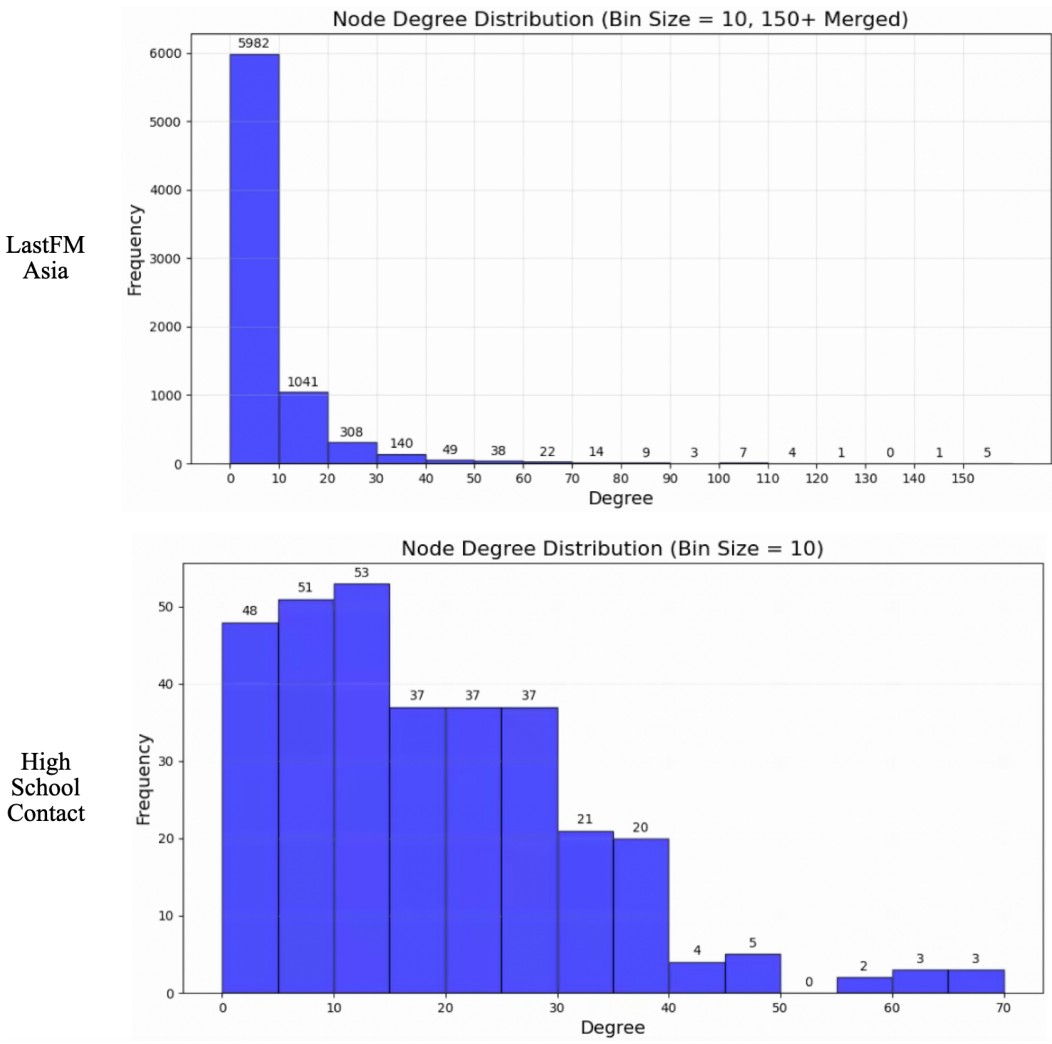

Figure 5: Node degree distributions of the LastFM Asia network and the high school contact network.

**Comparison between LastFM Asia and a human contact network.** The edges of the LastFM Asia dataset are constructed based on mutual follower relationships among users. Due to the lack of large human contact networks, we utilize the structure of this dataset to simulate the spread of COVID-19 in the proposed Spreading COVID-19 dataset, based on the rationale that online networks share structural characteristics similar to offline networks. To further validate the use of the graph structure of the LastFM Asia dataset, we thoroughly compare it with the High School Contact network (Mastrandrea et al., 2015).

Figure 5 compares the node degree distributions of the two real-world networks. Human contact networks often feature super-spreaders with high node degrees. As expected, the High School Contact network contains super-spreaders with node degrees of 50 or more, as shown in the figure. Similarly, the LastFM Asia network also includes high-degree nodes, with some exceeding degrees of 140, which can act as super-spreaders. Although the network sizes of the two datasets differ (327 nodes for the High School Contact network and 7624 nodes for the LastFM Asia network), both networks exhibit similar characteristics in terms of the presence of super-spreaders.

Furthermore, we compare the average episode duration, defined as the number of steps required for 75% of nodes to become infected. For the SI, SIS, and SEIR (Susceptible-Exposed-Infectious-Recovered) models, the High School Contact network requires 8.3, 10.1, and 18.4 steps, respectively, while the LastFM Asia network requires 10.0, 11.1, and 24.6 steps, respectively. Considering the difference in network sizes, we can confirm that the human contact network did not lead to faster spreading compared to the LastFM Asia network.

Although we demonstrate that the LastFM Asia dataset and the human contact network share similar properties in terms of graph structure, we further simulate COVID-19 spreading on the High School Contact network and perform spreading OOD detection to validate the effectiveness of EDBD in those scenarios as well (in Appendix C.8).

**Justification on Parameter Setting for Epidemic Models.** We selected the parameter values for the SI and SIS models based on recent work investigating COVID-19 dynamics (Eikenberry et al., 2020). In continuous-time SIS models, the ranges of the transmission rate $\gamma$ and the recovery rate $\delta$ are estimated to be $[0.5, 1.5](\text{day}^{-1})$ and $[1/30, 1/3](\text{day}^{-1})$, respectively. When converting the ranges of $\gamma$ and $\delta$ from continuous to discrete-time models, with the time interval set to one day, they become $[0.393, 0.777]$ and $[0.033, 0.283]$, respectively. Therefore, for the SIS model, we selected 0.5 for $\gamma$ and 0.1 for $\delta$ within these ranges. The SI model is the SIS model that approximates $\delta$ of 0.033 to zero. Moreover, it is easily feasible to construct new datasets by simply adjusting $\gamma$ and $\delta$ within the ranges.

**Difference between the Spreading COVID-19 dataset and general infectious disease datasets.** While general infectious disease datasets (Singh et al., 2021; Alrasheed et al., 2020; Rafiq et al., 2023) focus solely on the spreading patterns of infectious diseases, ours includes person-level symptom-based features. Furthermore, we provide ID classes for each sample: normal (no illness), allergies, cold, and flu. This setup enables the formulation of COVID-19 detection as OOD detection, where classifiers trained only on ID data attempt to detect the new disease, COVID-19. To the best of our knowledge, the Spreading COVID-19 dataset is the first COVID-19-related dataset to include both sample-level features and class labels.

# B EXPERIMENTAL DETAILS

## B.1 DATASET DETAILS

We conduct experiments on seven benchmark datasets, including Cora, Amazon-Photo, Amazon-Computers, Coauthor-CS, Spreading COVID-19, Cora with spreading OOD, and LastFM Asia with spreading OOD. Table 5 shows the statistics of datasets used in this paper. All the datasets are provided in Pytorch Geometric (Fey & Lenssen, 2019). The Cora dataset is a citation network, where nodes and edges represent publications and citation links, respectively. The LastFM dataset is a social network, with nodes as individuals and edges reflecting social connections between them. Amazon-photo is a recommendation network, where nodes represent goods, and an edge connects two nodes only when the goods are frequently bought together. Coauthor-CS is a co-authorship graph, where nodes are authors and an edge connects two authors if they co-authored a paper. The Cora, Amazon-Photo, Amazon-Computers, Coauthor-CS datasets are MIT-licensed. The LastFM Asia dataset is licensed under a Creative Commons Attribution 4.0 international (CC BY 4.0) license.

## B.2 IMPLEMENTATION DETAILS

We conduct all the experiments on a single NVIDIA GeForce RTX 2080 Ti GPU with 11GB memory and an Intel Core I5-6600 CPU @ 3.30 GHz. For training, we leverage Adam optimizer (Kingma & Ba, 2014) and set the maximum number of epochs to 200. We report test performance at an epoch

Table 5: Dataset statistics.

| Dataset | #Nodes | #Edges | #Features | #Classes |
|---|---|---|---|---|
| CORA | 2,708 | 5,429 | 1,433 | 7 |
| LASTFM ASIA | 7,624 | 27,806 | 128 | 18 |
| AMAZON-PHOTO | 7,650 | 238,162 | 745 | 8 |
| AMAZON-COMPUTERS | 13,752 | 491,722 | 767 | 10 |
| COAUTHOR-CS | 18,333 | 163,788 | 6805 | 15 |

which yields the lowest validation loss. Learning rates are selected within $\{0.01, 0.001, 0.0001\}$ by using a grid search. For a fair comparison, for MSP, ODIN, Mahalanobis, Energy, GNNSAFE, and EDBD, we use the same encoder backbone, a GCN with two layers and a hidden dimension of 64. For GKDE and GPN, we employ their public implementations and adhere to the hyperparameters reported in their respective paper.

### B.2.1 SPREADING OOD DETECTION

There is no dataset specifically designed for node-level OOD detection where OOD samples are separated. Thus, to perform spreading OOD detection on the Cora and LastFM Asia datasets, we first create additional OOD samples. We employ a Bernoulli distribution with $p = 0.1$ to generate features of these OOD samples (*i.e.,* $\mathbf{x} \sim \text{Ber}(0.1)$). Next, we partition samples within the original datasets into $\mathcal{D}_{in}^{train}$, $\mathcal{D}_{in}^{val}$, and $\mathcal{D}_{in}^{test}$. We exclude 20 nodes per each class. These excluded nodes are then divided into $\mathcal{D}_{in}^{train}$ and $\mathcal{D}_{in}^{val}$ at a ratio of 9:1. On a graph consisting of the remaining nodes in $\mathcal{D}_{in}^{test}$, we spread OOD samples randomly sampled from $\mathcal{D}_{out}$. Utilizing the SI or SIS model, we generate 15 episodes. We set $\gamma = 0.5$ for the SI model and $(\gamma, \delta) = (0.5, 0.1)$ for the SIS model. Among these episodes, 5 episodes are designated as the validation set and 10 episodes as the test set for OOD detection. Similar to Spreading COVID-19, for each dataset and epidemic model pair, we generate 15 episodes, each concluding when more than 75% of the nodes are infected. Among the 15 episodes, we allocate five episodes for the validation set and ten episodes for the test set.

### B.2.2 LABEL LEAVE-OUT

We follow the setting of label leave-out in Wu et al. (2023). The number of ID classes for Cora, Amazon-Photo, and Coauthor-CS is set to 3, 3, and 10, respectively. For a training/validation/test split on ID nodes in the Cora datasets, we adhere to the split used in Kipf & Welling (2016). For splits on ID nodes in the Amazon-Photo and Coauthor-CS datasets, we use random splits for training, validation, and test nodes with proportions of 0.1, 0.1, and 0.8, respectively.

### B.2.3 EDBD IMPLEMENTATION

We employ a grid search for hyperparameter tuning and EDBD-specific hyperparameters $(\alpha, \beta, \epsilon)$ are selected from $\{(\alpha, \beta, \epsilon) | \alpha \in \{0.1, 0.2, 0.3, 0.5\}, \beta \in \{1, \frac{1}{2}, \frac{1}{3}, \frac{1}{4}\}, \epsilon \in \{0.01, 0.05, 0.1, 0.5, 0.75\}\}$ based on validation sets. The number of aggregation, $K$, is chosen from $\{1, 2\}$.

### B.2.4 IMPLEMENTATION OF BASELINES

For all the baselines except for OODGAT, we utilize the implementations provided in the GitHub repository[3] released by Wu et al. (2023). All implementations strictly follow the descriptions provided in the respective paper. For OODGAT, we utilize the implementation provided in the GitHub repository[4]. Both publicly available repositories do not contain any statements regarding licenses. Across all the baselines, we adhere to the hyperparameter tuning strategies and settings described in their respective papers.

---

[3] https://github.com/qitianwu/GraphOOD-GNNSafe
[4] https://github.com/songyyyy/kdd22-oodgat

Table 6: Training and inference times (seconds).

| Method | CORA | | AMAZON-PHOTO | |
|---|---|---|---|---|
| | Tr. time | In. time | Tr. time | In. time |
| MSP | 0.006 | 0.010 | 0.007 | 0.021 |
| ODIN | 0.013 | 0.020 | 0.014 | 0.047 |
| Mahalanobis | 0.006 | 0.125 | 0.008 | 0.251 |
| Energy | 0.010 | 0.012 | 0.014 | 0.023 |
| GKDE | 0.007 | 0.018 | 0.008 | 0.026 |
| GPN | 0.273 | 0.959 | 1.035 | 3.004 |
| GNNSAFE | 0.010 | 0.013 | 0.014 | 0.024 |
| FastEDBD | 0.010 | 0.018 | 0.014 | 0.035 |
| EDBD | 0.010 | 0.062 | 0.014 | 0.438 |

Table 7: Memory usage for different datasets.

| Dataset | Memory usage (GB) |
|---|---|
| CORA | 1.199 |
| SPREADING COVID-19 | 1.203 |
| LASTFM ASIA | 1.195 |
| AMAZON-PHOTO | 1.451 |
| AMAZON-COMPUTERS | 1.609 |
| COAUTHOR-CS | 2.285 |
| OGBN-ARXIV | 3.551 |

## C ADDITIONAL EXPERIMENTS

### C.1 COMPLEXITY ANALYSIS

Here we discuss the time complexity of EDBD. Energy similarity-based aggregation calculates the energy similarity for each edge, leading to $O(|\mathcal{E}|)$ complexity. Then, energy consistency-based aggregation measures energy consistency for each node by considering its neighboring nodes. Hence, the complexity of energy consistency-based aggregation is $O(N^2)$. EDBD operates with a time complexity of $O(|\mathcal{E}| + N^2)$. To quantitatively demonstrate that EDBD, which has outstanding performance, is also efficient, we compare training and inference times of EDBD with those of baselines.

We empirically find that energy consistency-based aggregation is the computational bottleneck of EDBD. In Appendix 5.4, we confirm that EDBD without energy consistency-based aggregation also demonstrates good performance, although there is a performance loss in FPR95/FPR95-T compared to the original EDBD. Therefore, we suggest a light version of EDBD, called FastEDBD, for situations where fast inference is necessary. Table 6 shows training and inference times of methods including FastEDBD. Since methods using energy (Energy, GNNSafe, and EDBD) simply train classifiers with cross-entropy loss, they commonly have short training times. While EDBD exhibits long inference times, FastEDBD demonstrates substantial reductions in inference times. We recommend choosing between EDBD and FastEDBD based on what is more important in a given situation: performance or inference time.

The memory complexity of EDBD is $O(|\theta|) + O(B(F + C)) + O(N) + O(|\mathcal{E}|)$, where $\theta$ denotes trainable parameters. We conduct additional experiments on the OGBN-Arxiv dataset, which contains 169,343 nodes, representing a large-scale graph. We confirm that EDBD operates well on the OGBN-Arxiv dataset. The table 7 presents the memory usage of EDBD across various datasets. As shown in the table, EDBD maintains reasonable memory usage across different datasets, including the large-scale OGBN-Arxiv. This demonstrates EDBD's scalability in handling large-scale graphs.

We further compare the complexity of all methods used in this paper. Table 8 compares the input and memory complexity of scCR with other state-of-the-art methods. Although EDBD utilize additional

Table 8: Comparison of Big-O memory complexity. $\theta$ and $B$ represent the batch trainable parameters and the batch size, respectively.

| Method | Big-O |
|---|---|
| MSP | $O(|\theta|) + O(B(F + C))$ |
| ODIN | $O(|\theta|) + O(B(F + C))$ |
| Mahalanobis | $O(|\theta|) + O(B(F + C)) + O(d^2)$ |
| Energy | $O(|\theta|) + O(B(F + C))$ |
| GKDE | $O(|\theta|) + O(B(F + C)) + O(N) + O(|\mathcal{E}|)$ |
| GPN | $O(|\theta|) + O(B(F + C)) + O(N^2) + O(|\mathcal{E}|)$ |
| GNNSAFE | $O(|\theta|) + O(B(F + C)) + O(N) + O(|\mathcal{E}|)$ |
| EDBD | $O(|\theta|) + O(B(F + C)) + O(N) + O(|\mathcal{E}|)$ |

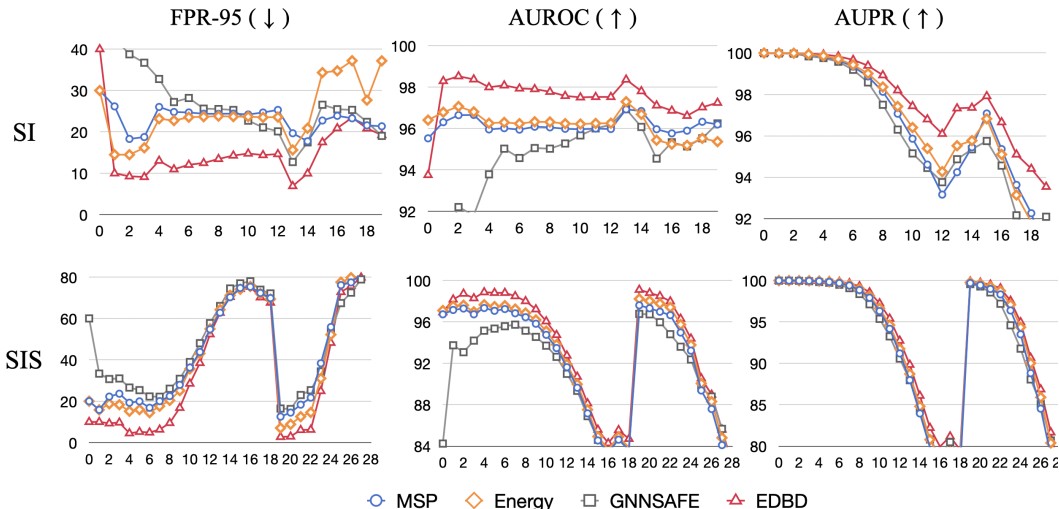

Figure 6: Performance on Spreading OOD detection according to time stamp $t$, measured by FPR-95 (%), AUROC(%), and AUPR(%).

matrices $\mathbf{S} \in \mathbb{R}^{N \times N}$ and $\mathbf{C} \in \mathbb{R}^{N \times N}$, $\mathbf{S}$ is a weighted adjacency matrix with non-zero values corresponding to the number of edges, and $\mathbf{C}$ is a diagonal matrix. Thus, the memory complexity of EDBD is $O(\theta) + O(B(F + C)) + O(N) + O(\mathcal{E})$, which is the same as that of GNNSAFE.

## C.2 PERFORMANCE ON SPREADING OOD DETECTION ACCORDING TO $t$

The goal of spreading OOD detection is to effectively discriminate OOD nodes in every time stamp. Therefore, we use FPR95-T, AUROC-T, and AUPR-T, which are metrics averaged over $T$, and we confirm that EDBD shows the outperforming performance in terms of the three metrics. However, to thoroughly analyze the performance gains achieved by EDBD, it is necessary to evaluate the performance at each time stamp $t$. Figure 6 demonstrates the performance of node-level OOD methods on the Cora dataset according to time stamp $t$, in terms of FPR95, AUROC, and AUPR. The three most competitive baselines, MSP, Energy, and GNNSAFE, are compared with our EDBD. We find that EDBD consistently surpasses the comparing methods across time stamp $t$, regardless the epidemic model and used metric. We observe that performance show significant fluctuations in the latter part of time stamp $t$. This is due to the varing end points of 10 episodes. Since the performance is calculated as the average of the 10 episodes, fluctuations arise due to episodes that conclude earlier.

## C.3 HIGHLY DYNAMIC GRAPH STRUCTURES

We conduct additional experiments evaluating the robustness of our EDBD on highly dynamic graph structures. In these experiments, we vary the dynamicity of the graph by removing a certain percentage of existing edges and generating the same number of new edges at each timestamp of the

Table 9: Performance of EDBD on Spreading OOD detection under dynamic graph structure scenarios.

**Spreading OOD detection on Cora (SI)**

| $p$ | Disappeared ratio | New ratio | FPR95-T($\downarrow$) | AUROC-T($\uparrow$) | AUPR-T($\uparrow$) |
|---|---|---|---|---|---|
| 0 | 0.00% | 0.00% | $14.99 \pm 14.53$ | $97.54 \pm 1.81$ | $98.68 \pm 0.85$ |
| 0.01 | 10.6% | 10.6% | $18.04 \pm 14.52$ | $98.31 \pm 0.99$ | $97.23 \pm 1.73$ |
| 0.02 | 18.7% | 18.7% | $22.04 \pm 13.33$ | $96.71 \pm 1.69$ | $98.40 \pm 0.91$ |
| 0.05 | 39.5% | 39.5% | $23.43 \pm 15.14$ | $96.28 \pm 2.17$ | $98.13 \pm 0.77$ |
| 0.1 | 63.5% | 63.5% | $23.37 \pm 14.72$ | $96.26 \pm 1.83$ | $97.94 \pm 0.78$ |
| 0.2 | 86.6% | 86.6% | $27.23 \pm 15.19$ | $95.50 \pm 2.13$ | $97.82 \pm 0.90$ |
| 0.5 | 99.6% | 99.6% | $31.21 \pm 14.79$ | $94.81 \pm 2.47$ | $97.64 \pm 0.78$ |

**Spreading OOD detection on Cora (SIS)**

| $p$ | Disappeared ratio | New ratio | FPR95-T($\downarrow$) | AUROC-T($\uparrow$) | AUPR-T($\uparrow$) |
|---|---|---|---|---|---|
| 0 | 0.00% | 0.00% | $32.61 \pm 7.18$ | $94.17 \pm 1.22$ | $94.19 \pm 1.39$ |
| 0.01 | 12.3% | 12.3% | $26.98 \pm 9.44$ | $95.45 \pm 0.91$ | $95.89 \pm 0.66$ |
| 0.02 | 21.7% | 21.7% | $29.93 \pm 7.49$ | $95.02 \pm 0.89$ | $95.63 \pm 0.68$ |
| 0.05 | 44.1% | 44.1% | $28.69 \pm 5.02$ | $94.96 \pm 0.52$ | $95.55 \pm 0.83$ |
| 0.1 | 69.5% | 69.5% | $33.23 \pm 11.43$ | $94.41 \pm 1.26$ | $95.33 \pm 0.86$ |
| 0.2 | 90.2% | 90.2% | $31.90 \pm 6.68$ | $94.31 \pm 0.84$ | $95.56 \pm 0.85$ |
| 0.5 | 99.7% | 99.7% | $42.28 \pm 9.86$ | $92.64 \pm 1.63$ | $94.97 \pm 0.87$ |

Table 10: Performance of Feature Propagation (FP)+EDBD on Cora under feature missing scenarios for varying missing rates ($r_m$).

| Task | Label leave-out on Cora | | | Spreading OOD Detection (SI) | | |
|---|---|---|---|---|---|---|
| $r_m$ | FPR95($\downarrow$) | AUROC($\uparrow$) | AUPR($\uparrow$) | FPR95-T($\downarrow$) | AUROC-T($\uparrow$) | AUPR-T($\uparrow$) |
| 0 | $30.48 \pm 1.11$ | $92.95 \pm 0.38$ | $82.31 \pm 0.38$ | $14.99 \pm 14.53$ | $97.54 \pm 1.81$ | $98.68 \pm 0.85$ |
| 0.2 | $21.15 \pm 3.23$ | $94.44 \pm 0.08$ | $85.79 \pm 0.97$ | $26.00 \pm 5.67$ | $95.52 \pm 1.09$ | $95.16 \pm 1.27$ |
| 0.4 | $21.63 \pm 2.03$ | $94.38 \pm 0.31$ | $84.35 \pm 1.19$ | $26.51 \pm 6.14$ | $95.50 \pm 1.13$ | $95.16 \pm 1.26$ |
| 0.6 | $32.30 \pm 5.67$ | $90.49 \pm 4.88$ | $83.36 \pm 0.34$ | $26.52 \pm 6.13$ | $95.44 \pm 1.17$ | $95.16 \pm 1.26$ |
| 0.8 | $43.28 \pm 7.04$ | $91.98 \pm 0.72$ | $82.73 \pm 0.47$ | $27.60 \pm 7.36$ | $95.36 \pm 1.24$ | $95.16 \pm 1.26$ |
| 0.9 | $47.18 \pm 9.96$ | $90.57 \pm 1.75$ | $80.17 \pm 3.40$ | $27.01 \pm 5.93$ | $95.48 \pm 1.05$ | $95.16 \pm 1.26$ |
| 0.99 | $49.54 \pm 11.02$ | $90.16 \pm 1.47$ | $80.40 \pm 2.79$ | $30.29 \pm 8.37$ | $94.88 \pm 1.41$ | $95.09 \pm 1.27$ |

spreading OOD process. Specifically, at each time stamp of spreading OOD, $p\%$ of the existing edges are removed and the same number of new edges are generated. Disappeared ratio and New ratio represent the percentage of edges that have disappeared and the percentage of new edges formed, respectively, when the OOD spread has finished. Table 9 shows the results of the experiments. Despite the minor degradation, EDBD maintains a reasonable performance level even under highly dynamic conditions, showcasing its robustness across various missing scenarios.

## C.4 MISSING FEATURES

Since missing data can occur not only in edges but also in features. Hence, we further consider incomplete features in graph-structured data, which are also prevalent in real-world scenarios. In this case, diffusion-based imputation methods (Rossi et al., 2022; Um et al., 2023; 2025b;a) (distinct from generative diffusion models (Ho et al., 2020; Dhariwal & Nichol, 2021; Yeo & Um, 2025)) that have demonstrated their effectiveness in graph-structured data can assist EDBD by imputing missing features. Table 10 presents the performance of EDBD using features imputed by FP, a diffusion-based imputation method, under missing feature scenarios. We can confirm that EDBE using FP yields reasonable performance in OOD detection tasks even with 99% missing features.

Table 11: Performance comparison of neighborhood mean, GNNSAFE, and our EDBD.

| Spreading OOD detection on Cora (SI) | | | |
|---|---|---|---|
| Method | FPR95-T($\downarrow$) | AUROC-T($\uparrow$) | AUPR-T($\uparrow$) |
| Neighborhood mean | $41.87 \pm 9.22$ | $86.95 \pm 3.52$ | $95.45 \pm 0.83$ |
| GNNSAFE | $31.15 \pm 10.74$ | $93.45 \pm 2.16$ | $97.59 \pm 0.71$ |
| EDBD | $\mathbf{14.99 \pm 14.53}$ | $\mathbf{97.54 \pm 1.81}$ | $\mathbf{98.68 \pm 0.85}$ |
| Spreading OOD detection on Cora (SIS) | | | |
| Method | FPR95-T($\downarrow$) | AUROC-T($\uparrow$) | AUPR-T($\uparrow$) |
| Neighborhood mean | $56.36 \pm 10.77$ | $86.12 \pm 4.48$ | $91.49 \pm 1.79$ |
| GNNSAFE | $47.15 \pm 11.24$ | $90.91 \pm 2.78$ | $93.19 \pm 1.54$ |
| EDBD | $\mathbf{32.61 \pm 7.18}$ | $\mathbf{94.17 \pm 1.22}$ | $\mathbf{94.19 \pm 1.39}$ |
| Label leave-out on Cora | | | |
| Method | FPR95($\downarrow$) | AUROC($\uparrow$) | AUPR($\uparrow$) |
| Neighborhood mean | $32.86 \pm 1.58$ | $92.67 \pm 0.37$ | $82.20 \pm 0.35$ |
| GNNSAFE | $31.31 \pm 1.11$ | $92.87 \pm 0.38$ | $82.22 \pm 0.40$ |
| EDBD | $\mathbf{30.48 \pm 1.11}$ | $\mathbf{92.95 \pm 0.38}$ | $\mathbf{82.31 \pm 0.38}$ |

## C.5 COMPARISON WITH NEIGHBORHOOD MEAN

As an entropy-like function for class scores from a trained neural network, the energy function is verified to be an effective indicator to distinguish OOD samples (Liu et al., 2020). In graph-structured data, connected two nodes often share similar properties, such as features and classes. Therefore, whether a node is OOD can be aided by its neighboring nodes. Motivated by this, GNNSAFE (Wu et al., 2023) refines the energy of each node by simply aggregating the energies from its neighboring nodes through multiple aggregation steps. Here, GNNSAFE performs energy aggregation based solely on the graph structure. However, an ID node can have OOD nodes in its neighbors and vice versa. In this case, this simple aggregation scheme of GNNSAFE, may lead to undesired energy mixing between ID and OOD nodes. The conceptually simplest way, neighborhood aggregation can be seen as a special case of GNNSAFE with a single aggregation step.

In contrast, our EDBD employs energy as an OOD indicator for a node during the energy aggregation process. This implies that before the aggregation process, EDBD briefly classifies nodes into OOD nodes and ID nodes to prevent the energy mixing between OOD nodes and ID nodes. EDBD allows the energies to control the entire aggregation process themselves, leading to significant performance improvement in both spreading OOD detection and conventional OOD detection tasks on graphs.

To verify that EDBD outperforms the conceptually simplest aggregation strategy, which is referred to as neighborhood mean, we conduct additional experiments in spreading OOD detection and conventional OOD detection tasks. Table 11 shows performance comparison of neighborhood mean, GNNSAFE, and EDBD in label leave-out (conventional OOD detection) and spreading OOD detection using SI and SIS models. As shown in the table, EDBD demonstrates its superiority, irrespective of tasks and metrics.

Table 12: $p$-values comparing our EDBD to the runner-up in each setting. * denotes state-of-the-art, not a runner-up.

**For Table 1**

| Dataset | Metric | Runner-up | Runner-up's | Ours | $p$-value |
|---|---|---|---|---|---|
| Cora | FPR95($\downarrow$) | GNNSAFE | $31.31 \pm 1.11$ | $\mathbf{30.48 \pm 1.11}$ | $0.119$ |
| | AUROC($\uparrow$) | GNNSAFE | $92.84 \pm 0.38$ | $\mathbf{92.95 \pm 0.38}$ | $0.583$ |
| | AUPR($\uparrow$) | GNNSAFE | $82.22 \pm 0.40$ | $\mathbf{82.31 \pm 0.38}$ | $0.656$ |
| Photo | FPR95($\downarrow$) | GNNSAFE | $6.57 \pm 0.38$ | $\mathbf{5.82 \pm 0.66}$ | $6.61 \times 10^{-2}$ |
| | AUROC($\uparrow$) | GNNSAFE | $97.36 \pm 0.04$ | $\mathbf{97.48 \pm 0.07}$ | $6.90 \times 10^{-4}$ |
| | AUPR($\uparrow$) | GNNSAFE | $97.13 \pm 0.10$ | $\mathbf{97.60 \pm 0.06}$ | $1.73 \times 10^{-6}$ |
| Computers | FPR95($\downarrow$) | GNNSAFE | $39.94 \pm 6.84$ | $\mathbf{35.59 \pm 6.94}$ | $0.137$ |
| | AUROC($\uparrow$) | GNNSAFE | $89.75 \pm 1.79$ | $\mathbf{92.45 \pm 0.98}$ | $3.60 \times 10^{-2}$ |
| | AUPR($\uparrow$) | GNNSAFE | $85.63 \pm 3.36$ | $\mathbf{90.34 \pm 1.45}$ | $3.36 \times 10^{-3}$ |
| Coauthor-CS | FPR95($\downarrow$) | GNNSAFE | $11.31 \pm 1.69$ | $\mathbf{10.19 \pm 1.63}$ | $0.152$ |
| | AUROC($\uparrow$) | GNNSAFE | $97.44 \pm 0.35$ | $\mathbf{97.68 \pm 0.35}$ | $0.123$ |
| | AUPR($\uparrow$) | GNNSAFE | $99.06 \pm 0.13$ | $\mathbf{99.14 \pm 0.13}$ | $0.257$ |

**For Table 2 (single-seed)**

| Model | Metric | Runner-up | Runner-up's | Ours | $p$-value |
|---|---|---|---|---|---|
| SI | FPR95-T($\downarrow$) | Energy | $54.82 \pm 17.43$ | $\mathbf{54.67 \pm 17.58}$ | $0.981$ |
| | AUROC-T($\uparrow$) | Energy | $80.46 \pm 8.54$ | $\mathbf{81.60 \pm 8.35}$ | $0.784$ |
| | AUPR-T($\uparrow$) | Energy | $88.00 \pm 5.52$ | $\mathbf{88.22 \pm 5.59}$ | $0.948$ |
| SIS | FPR95-T($\downarrow$) | Energy | $67.94 \pm 18.98$ | $\mathbf{67.42 \pm 19.75}$ | $0.959$ |
| | AUROC-T($\uparrow$) | Energy | $73.94 \pm 14.49$ | $\mathbf{76.22 \pm 14.58}$ | $0.769$ |
| | AUPR-T($\uparrow$) | Energy | $85.51 \pm 5.70$ | $\mathbf{85.94 \pm 5.94}$ | $0.896$ |

**For Table 2 (multi-seed)**

| Model | Metric | Runner-up | Runner-up's | Ours | $p$-value |
|---|---|---|---|---|---|
| SI | FPR95-T($\downarrow$) | Energy | $49.78 \pm 14.73$ | $\mathbf{49.62 \pm 15.22}$ | $0.982$ |
| | AUROC-T($\uparrow$) | Energy | $82.66 \pm 6.79$ | $\mathbf{83.29 \pm 6.68}$ | $0.850$ |
| | AUPR-T($\uparrow$) | GNNSAFE | $86.11 \pm 4.33$ | $\mathbf{86.11 \pm 5.01}$ | $1.000$ |
| SIS | FPR95-T($\downarrow$) | Energy | $60.40 \pm 10.33$ | $\mathbf{59.93 \pm 10.47}$ | $0.921$ |
| | AUROC-T($\uparrow$) | Energy | $80.18 \pm 6.24$ | $\mathbf{80.68 \pm 6.16}$ | $0.885$ |
| | AUPR-T($\uparrow$) | Energy | $81.63 \pm 4.84$ | $\mathbf{81.70 \pm 4.92}$ | $0.973$ |

**For Table 3 (SI)**

| Dataset | Metric | Runner-up | Runner-up's | Ours | $p$-value |
|---|---|---|---|---|---|
| Cora | FPR95-T($\downarrow$) | Energy | $22.79 \pm 18.21$ | $\mathbf{14.99 \pm 14.53}$ | $0.290$ |
| | AUROC-T($\uparrow$) | Energy | $96.36 \pm 2.12$ | $\mathbf{97.54 \pm 1.81}$ | $0.269$ |
| | AUPR-T($\uparrow$) | Energy | $98.03 \pm 1.07$ | $\mathbf{98.68 \pm 0.85}$ | $0.191$ |
| LastFM Asia | FPR95-T($\downarrow$) | MSP | $52.98 \pm 0.41$ | $\mathbf{46.68 \pm 14.83}$ | $0.236$ |
| | AUROC-T($\uparrow$) | MSP | $93.71 \pm 1.80$ | $\mathbf{94.10 \pm 1.71}$ | $0.589$ |
| | AUPR-T($\uparrow$) | MSP | $97.96 \pm 0.44$ | $\mathbf{98.11 \pm 0.41}$ | $0.557$ |

**For Table 3 (SIS)**

| Dataset | Metric | Runner-up | Runner-up's | Ours | $p$-value |
|---|---|---|---|---|---|
| Cora | FPR95-T($\downarrow$) | Energy | $38.95 \pm 9.16$ | $\mathbf{32.61 \pm 7.18}$ | $2.72 \times 10^{-2}$ |
| | AUROC-T($\uparrow$) | Energy | $93.27 \pm 1.34$ | $\mathbf{94.17 \pm 1.22}$ | $0.189$ |
| | AUPR-T($\uparrow$) | Energy | $93.59 \pm 1.55$ | $\mathbf{94.19 \pm 1.39}$ | $0.447$ |
| LastFM Asia | FPR95-T($\downarrow$) | MSP | $57.19 \pm 18.25$ | $\mathbf{52.37 \pm 19.93}$ | $0.501$ |
| | AUROC-T($\uparrow$) | MSP | $91.48 \pm 2.88$ | $\mathbf{91.78 \pm 3.04}$ | $0.814$ |
| | AUPR-T($\uparrow$) | Mahalanobis | $\mathbf{94.87 \pm 1.96}^{*}$ | $94.40 \pm 1.52$ | $0.618$ |

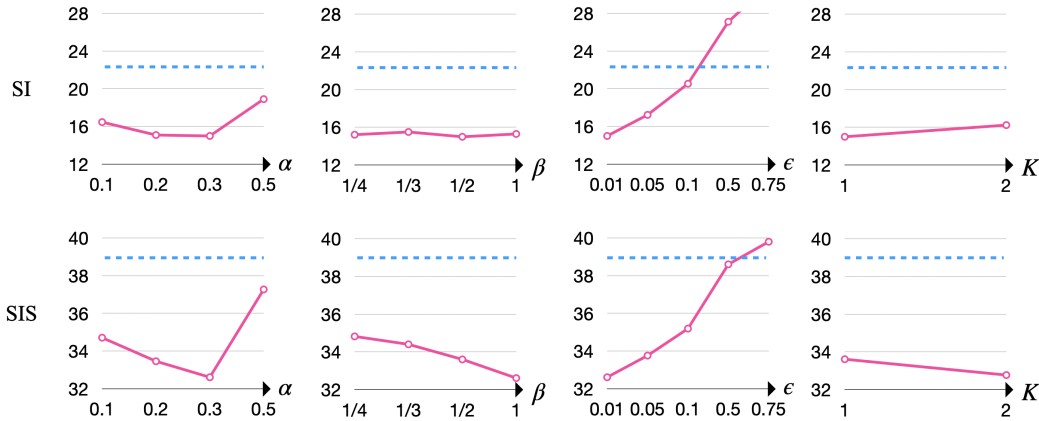

Figure 7: Performance of EDBD in spreading OOD detection, measured by FPR95-T (where lower values indicate better performance), on the Cora dataset is evaluated under two settings using SI and SIS models, respectively, with varying values of $\alpha$, $\beta$, $\epsilon$, and $K$, under two settings using SI and SIS models, respectively. The blue dashed lines represent existing state-of-the-art performance.

Table 13: Performance on Spreading OOD detection in the High School Contact dataset. The average FPR95-T (%), average AUROC-T (%), and average AUPR-T (%) across 10 episodes are reported with standard deviation errors.

| Epidemic model | SI | | | SIS | | |
|---|---|---|---|---|---|---|
| Method | FPR95-T ($\downarrow$) | AUROC-T ($\uparrow$) | AUPR-T ($\uparrow$) | FPR95-T ($\downarrow$) | AUROC-T ($\uparrow$) | AUPR-T ($\uparrow$) |
| MSP | $96.93 \pm 2.94$ | $36.60 \pm 8.39$ | $62.83 \pm 17.40$ | $96.95 \pm 2.67$ | $37.77 \pm 7.19$ | $61.70 \pm 15.19$ |
| ODIN | $90.92 \pm 10.03$ | $63.40 \pm 8.39$ | $73.51 \pm 15.63$ | $91.21 \pm 9.72$ | $62.23 \pm 7.19$ | $72.10 \pm 15.19$ |
| Mahalanobis | $71.15 \pm 33.90$ | $64.08 \pm 18.19$ | $72.07 \pm 22.60$ | $72.30 \pm 31.10$ | $62.80 \pm 28.62$ | $70.98 \pm 21.96$ |
| Energy | $55.82 \pm 13.72$ | $82.57 \pm 7.89$ | $89.36 \pm 7.22$ | $67.21 \pm 8.69$ | $79.21 \pm 6.87$ | $86.05 \pm 9.00$ |
| GPN | $81.94 \pm 3.36$ | $71.28 \pm 1.69$ | $75.74 \pm 1.41$ | $86.40 \pm 2.30$ | $66.92 \pm 1.20$ | $71.85 \pm 0.65$ |
| GNNSAFE | $59.05 \pm 15.41$ | $82.90 \pm 5.70$ | $89.49 \pm 7.40$ | $68.39 \pm 14.80$ | $80.96 \pm 6.28$ | $86.92 \pm 9.91$ |
| EDBD | $\mathbf{53.88 \pm 19.95}$ | $\mathbf{83.13 \pm 8.21}$ | $\mathbf{89.63 \pm 7.09}$ | $\mathbf{66.90 \pm 12.97}$ | $\mathbf{81.10 \pm 7.11}$ | $\mathbf{87.36 \pm 8.42}$ |

## C.6 STATISTICAL ANALYSIS

We conduct additional experiments to evaluate the statistical significance of our EDBD's superior performance. Table 12 shows $p$-values comparing EDBD to the runner-up in each setting for all the results in Tables 1, 2, and 3. As shown in the table, the $p$-values for Table 2 and Table 3 are higher compared to those of Table 1. However, spreading OOD detection tasks involve performing spreading simulations for each run, resulting in varying spreading patterns for OOD. These different spreading patterns significantly impact the difficulty level of each run, causing substantial variations in the overall performance of methods and consequently leading to large standard deviations. When the standard deviation is high, the statistical test does not show a significant difference between methods, resulting in higher $p$-values, even if the mean performance differs between our EDBD and baselines. Nevertheless, while the runner-ups change depending on the setting, EDBD consistently outperforms in all settings and metrics, except for AUPR-T in spreading OOD detection on LastFM Asia using SIS model. This consistency demonstrates its generalizability across various graph OOD-related tasks.

## C.7 HYPERPARAMETER SENSITIVITY

We conduct additional experiments to provide a comprehensive analysis of the impact of different hyperparameters, including $\alpha$, $\beta$, $\epsilon$, and $K$ on the performance of EDBD. Under two settings using SI and SIS models, respectively, we report FPR95-T in spreading OOD detection on the Cora dataset. Figure 7 shows the results. When varing each hyperparameter, the other hyperparameters are set to their optimal values. Compared to existing state-of-the-art performance (22.79% and 38.95% for the SI and SIS settings, respectively), EDBD models consistently exceed it by a considerable margin regardless of the values of $\alpha$, $\beta$, and $K$. While EDBD with $\epsilon \in \{0.75\}$ commonly shows worse

Table 14: Performance on Spreading OOD detection in the Spreading COVID-19 dataset using SIS models with various combinations of $(\gamma, \delta)$. The average FPR95-T (%), average AUROC-T (%), and average AUPR-T (%) across 10 episodes are reported with standard deviation errors.

| $(\gamma, \delta)$ | (0.7, 0.1) | | | (0.3, 0.1) | | |
|---|---|---|---|---|---|---|
| Method | FPR95-T ($\downarrow$) | AUROC-T ($\uparrow$) | AUPR-T ($\uparrow$) | FPR95-T ($\downarrow$) | AUROC-T ($\uparrow$) | AUPR-T ($\uparrow$) |
| MSP | $98.22 \pm 1.34$ | $33.12 \pm 6.09$ | $72.66 \pm 3.52$ | $98.43 \pm 1.41$ | $34.66 \pm 8.26$ | $66.97 \pm 16.67$ |
| ODIN | $91.68 \pm 5.43$ | $66.88 \pm 6.09$ | $82.44 \pm 3.94$ | $93.04 \pm 4.68$ | $65.34 \pm 8.26$ | $76.63 \pm 12.29$ |
| Mahalanobis | $79.52 \pm 18.92$ | $60.93 \pm 42.25$ | $75.08 \pm 14.05$ | $79.64 \pm 19.21$ | $57.19 \pm 36.43$ | $72.54 \pm 20.73$ |
| Energy | $64.44 \pm 20.22$ | $74.62 \pm 15.09$ | $87.54 \pm 5.46$ | $71.33 \pm 27.08$ | $72.02 \pm 14.51$ | $80.24 \pm 11.66$ |
| GPN | $95.48 \pm 1.58$ | $58.22 \pm 17.61$ | $83.47 \pm 4.44$ | $94.18 \pm 1.18$ | $55.28 \pm 15.30$ | $68.32 \pm 7.95$ |
| GNNSAFE | $73.43 \pm 16.52$ | $75.96 \pm 13.13$ | $\mathbf{88.15 \pm 5.22}$ | $86.88 \pm 10.52$ | $73.60 \pm 12.81$ | $80.12 \pm 11.45$ |
| EDBD | $\mathbf{62.76 \pm 21.16}$ | $\mathbf{76.31 \pm 15.09}$ | $87.69 \pm 5.67$ | $\mathbf{70.02 \pm 26.98}$ | $\mathbf{73.65 \pm 13.97}$ | $\mathbf{80.33 \pm 11.69}$ |
| $(\gamma, \delta)$ | (0.5, 0.2) | | | (0.5, 0.05) | | |
| Method | FPR95-T ($\downarrow$) | AUROC-T ($\uparrow$) | AUPR-T ($\uparrow$) | FPR95-T ($\downarrow$) | AUROC-T ($\uparrow$) | AUPR-T ($\uparrow$) |
| MSP | $98.01 \pm 1.52$ | $37.40 \pm 6.56$ | $66.52 \pm 11.38$ | $98.19 \pm 1.52$ | $35.22 \pm 7.51$ | $73.05 \pm 9.67$ |
| ODIN | $92.94 \pm 4.69$ | $62.60 \pm 6.56$ | $75.37 \pm 9.07$ | $92.64 \pm 5.42$ | $64.78 \pm 7.51$ | $82.37 \pm 7.00$ |
| Mahalanobis | $78.94 \pm 16.92$ | $62.60 \pm 6.56$ | $75.37 \pm 9.07$ | $78.19 \pm 19.03$ | $61.02 \pm 42.42$ | $77.81 \pm 7.84$ |
| Energy | $75.25 \pm 16.88$ | $70.70 \pm 12.32$ | $80.22 \pm 8.82$ | $67.77 \pm 22.89$ | $73.99 \pm 15.31$ | $87.57 \pm 7.90$ |
| GPN | $95.28 \pm 1.88$ | $60.23 \pm 15.36$ | $75.82 \pm 8.69$ | $95.08 \pm 2.09$ | $61.56 \pm 18.04$ | $82.99 \pm 10.26$ |
| GNNSAFE | $82.27 \pm 12.96$ | $73.58 \pm 12.32$ | $80.26 \pm 8.55$ | $77.43 \pm 17.52$ | $78.18 \pm 14.14$ | $87.65 \pm 7.65$ |
| EDBD | $\mathbf{73.72 \pm 17.56}$ | $\mathbf{73.68 \pm 12.03}$ | $\mathbf{80.35 \pm 8.90}$ | $\mathbf{66.82 \pm 23.73}$ | $\mathbf{78.45 \pm 14.80}$ | $\mathbf{87.86 \pm 7.94}$ |

Table 15: Performance on Spreading OOD detection for RSV. The average FPR95-T (%), average AUROC-T (%), and average AUPR-T (%) across 10 episodes are reported with standard deviation errors.

| Epidemic model | SI | | | SIS | | |
|---|---|---|---|---|---|---|
| Method | FPR95-T ($\downarrow$) | AUROC-T ($\uparrow$) | AUPR-T ($\uparrow$) | FPR95-T ($\downarrow$) | AUROC-T ($\uparrow$) | AUPR-T ($\uparrow$) |
| MSP | $85.48 \pm 7.42$ | $63.93 \pm 6.13$ | $80.99 \pm 3.15$ | $86.33 \pm 6.74$ | $63.09 \pm 5.59$ | $79.87 \pm 2.30$ |
| ODIN | $99.97 \pm 0.05$ | $36.07 \pm 6.13$ | $70.01 \pm 4.34$ | $99.97 \pm 0.05$ | $36.91 \pm 5.59$ | $69.34 \pm 3.13$ |
| Mahalanobis | $75.35 \pm 32.56$ | $50.79 \pm 38.40$ | $79.24 \pm 14.42$ | $75.55 \pm 32.59$ | $50.72 \pm 38.38$ | $78.30 \pm 15.19$ |
| Energy | $72.18 \pm 15.73$ | $72.46 \pm 9.39$ | $84.00 \pm 5.29$ | $75.31 \pm 13.36$ | $71.34 \pm 8.96$ | $82.99 \pm 3.96$ |
| GPN | $92.67 \pm 8.44$ | $53.38 \pm 13.79$ | $84.18 \pm 2.13$ | $93.17 \pm 7.47$ | $54.15 \pm 12.89$ | $81.81 \pm 1.98$ |
| GNNSAFE | $80.99 \pm 16.19$ | $74.10 \pm 9.07$ | $84.69 \pm 5.07$ | $74.54 \pm 14.23$ | $71.71 \pm 8.56$ | $83.60 \pm 3.50$ |
| EDBD | $\mathbf{70.94 \pm 21.05}$ | $\mathbf{74.18 \pm 9.51}$ | $\mathbf{84.74 \pm 5.45}$ | $\mathbf{73.96 \pm 18.11}$ | $\mathbf{72.75 \pm 9.01}$ | $\mathbf{83.69 \pm 4.06}$ |

performance compared to the existing state-of-the-art performance, the small value of $\epsilon$ indicates that energy aggregation focuses less on energy similarity and more on the graph structure. This result highlights the importance of considering energy distribution in the energy aggregation process. As mentioned in Appendix B.2.3, we determine the hyperparameters of EDBD for each setting using grid search within fixed ranges, based on performance on the validation set.

## C.8 SPREADING COVID-19 ON A CONTACT NETWORK

To validate EDBD's effectiveness in highly realistic scenarios, we conduct additional experiments using a contact network dataset collected in a high school in France (Mastrandrea et al., 2015). Since the original dataset comprises numerous networks collected over short time intervals, we aggregate them into a single network. We then measure the performance of methods on spreading OOD detection by replacing only the graph structure in the Spreading COVID-19 dataset with this network.

Table 13 presents the results when SI and SIS models are used, respectively. As shown in the table, our EDBD consistently outperforms state-of-the-art methods across all metrics and epidemic models. These results demonstrate its effectiveness even in highly realistic scenarios.

## C.9 SIS MODELS WITH VARIOUS PARAMETER COMBINATIONS

We adjust the parameters of the SIS model and compare the performance of the methods across various SIS models by performing spreading OOD detection on the Spreading COVID-19 dataset. Specifically, we conduct experiments by modifying the original $(\gamma, \delta)$ values of $(0.5, 0.1)$ to $(0.7, 0.1)$ and $(0.3, 0.1)$. Additionally, we performed experiments by changing the original $(\gamma, \delta)$ values to

Table 16: Performance of EDBD+similarity-based link imputation in spreading OOD detection on Cora under edge missing scenarios. The average FPR95-T (%), average AUROC-T (%), and average AUPR-T (%) across 10 episodes are reported with standard deviation errors.

| Method | EDBD | | | EDBD with Similarity-Based Link Imputation | | |
|---|---|---|---|---|---|---|
| Missing rate | FPR95-T ($\downarrow$) | AUROC-T ($\uparrow$) | AUPR-T ($\uparrow$) | FPR95-T ($\downarrow$) | AUROC-T ($\uparrow$) | AUPR-T ($\uparrow$) |
| 0 | $14.99 \pm 14.53$ | $97.54 \pm 1.81$ | $98.68 \pm 0.85$ | $14.99 \pm 14.53$ | $97.54 \pm 1.81$ | $98.68 \pm 0.85$ |
| 0.05 | $18.49 \pm 14.78$ | $97.07 \pm 2.10$ | $98.55 \pm 0.99$ | $21.37 \pm 17.38$ | $96.47 \pm 2.55$ | $98.48 \pm 0.98$ |
| 0.1 | $18.17 \pm 15.76$ | $97.00 \pm 2.13$ | $98.53 \pm 0.75$ | $17.76 \pm 16.75$ | $97.04 \pm 2.16$ | $98.62 \pm 0.88$ |
| 0.2 | $19.74 \pm 17.07$ | $96.88 \pm 1.94$ | $98.17 \pm 0.92$ | $15.74 \pm 17.60$ | $97.22 \pm 2.48$ | $98.55 \pm 0.75$ |
| 0.3 | $23.43 \pm 16.36$ | $96.06 \pm 2.79$ | $97.47 \pm 1.09$ | $15.88 \pm 16.26$ | $97.40 \pm 2.10$ | $98.78 \pm 0.69$ |
| 0.4 | $40.80 \pm 25.99$ | $91.84 \pm 10.05$ | $96.17 \pm 2.60$ | $16.20 \pm 17.09$ | $97.12 \pm 2.35$ | $98.66 \pm 0.90$ |
| 0.5 | $41.23 \pm 32.91$ | $90.62 \pm 12.02$ | $95.98 \pm 2.47$ | $23.00 \pm 30.39$ | $95.40 \pm 5.77$ | $98.63 \pm 0.82$ |

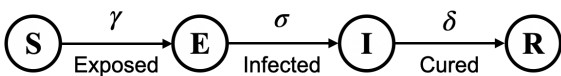

Figure 8: A state transition diagram of SEIR model. $S$, $E$, $I$, and $R$ represent the four possible state of a node, susceptible, exposed, infectious, and recovered, respectively.

$(0.5, 0.2)$ and $(0.5, 0.05)$. Table 14 presents the results. As shown in the table, except for AUPR-T under $(\gamma, \delta) = (0.7, 0.1)$, EDBD consistently outperforms state-of-the-art methods and demonstrates robustness against various spreading patterns.

### C.10 SPREADING OOD DETECTION FOR RSV

We conduct additional experiments to detect an urgent contagion, Respiratory Syncytial Virus (RSV), using spreading OOD detection. RSV is currently garnering significant attention due to a surge in infections and its particularly severe impact on infants and the elderly (Langedijk & Bont, 2023; Moline, 2024). We generate samples with features based on information about symptoms of respiratory illness (Healthline; Sample; Wallace), following the same procedure specified in Appendix A. We replace COVID-19 samples in the Spreading COVID-19 dataset with RSV samples, and compare the performance of methods in spreading OOD detection. Table 15 presents the results for RSV spreading scenarios using SI and SIS models. As shown in the table, EDBD outperforms state-of-the-art methods in both settings across all metrics. These results on the recent contagion show the generalizability of EDBD.

### C.11 INCOMPLETE GRAPH STRUCTURES

Since EDBD utilizes the graph structure for energy aggregation, we conduct additional experiments to evaluate its performance under edge-missing settings. As performance degradation is expected when edges are incomplete, we apply a simple missing link imputation technique to mitigate the degradation. This link imputation technique creates a single edge for each node by connecting it to the node with the highest feature similarity. Table 16 the results in spreading OOD detection on the Cora dataset. As shown in the table, while the performance of EDBD worsens as the missing rate increases, EDBD with the similarity-based link imputation effectively mitigates performance degradation. We confirm that EDBD with similarity-based link imputation maintains reasonable performance even with an edge missing rate of 50%.

### C.12 SPREADING OOD DETECTION USING SEIR MODELS

To validate the effectiveness of EDBD in OOD spreading using complex epidemic models, we conduct additional experiments. We adopt Susceptible-Exposed-Infectious-Recovered (SEIR) model (Rachah et al., 2018) as a complex epidemic model. As illustrated in Figure 8, which shows the state transition diagram of SEIR, each node can exist in one of four states: susceptible, exposed, infectious, or

Table 17: Performance on Spreading OOD detection in the Spreading COVID-19 dataset using SEIR model. The average FPR95-T (%), average AUROC-T (%), and average AUPR-T (%) across 10 episodes are reported with standard deviation errors.

| Epidemic model | SEIR | | |
|---|---|---|---|
| Method | FPR95-T ($\downarrow$) | AUROC-T ($\uparrow$) | AUPR-T ($\uparrow$) |
| MSP | $98.02 \pm 1.16$ | $35.46 \pm 6.21$ | $67.50 \pm 4.03$ |
| ODIN | $92.15 \pm 3.80$ | $64.54 \pm 6.00$ | $78.32 \pm 2.15$ |
| Mahalanobis | $79.56 \pm 18.93$ | $60.55 \pm 32.23$ | $82.78 \pm 15.47$ |
| Energy | $69.11 \pm 17.63$ | $73.56 \pm 13.14$ | $83.52 \pm 6.43$ |
| GPN | $92.27 \pm 2.98$ | $61.76 \pm 11.78$ | $44.59 \pm 5.41$ |
| GNNSAFE | $76.09 \pm 16.75$ | $75.93 \pm 12.38$ | $84.21 \pm 6.02$ |
| EDBD | $\mathbf{68.27 \pm 18.53}$ | $\mathbf{76.31 \pm 13.32}$ | $\mathbf{84.23 \pm 6.58}$ |

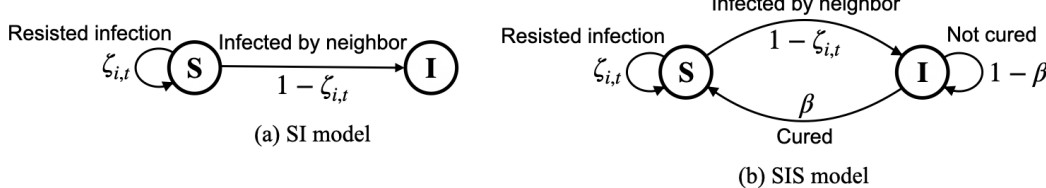

(a) SI model

(b) SIS model

Figure 9: Illustration of (a) SI model and (b) an SIS model, viewed from the perspective of a single node. **S** and **I** represent the two possible state of a node, susceptible and infected, respectively. In both models, $\zeta_{i,t}$ depends on $\gamma$ (epidemic transmission rate via an edge connected to an infected node) and the graph structure. This is because a susceptible node may be simultaneously exposed to infection attempts by multiple infected neighbors.

recovered. In SEIR model, a susceptible node can transition only to the exposed state, an exposed node can transition only to the infectious state, and an infectious node can transition only to the recovered state. The probabilities of these transitions are denoted by $\gamma$, $\sigma$, and $\delta$. Based on recent research investigating COVID-19 dynamics (Eikenberry et al., 2020), we set $\gamma$, $\sigma$, and $\delta$ to 0.5, 0.2, and 0.1, respectively. We follow the experimental setup for spreading OOD detection as specified in Appendix B.2.1. Table 17 demonstrates the results on spreading OOD detection in the Spreading COVID-19 dataset using the SEIR model. As shown in the figure, EDBD outperforms state-of-the-art methods across all metrics, demonstrating its superiority even in complex spreading scenarios.

## D STATE TRANSITION DIAGRAM OF SI AND SIS MODELS

Spreading processes using SI and SIS models can be modeled as a Markov chain because the configuration of $N$ nodes at the $t + 1$-th time stamp depends solely on their configuration at the $t$-th time stamp. Since each node can be in the two states, there exist $2^N$ possible configurations of this system at each time stamp. Figure 9 shows the state transition diagrams of SI model and SIS model, respectively.

## E THEORETICAL ANALYSIS

The key concept of EDBD is to enable the energies (OOD scores) to control their own update process through energy aggregation. In contrast, GNNSAFE, which performs energy aggregation based solely on the graph structure, cannot prevent the undesired mixing of energies between ID and OOD nodes. We provide a theoretical analysis to explain why GNNSAFE leads to this undesired mixing of energies. Before proceeding, it is important to note that the goal of energy aggregation is to enhance the separation between the energies of ID nodes and those of OOD nodes.

Since energies are positive values, we first assume that the energies of ID nodes and OOD nodes follow gamma distributions. Given that the energies of ID nodes tend to be smaller than those of OOD nodes, we define the energy distributions as follows: (1) ID energies have a mean $\mu_{in}$ and variance $\sigma^2$; (2) OOD energies have a mean $\mu_{out}$ and variance $\sigma^2$, where $\mu_{in} < \mu_{out}$. Additionally, we assume that the energies of nodes in each type are IID (independent and identically distributed). The energy update rule of GNNSAFE is defined as:

$$\mathbf{E}_i^{(1)} = \lambda \mathbf{E}_i^{(0)} + (1 - \lambda) \frac{1}{|\mathcal{N}(v_i)|} \sum_{j \in \mathcal{N}(v_i)} \mathbf{E}_j^{(0)}, \tag{8}$$

where $\lambda \in [0, 1]$ is a hyperparamter and $\mathcal{N}(v_i)$ denotes the set of neighbors of node $v_i$.

Consider an ID node $v_i$ with four neighbors, and analyze the following two cases: First, all four neighbors are ID nodes. Second, two neighbors are ID nodes, and the other two are OOD nodes.

In the first case,

$$\mathbb{E}[\mathbf{E}_i^{(1)}] = \lambda \mathbb{E}[\mathbf{E}_i^{(0)}] + (1 - \lambda) \frac{1}{4} \sum_{j \in \mathcal{N}(v_i)} \mathbf{E}_j^{(0)} = \lambda \mu_{in} + (1 - \lambda) \mu_{in} = \mu_{in}. \tag{9}$$

That is, the mean of the updated energy remains the same as the original mean, $\mu_{in}$.

In terms of variance, given the *i.i.d.* property, the neighbors' energies $\mathbf{E}_j^{(0)}$ are independent of each other and of $\mathbf{E}_i^{(0)}$, Therefore, the variance can be expressed as:

$$\text{Var}(\mathbf{E}_i^{(1)}) = \lambda^2 \text{Var}(\mathbf{E}_i^{(0)}) + (1 - \lambda)^2 \text{Var}\left(\frac{1}{4} \sum_{j \in \mathcal{N}(v_i)} \mathbf{E}_j^{(0)}\right). \tag{10}$$

We simplify the neighbor term and factor out $\sigma^2$:

$$\text{Var}(\mathbf{E}_i^{(1)}) = \sigma^2 \left(\lambda^2 + \frac{(1 - \lambda)^2}{4}\right). \tag{11}$$

Since $\lambda \in [0, 1]$, the term $\lambda^2 + \frac{(1-\lambda)^2}{4} = \frac{5\lambda^2}{4} - \frac{\lambda}{2} + \frac{1}{4} < 1$. This implies:

$$\text{Var}(\mathbf{E}_i^{(1)}) < \sigma^2. \tag{12}$$

This confirms that, in the first case, GNNSAFE ensures that the mean remains the same while the standard deviation decreases, probabilistically enhancing the discriminative energy of the ID node $v_i$ compared to the energies of OOD nodes.

For the second case, node $v_i$ has two ID neighbors and two OOD neighbors. Let us analyze the mean and variance of the updated energy $\mathbf{E}_i^{(1)}$. Using the energy update rule:

$$\mathbb{E}[\mathbf{E}_i^{(1)}] = \lambda \mathbb{E}[\mathbf{E}_i^{(0)}] + (1 - \lambda) \frac{1}{4} \sum_{j \in \mathcal{N}(v_i)} \mathbb{E}[\mathbf{E}_j^{(0)}]. \tag{13}$$

Substituting the expectations for ID ($\mu_{in}$) and OOD ($\mu_{out}$) nodes, and noting that two neighbors are ID nodes and two are OOD nodes:

$$\mathbb{E}[\mathbf{E}_i^{(1)}] = \lambda \mu_{in} + (1 - \lambda) \frac{1}{4} \left(2\mu_{in} + 2\mu_{out}\right). \tag{14}$$

Simplification gives:

$$\mathbb{E}[\mathbf{E}_i^{(1)}] = \lambda \mu_{in} + (1 - \lambda) \frac{1}{2} (\mu_{in} + \mu_{out}) = \mu_{in} + \frac{(1 - \lambda)}{2} (\mu_{out} - \mu_{in}). \tag{15}$$

By GNNSAFE, the mean of the ID node's energy shifts towards the higher energy values of the OOD nodes due to their influence in the aggregation term. This undesired mixing reduces the separation between ID and OOD energies, potentially decreasing the effectiveness of OOD detection.

The variance of $\mathbb{E}[\mathbf{E}_i^{(1)}]$ can be expressed as:

$$\text{Var}(\mathbf{E}_i^{(1)}) = \lambda^2 \text{Var}(\mathbf{E}_i^{(0)}) + (1-\lambda)^2 \text{Var}\left(\frac{1}{4} \sum_{j \in \mathcal{N}(v_i)} \mathbf{E}_j^{(0)}\right). \tag{16}$$

For the neighbor aggregation term:

$$\text{Var}\left(\frac{1}{4} \sum_{j \in \mathcal{N}(v_i)} \mathbf{E}_j^{(0)}\right) = \frac{1}{16} \sum_{j \in \mathcal{N}(v_i)} \text{Var}(\mathbf{E}_j^{(0)}) = \frac{1}{16}\left(2\sigma^2 + 2\sigma^2\right) = \frac{4\sigma^2}{16} = \frac{\sigma^2}{4}. \tag{17}$$

Finally,

$$\text{Var}(\mathbf{E}_i^{(1)}) = \lambda^2 \sigma^2 + (1-\lambda)^2 \frac{\sigma^2}{4} = \sigma^2 \left(\lambda^2 + \frac{(1-\lambda)^2}{4}\right). \tag{18}$$

As in the first case, the variance decreases compared to the initial variance $\sigma^2$, but the influence of the OOD neighbors shifts the energy distribution. However, the shift in the mean undermines the goal of energy aggregation, which is to enhance the separation between ID and OOD energies.

This analysis can be easily extended to the general case where the ID node $v_i$ has $n$ neighbors, among which $n_{in}$ are ID nodes and $n_{out}$ are OOD nodes ($n_{in} + n_{out} = n$). Let us derive the mean and variance for the updated energy $\mathbf{E}_i^{(1)}$ for this general case.

The energy update rule for $\mathbf{E}_i^{(1)}$ is given by:

$$\begin{aligned}
\mathbb{E}[\mathbf{E}_i^{(1)}] &= \lambda \mathbb{E}[\mathbf{E}_i^{(0)}] + (1-\lambda)\frac{1}{n} \sum_{j \in \mathcal{N}(v_i)} \mathbb{E}[\mathbf{E}_j^{(0)}] \\
&= \lambda \mu_{in} + (1-\lambda)\frac{1}{n}\left(n_{in}\mu_{in} + n_{out}\mu_{out}\right) \\
&= \lambda \mu_{in} + (1-\lambda)\left(\frac{n_{in}}{n}\mu_{in} + \frac{n_{out}}{n}\mu_{out}\right) \\
&= \mu_{in} + (1-\lambda)\frac{n_{out}}{n}(\mu_{out} - \mu_{in}).
\end{aligned} \tag{19}$$

Here, the term $(1-\lambda)\frac{n_{out}}{n}(\mu_{out} - \mu_{in})$ quantifies the shift in the mean of the ID node's energy due to the influence of OOD neighbors. The higher the proportion of OOD neighbors ($n_{out}/n$) or the larger the difference ($\mu_{out} - \mu_{in}$), the greater the shift in the mean towards the OOD energy values. This undesired shift undermines the separation between ID and OOD energies.

The variance of $\mathbf{E}_i^{(1)}$ is:

$$\begin{aligned}
\text{Var}(\mathbf{E}_i^{(1)}) &= \lambda^2 \text{Var}(\mathbf{E}_i^{(0)}) + (1-\lambda)^2 \text{Var}\left(\frac{1}{n} \sum_{j \in \mathcal{N}(v_i)} \mathbf{E}_j^{(0)}\right) \\
&= \lambda^2 \sigma^2 + (1-\lambda)^2 \text{Var}\left(\frac{1}{n} \sum_{j \in \mathcal{N}(v_i)} \mathbf{E}_j^{(0)}\right) \\
&= \lambda^2 \sigma^2 + (1-\lambda)^2 \frac{1}{n^2} \sum_{j \in \mathcal{N}(v_i)} \text{Var}(\mathbf{E}_j^{(0)}) \\
&= \lambda^2 \sigma^2 + (1-\lambda)^2 \frac{1}{n^2}(n_{in}\sigma^2 + n_{out}\sigma^2) \\
&= \lambda^2 \sigma^2 + (1-\lambda)^2 \frac{\sigma^2}{n} \\
&= \sigma^2 \left(\lambda^2 + \frac{(1-\lambda)^2}{n}\right).
\end{aligned} \tag{20}$$

Since $n \geq 1$, the term $\lambda^2 + \frac{(1-\lambda)^2}{n} \leq 1$, confirming that the variance of $\mathbf{E}_i^{(1)}$ is always reduced compared to the initial variance $\sigma^2$.

For the general case, the mean $\mathbb{E}[\mathbf{E}_i^{(1)}]$ is shifted towards OOD energies when OOD neighbors are present, reducing the separation between ID and OOD energies. The variance $\text{Var}(\mathbf{E}_i^{(1)})$ decreases, but the shift in the mean dominates, leading to the undesired mixing.

This theoretical analysis highlights why GNNSAFE's reliance on graph structure can lead to the undesired mixing of ID and OOD energies, reducing its effectiveness in maintaining energy separation. Furthermore, this analysis shows that ID nodes aggregating energy from surrounding ID nodes and OOD nodes aggregating energy from surrounding OOD nodes are beneficial for updating energies. Unlike GNNSAFE, EDBD prevents undesired energy mixing by allowing ID nodes to focus on their neighboring ID nodes and OOD nodes to focus on their neighboring OOD nodes during energy aggregation.

# F DISCUSSIONS

**What is the main challenge of spreading OOD detection?** The main challenge of spreading OOD detection is that only $\mathcal{G}^t$ with the graph structure and $\mathbf{X}^t$ are given without any additional information such as what the current $t$ is, how much spreading has occurred, and the location of the seed node. This is inspired by the fact that in most real-world OOD-related problems, information about the spreading is not provided. Accordingly, depending on $t$ and the spreading pattern of an episode, the best method can be different. Due to this challenge of the spreading OOD detection task, the goal is to perform OOD detection well under any spreading situations regardless of $t$. Hence, new three metrics for spreading OOD detection are FPR95-T, AUROC-T, and AUPR-T, which are the averaged FPR95, AUROC, and AUPR across each graph in $\{\mathcal{G}^t\}_{t \in \{1,...,T\}}$.

Despite this main challenge of the spreading OOD detection task, EDBD can address various spreading situations through its edge-level and node-level aggregation controllers, which are the energy similarity matrix and the energy consistency matrix, respectively. EDBD does not rely solely on the graph structure for energy aggregation; instead, it performs energy aggregation adaptively for each node according to its specific situation. Specifically, these two controllers tailor energy aggregation for each node based on the energy distribution of the node and its neighboring nodes. This energy-based energy aggregation process prevents undesired energy mixing between in-distribution (ID) nodes and OOD nodes.

**Comparison of spreading OOD detection and dynamic event detection.** While dynamic event detection (Zou et al., 2019; Zhang et al., 2022; Sridhar & Poor, 2022) seems similar to our spreading OOD detection in that they address the problem of dynamic events spreading through a graph, there are clear differences between dynamic event detection and our work from three perspectives. Firstly, dynamic event detection relies on prior knowledge of the distribution of anomalies (*i.e.*, OOD), such as what the distribution is or the shape of the distribution. In contrast, spreading OOD detection assumes that any information about OOD samples is inaccessible, which is the nature of anomalies or OOD in many real-world scenarios. Secondly, dynamic event detection assume that they completely know the distribution from which normal samples (*i.e.*, ID samples) are drawn. However, following OOD detection tasks (Liu et al., 2020; Rawat et al., 2021), we utilize only observed ID features and partial labels for ID samples without fully knowing the distribution. Lastly, studies on dynamic event detection (Zou et al., 2019; Zhang et al., 2022; Sridhar & Poor, 2022) use statistical models related to this distribution, while we use a neural network trained on ID data to determine OOD based on class scores.

**Does EDBA take into account information about temporal dynamics?** EDBA does not take into account information about temporal dynamics. In many real-world OOD-related problems, the full details of the temporal dynamics, such as when OOD spreading begins, are often unknown. Therefore, in the task of spreading OOD detection, which reflects realistic scenarios, information about temporal dynamics is not provided, although different time slices are used for evaluation. Spreading OOD detection assumes that each time slice is independent, meaning that information from other time slices is inaccessible. Consequently, EDBA, tailored for spreading OOD detection, does not consider temporal dynamics.

**Why are human contact studies not considered?** Human contact (Mossong et al., 2008; Balcan et al., 2009; Pastor-Satorras et al., 2015; Mastrandrea et al., 2015) is a highly important topic in the context of the spread of infectious diseases and can play a key role in transmission pathway identification and disease dynamics modeling. However, we emphasize that this paper focuses on introducing realistic benchmarks by simulating spreading, which can model the spread of brand-new products, contaminants, as well as infectious diseases. Although we used simple models, such as SI and SIS models, for OOD spreading to cover these broad scenarios, incorporating human contact modeling would be beneficial for conducting highly realistic simulations, particularly in the context of infectious diseases.

**Why OOD detection is not approached as a binary classification?** OOD detection aims to identify samples that were not seen during the training phase (*i.e.*, OOD samples) when performing classification into ID (in-distribution) classes. Since it is not always feasible to use samples from all possible classes during training, OOD detection addresses a practical scenario. In contrast, binary classification requires samples from both classes to be provided during the training phase. By leveraging the OOD detection approach, we can handle completely new samples (*e.g.*, COVID-19, brand-new products, or computer viruses) without any prior information.

**OOD Detection vs. Outlier Detection** OOD detection and outlier detection may appear similar but are fundamentally different problems, with the main difference lying in their target for detection. OOD detection identifies unseen test samples that differ from the training data distribution. In contrast, outlier detection directly processes all observations and aims to identify outliers within a contaminated dataset (Yang et al., 2021). In other words, the term "in-distribution" (ID) in outlier detection refers to the majority of the observations, whereas for OOD detection, ID refers specifically to the training data.

