# OpenReview forum: "Spreading Out-of-Distribution Detection on Graphs"
_ICLR.cc/2025/Conference — ICLR 2025 Poster_

### Official Review · Reviewer_peqA · 2024-10-31

**Soundness:** 2
**Presentation:** 3
**Contribution:** 2
**Rating:** 5
**Confidence:** 3

**Summary:**

This paper formulates spreading OOD detection, a new benchmark setup for node-level OOD detection method that incorporates interactions among OOD samples and establishes a new dataset called Spreading COVID-19 to benchmark the spreading OOD detection scenario in real-world data.

A node-level OOD detector called EDBD employs an attentive energy aggregation scheme that prevents the mixing of energies between ID nodes and OOD nodes. EDBD is applied to enhance the OOD detection.

**Strengths:**

**S1.**  This paper proposes a new benchmark that models the realistic spread of OOD properties across graph nodes. This setup provides a more applicable framework for real-world scenarios.

**S2.**  This paper leverages EDBD to prevent unwanted mixing of OOD and in-distribution (ID) nodes. By controlling the influence of energy distribution, EDBD achieves better OOD discrimination on proposed datasets and spreading situations.

**S3.**  The paper's aggregation scheme in EDBD is versatile and adaptable to various scenarios on graph-structured data.

**Weaknesses:**

**W1.**  The paper’s reliance on COVID-19 to introduce and validate the spreading OOD detection model may feel outdated. As COVID-19 has become less urgent as a topic, using it as the central example could reduce the model's perceived relevance. A more contemporary application, such as misinformation spread or novel pathogens, might enhance the model's applicability and appeal

**W2.** What’s the relationship of this new problem with other problems, e.g., node classification or outlier detection on graphs? I’m not sure what’s the main difference between this problem and the outlier detection problem. The outliers are not seen in the training set? Can these algorithms be used as the baselines?

**W3.** The effectiveness of the EDBD approach heavily depends on the graph structure and quality of connectivity information. In cases where graph structures are incomplete or noisy, the method’s performance may degrade, potentially affecting its robustness in real-world scenarios.

**W4.** While the paper employs SI and SIS models for simulating OOD spreading, it leaves the integration of more complex epidemic models for future work. This limits the current approach's applicability to more intricate real-world spreading phenomena that may not be accurately represented by simple models.

**W5.** Although the "Spreading COVID-19" dataset simulates virus spread, the lack of training on actual epidemic or contagion data might reduce the generalizability of the model when faced with real-world data that includes nuanced spreading patterns not captured by synthetic simulations.

**Questions:**

**Q1.** How might the proposed spreading OOD detection model adapt to more complex epidemic models, such as SEIR (Susceptible-Exposed-Infectious-Recovered) or multi-host transmission models? Are there specific limitations or computational challenges that prevent their integration?

**Q2.** What steps could be taken to ensure the model’s robustness on graphs with missing or noisy edges and nodes? Would incorporating graph-denoising techniques or imputing missing links improve EDBD’s performance?

**Q3.** Have alternative case studies, such as the spread of misinformation or emergent diseases, been considered to showcase the model’s versatility? How might these cases impact the model's applicability and reception in current contexts?

**Q4.** What’s the relationship of this new problem with other problems, e.g., node classification or outlier detection on graphs?

---

> ### Author Response · Authors · 2024-11-23
> **Response to Reviewer peqA (1/3)**
>
> We sincerely appreciate the reviewer’s thorough evaluation and detailed feedback, which offer valuable insights and help us identify ways to enhance our work. We provide our responses below.
>
> >**W1.**  The paper’s reliance on COVID-19 to introduce and validate the spreading OOD detection model may feel outdated. As COVID-19 has become less urgent as a topic, using it as the central example could reduce the model's perceived relevance. A more contemporary application, such as misinformation spread or novel pathogens, might enhance the model's applicability and appeal.\
> **W5.** Although the "Spreading COVID-19" dataset simulates virus spread, the lack of training on actual epidemic or contagion data might reduce the generalizability of the model when faced with real-world data that includes nuanced spreading patterns not captured by synthetic simulations.\
> **Q3.** Have alternative case studies, such as the spread of misinformation or emergent diseases, been considered to showcase the model’s versatility? How might these cases impact the model's applicability and reception in current contexts?
>
> **A1.** While we understand the reviewer's concern that COVID-19 may feel like a less urgent topic, **we have chosen to use COVID-19 because it has had a tremendous global impact**. Moreover, **testing OOD detection requires the feature distribution of OOD, which poses a challenge for using entirely novel pathogens**.
>
> To address the reviewer’s concern, **we conduct additional experiments to detect an urgent contagion, Respiratory Syncytial Virus (RSV)**, via spreading OOD detection. RSV has recently garnered significant attention due to a surge in infections and its particularly severe impact on infants and the elderly [1,2]. We generate samples with features, created based on information about symptoms of respiratory illness [3,4,5]. We replace COVID-19 samples in the Spreading COVID-19 dataset with RSV samples, and compare the performance of methods in spreading OOD detection. Table 15 in the revised manuscript presents the results for RSV spreading scenarios using SI and SIS models. As shown in the table, EDBD outperforms state-of-the-art methods in both settings across all metrics. **These results on the recent contagion show the generalizability of EDBD**.
>
> Additionally, we have conducted experiments on the COVID-19 dataset using **various epidemic models, including the SI, SIS, and SEIR models, which are widely used in epidemic spreading studies**, and **have confirmed the consistent superiority of our EDBD**. Furthermore, we emphasize that **the Spreading COVID-19 dataset is one of our contributions** aimed at demonstrating the applicability of spreading OOD detection in real-world situations. Our **EDBD outperforms state-of-the-art methods in both conventional node-level OOD detection tasks and spreading OOD detection**. Moreover, the problem of **spreading OOD detection can model various real-world scenarios** on networks influenced by interactions among nodes, including epidemics, and can be applied to diverse graph datasets.
>
> **Table 15**: https://anonymous.4open.science/r/ICLR_2025_6620-B3C2/Table%2015.png
>
> [1] Langedijk, Annefleur C., and Louis J. Bont. "Respiratory syncytial virus infection and novel interventions." Nature Reviews Microbiology 21.11 (2023): 734-749.\
> [2] Moline, Heidi L. "Early Estimate of Nirsevimab Effectiveness for Prevention of Respiratory Syncytial Virus–Associated Hospitalization Among Infants Entering Their First Respiratory Syncytial Virus Season—New Vaccine Surveillance Network, October 2023–February 2024." MMWR. Morbidity and mortality weekly report 73 (2024).\
> [3] https://www.healthline.com/health/flu-covid-rsv-comparison#symptoms \
> [4] https://www.goodrx.com/conditions/covid-19/covid-vs-flu-vs-rsv \
> [5] https://www.chla.org/blog/advice-experts/covid-vs-flu-vs-common-cold-vs-rsv-what-you-need-know

---

> ### Author Response · Authors · 2024-11-23
> **Response to Reviewer peqA (2/3)**
>
> > **W2.** What’s the relationship of this new problem with other problems, e.g., node classification or outlier detection on graphs? I’m not sure what’s the main difference between this problem and the outlier detection problem. The outliers are not seen in the training set? Can these algorithms be used as the baselines?\
> **Q4.** What’s the relationship of this new problem with other problems, e.g., node classification or outlier detection on graphs?
>
> **A2.**
> * OOD Detection vs Outlier Detection
>
> OOD detection and outlier detection may appear similar but are fundamentally different problems, with the main difference lying in their target for detection. OOD detection identifies unseen test samples that differ from the training data distribution. In contrast, outlier detection directly processes all observations and aims to identify outliers within a contaminated dataset [6]. In other words, the term **``in-distribution” (ID) in outlier detection refers to the majority of the observations**, whereas **for OOD detection, ID refers specifically to the training data**.
>
> * OOD Detection vs Node Classification
>
> **OOD detection identifies samples that do not belong to any class seen during training, while classification assigns samples to one of the predefined classes based on the training data**. Since it is not always feasible to use samples from all possible classes during training, OOD detection can address a practical scenario. By leveraging the OOD detection approach, we can handle completely new samples (*e.g.*, COVID-19, brand-new products, or computer viruses) without any prior information.
>
> It is important to note that the problem we propose is not general OOD detection on graphs. **We formulate spreading OOD detection, which accounts for interactions among OOD samples** to cover various real-world scenarios. We have added these important discussion to Appendix F in the revised manuscript.
>
>
> [6] Yang, Jingkang, et al. "Generalized out-of-distribution detection: A survey." International Journal of Computer Vision (2024): 1-28.
>
>
> > **W3.** The effectiveness of the EDBD approach heavily depends on the graph structure and quality of connectivity information. In cases where graph structures are incomplete or noisy, the method’s performance may degrade, potentially affecting its robustness in real-world scenarios.\
> **Q2.** What steps could be taken to ensure the model’s robustness on graphs with missing or noisy edges and nodes? Would incorporating graph-denoising techniques or imputing missing links improve EDBD’s performance?
>
> **A3.** To address the reviewer’s concern, **we conduct additional experiments to evaluate its performance under edge-missing settings by varying the edge missing rate**. As performance degradation is expected when edges are incomplete, **we apply a simple missing link imputation technique** to mitigate the degradation. This link imputation technique **creates a single edge for each node by connecting it to the node with the highest feature similarity**. Table 16 in the revised manuscript presents the results in spreading OOD detection on the Cora dataset. As shown in the table, while the performance of EDBD worsens as the missing rate increases, EDBD with the similarity-based link imputation effectively mitigates performance degradation. We confirm that **EDBD with similarity-based link imputation maintains reasonable performance even with an edge missing rate of 50\%**. We have added this discussion to Appendix C.11 in the revised manuscript.
>
> **Table 16**: https://anonymous.4open.science/r/ICLR_2025_6620-B3C2/Table%2016.png

---

> ### Author Response · Authors · 2024-11-23
> **Response to Reviewer peqA (3/3)**
>
> > **W4.** While the paper employs SI and SIS models for simulating OOD spreading, it leaves the integration of more complex epidemic models for future work. This limits the current approach's applicability to more intricate real-world spreading phenomena that may not be accurately represented by simple models.\
> **Q1.** How might the proposed spreading OOD detection model adapt to more complex epidemic models, such as SEIR (Susceptible-Exposed-Infectious-Recovered) or multi-host transmission models? Are there specific limitations or computational challenges that prevent their integration?
>
> **A4.** To address the reviewer's concern regarding the effectiveness of EDBD in OOD spreading using complex epidemic models, we conduct additional experiments. **We adopt the Susceptible-Exposed-Infectious-Recovered (SEIR) model [7]** as a complex epidemic model. As illustrated in Figure 8 of the revised manuscript, which shows the state transition diagram of SEIR, each node can exist in one of four states: susceptible, exposed, infectious, or recovered. In the SEIR model, a susceptible node can transition only to the exposed state, an exposed node can transition only to the infectious state, and an infectious node can transition only to the recovered state. The probabilities of these transitions are denoted by $\gamma$, $\sigma$, and $\delta$. Based on recent research investigating COVID-19 dynamics [8], we set $\gamma$, $\sigma$, and $\delta$ to 0.5, 0.2, and 0.1, respectively. We follow the experimental setup for spreading OOD detection as specified in Appendix B.2.1 of the manuscript. Table 17 in the revised manuscript demonstrates the results on spreading OOD detection in the Spreading COVID-19 dataset using the SEIR model. As shown in the figure, **EDBD outperforms state-of-the-art methods across all metrics, demonstrating its superiority even in complex spreading scenarios**. We have included these experimental results in Appendix C.12 of the revised manuscript.
>
> **Figure 8**: https://anonymous.4open.science/r/ICLR_2025_6620-B3C2/Figure%208.png \
> **Table 17**: https://anonymous.4open.science/r/ICLR_2025_6620-B3C2/Table%2017.png
>
> [7] Rachah, Amira. "Analysis, simulation and optimal control of a SEIR model for Ebola virus with demographic effects." Communications Faculty of Sciences University of Ankara Series A1 Mathematics and Statistics 67.1 (2018): 179-197.\
> [8] Eikenberry, Steffen E., et al. "To mask or not to mask: Modeling the potential for face mask use by the general public to curtail the COVID-19 pandemic." Infectious disease modelling 5 (2020): 293-308.

---

> > ### Comment · Reviewer_peqA · 2024-11-25
> >
> > Thanks for the response. I have adjusted my ratings.

---

> > > ### Author Response · Authors · 2024-12-02
> > > **[Gentle Reminder] Kindly Seeking Your Response Before the Deadline**
> > >
> > > Dear **Reviewer peqA**,
> > >
> > > With only **22 hours remaining before the deadline**, we once again respectfully request clarification on the rationale behind this decision and inquire if there are any remaining concerns. In our rebuttal, we provided detailed, point-by-point responses to all the concerns and questions you raised. We strongly believe that we have thoroughly addressed all of the reviewer’s concerns and questions. We sincerely thank you for your professional and thoughtful feedback and are more than willing to discuss any additional improvements that could help you evaluate our work more positively.
> > >
> > > Sincerely,\
> > > The Authors

---

> > > > ### Author Response · Authors · 2024-12-03
> > > > **[Final Reminder] Respectful Request for Clarification on Reviewer peqA's Decision**
> > > >
> > > > Dear **Reviewer peqA**,
> > > >
> > > > With **only one hour remaining before the deadline**, we kindly remind you of our previous inquiry. We would greatly appreciate it if you could take a moment to respond.
> > > >
> > > > Thank you once again for your time and consideration.
> > > >
> > > > Sincerely,\
> > > > The Authors

---

> ### Author Response · Authors · 2024-11-26
> **Respectful Request for Clarification on Reviewer peqA's Decision**
>
> Thank you for taking the time to review our rebuttal and for raising your rating to "marginally below the acceptance threshold." However, we have responded to the five aspects that Reviewer peqA pointed out as weaknesses:
>
> * For Weakness 1: Reviewer peqA highlighted the need for a more contemporary application than COVID-19. In response, we conducted **additional experiments on an urgent contagion, Respiratory Syncytial Virus (RSV)**, and validated the effectiveness of our method, EDBD, in this scenario.
>
> * For Weakness 2: Reviewer peqA requested clarification on the relationship between the new problem of spreading OOD detection and outlier detection or node classification. In response, we provided a clear explanation of the **fundamental differences between spreading OOD detection and outlier detection, as well as between spreading OOD detection and node classification**.
>
> * For Weakness 3: Reviewer peqA requested validation of our model on graphs with incomplete graph structures. In response, we conducted **additional experiments under edge-missing settings** and confirmed that **EDBD, with similarity-based link imputation, maintains reasonable performance** even with an edge-missing rate of 50%.
>
> * For Weakness 4: Reviewer peqA requested experiments using a more complex epidemic model. In response, we conducted **additional experiments using the SEIR model suggested by the reviewer** and verified the superiority of our EDBD in these settings.
>
> * For Weakness 5: Reviewer peqA raised concerns about the lack of training on actual epidemic or contagion data in the context of the Spreading COVID-19 dataset. In response, we clarified that **(1) the Spreading COVID-19 dataset is one of the contributions of this paper,** along with **(2) the formulation of the Spreading OOD detection problem**, which can model various real-world scenarios, and **(3) our EDBD, which achieves state-of-the-art performance** in both conventional node-level OOD detection tasks and spreading OOD detection. We further clarify that **we conducted experiments on the COVID-19 dataset using various epidemic models, including the SI, SIS, and SEIR models, which are widely used in epidemic spreading studies**, and confirmed the **consistent superiority of our EDBD**.
>
> **We strongly believe that we have addressed all of the reviewer’s concerns and questions**, which have enhanced the quality of our paper. We have uploaded a revised manuscript that incorporates all of these updates. Nevertheless, Reviewer 4o3t has maintained a position inclined towards rejection. **We respectfully request clarification on the rationale behind this decision and inquire whether there are any remaining concerns**. If there are any such concerns, we are fully prepared to address them immediately.

---

> ### Author Response · Authors · 2024-12-04
> **Follow-Up on Our Clarifications and Request to Reviewer peqA**
>
> Dear Reviewer peqA,
>
> We sincerely appreciate the time and effort Reviewer peqA has dedicated to reviewing and evaluating our work.
>
> **While we have thoroughly addressed all of Reviewer peqA's concerns, we have not received any response to our clarifications or request over the past eight days, despite repeated attempts to seek confirmation**. We hope that Reviewer peqA will continue to review our clarifications and request even after the discussion period has concluded.
>
> Sincerely,\
> The Authors

---

### Official Review · Reviewer_t4Yw · 2024-11-02

**Soundness:** 3
**Presentation:** 3
**Contribution:** 2
**Rating:** 6
**Confidence:** 3

**Summary:**

This paper introduces a novel task called spreading out-of-distribution (OOD) detection, which focuses on identifying OOD nodes within graph structures, particularly emphasizing the interactions and influence these nodes have on their neighbors. The authors leverage epidemic spreading models to create realistic benchmarks that reflect the dynamics of OOD node propagation in real-world scenarios. The authors propose an innovative approach known as the Energy Distribution-Based Detector (EDBD), which utilizes energy scores to enhance the aggregation process of node information. This method aims to mitigate the mixing of OOD scores between in-distribution (ID) and OOD nodes, thereby improving the accuracy of OOD detection.

**Strengths:**

1.The creation of the Spreading COVID-19 dataset provides a realistic and relevant context for evaluating OOD detection methods. This dataset simulates real-world scenarios, making the findings more applicable to practical situations.
2. The authors conducted comprehensive experiments, comparing EDBD against several state-of-the-art methods across multiple datasets.
3. The paper is well-structured and clearly presents the methodology, experimental setup, and results, making it accessible for readers and researchers interested in the topic.

**Weaknesses:**

1. While the results demonstrate superior performance, the paper may lack comprehensive statistical analysis (e.g., significance testing) to support the claims of superiority over existing methods.
2.The energy-based aggregation approach may introduce additional complexity compared to simpler methods. This complexity could make the method less accessible for practitioners who may prefer more straightforward solutions.

**Questions:**

1. What is the sensitivity of EDBD to different hyperparameter settings? How do variations in these parameters affect the model's performance?
2. How does EDBD perform on larger graphs or more complex networks? Are there any scalability issues that arise when applying the method to real-world scenarios with extensive node interactions?
3. How interpretable are the results produced by EDBD? Can the authors provide insights into why certain nodes are classified as OOD based on the energy aggregation process?

---

> ### Author Response · Authors · 2024-11-23
> **Response to Reviewer t4Yw (1/2)**
>
> We sincerely thank the reviewer for their thoughtful evaluation, constructive feedback, and insightful questions, which have significantly improved our work. We provide our responses below.
>
> >**W1.** While the results demonstrate superior performance, the paper may lack comprehensive statistical analysis (e.g., significance testing) to support the claims of superiority over existing methods.
>
> **A1**. **We have conducted a statistical analysis to evaluate the statistical significance of our EDBD’s superior performance in Appendix C.6 of the manuscript**. Table 12 in the revised manuscript shows the $p$-values comparing EDBD to the runner-up in each setting for all the results in Tables 1, 2, and 3 of the manuscript. As shown in Table 12, the $p$-values for Table 2 and Table 3 are higher compared to those of Table 1. This is because spreading OOD detection tasks involve spreading simulations and feature sampling for each episode, leading to varying OOD spreading patterns and differing training and testing data. These variations significantly affect the difficulty level of each episode, resulting in substantial performance fluctuations and large standard deviations across methods. When the standard deviation is high, statistical tests are less likely to show significant differences between methods, yielding higher $p$-values, even when the mean performance differs between EDBD and the baselines.
>
> Nevertheless, **while the runner-up methods vary depending on the setting, EDBD consistently outperforms them across all settings and metrics, except for AUPR-T in spreading OOD detection on the LastFM Asia dataset using the SIS model**. Furthermore, as shown in Appendix C.9, EDBD consistently demonstrates its superiority when using SIS models with different combinations of parameters. **This consistency underscores EDBD’s robustness and generalizability** across various graph OOD-related tasks. If any additional statistical analyses are required, we are prepared to conduct them promptly.
>
> >**W2.** The energy-based aggregation approach may introduce additional complexity compared to simpler methods. This complexity could make the method less accessible for practitioners who may prefer more straightforward solutions.\
> >**Q2.** How does EDBD perform on larger graphs or more complex networks? Are there any scalability issues that arise when applying the method to real-world scenarios with extensive node interactions?
>
> **A2.** **We have discussed the complexity of EDBD in terms of both time and memory in Appendix C.1 in the manuscript**. In Appendix C.1, we compare the training and inference times of all methods, and also **propose FastEDBD, a light version of EDBD**. Furthermore, **we confirm that EDBD operates well on the OGBN-Arxiv dataset, which contains 169,343 nodes**, representing a large-scale graph. Nevertheless, if EDBD encounters scalability issues, we recommend using FastEDBD, which also delivers significant performance gains compared to existing state-of-the-art methods.
>
> In this discussion period, **we have compared the Big-O memory complexity with other methods to consider scalability**. Table 8 in the revised manuscript shows the comparison of Big-O memory complexity. Although EDBD utilize additional matrices $\mathbf{S} \in \mathbb{R}^{N \times N}$ and $\mathbf{C} \in \mathbb{R}^{N \times N}$, $\mathbf{S}$ is a weighted adjacency matrix with non-zero values corresponding to the number of edges, and $\mathbf{C}$ is a diagonal matrix. Thus, **the memory complexity of EDBD is $O(\theta) + O(B(F+C)) + O(N) +O(\mathcal{E})$, which is the same as that of GNNSAFE, the most competitive baseline**. We have added this discussion to Appendix C.1. in the revised manuscript.

---

> ### Author Response · Authors · 2024-11-23
> **Response to Reviewer t4Yw (2/2)**
>
> >**Q1.** What is the sensitivity of EDBD to different hyperparameter settings? How do variations in these parameters affect the model's performance?
>
> **A3.** We conduct additional experiments to address the reviewer’s concern and **provide a comprehensive analysis of the impact of different hyperparameters, including $\alpha$, $\beta$, $\epsilon$, and $K$**, on the performance of EDBD. Under two settings using SI and SIS models, respectively, we report FPR95-T in spreading OOD detection on the Cora dataset. $\alpha$, $\beta$, $\epsilon$, and $K$ are varied within the ranges of \{0.1,0.2,0.3,0.5\}, \{1/4,1/3,1/2,1\}, \{0.01, 0.05, 0.1, 0.5, 0.75\}, and \{1, 2\}, respectively, which are EDBD’s search ranges.
>
> Figure 7 in the revised manuscript shows the results. Compared to existing state-of-the-art performance, EDBD models consistently exceed it by a considerable margin regardless of the values of $\alpha$, $\beta$, and $K$. While EDBD with $\epsilon =0.75$ commonly shows worse performance compared to the existing state-of-the-art performance, the small value of $\epsilon$ indicates that energy aggregation focuses less on energy similarity and more on the graph structure. This result underscores the importance of considering energy distribution in the energy aggregation process. As stated in Appendix B.2.3 of the manuscript, we determine the hyperparameters of EDBD for each setting using grid search within fixed ranges, based on performance on the validation set. We have added this analysis of hyperparameter sensitivity to Appendix C.7 of the revised manuscript.
>
> **Figure 7:** https://anonymous.4open.science/r/ICLR_2025_6620-B3C2/Figure%207.png
>
> >**Q3.** How interpretable are the results produced by EDBD? Can the authors provide insights into why certain nodes are classified as OOD based on the energy aggregation process?
>
> The energy aggregation process aims at denoising energies (OOD scores) since OOD scores may not be perfect. **The theoretical analysis in Appendix E of the revised manuscript offers insights into how the energy aggregation process enables the correct distinction** between ID and OOD nodes. This analysis shows two points: **(1)** an ID node aggregating energies from surrounding ID nodes and an OOD node aggregating energies from surrounding OOD nodes are beneficial for updating energies and **(2)** when both ID nodes and OOD nodes coexist in neighbors, simple energy aggregation that solely relies on the graph structure (*i.e.,* GNNSAFE) leads to undesired mixing of energies. **Combining (1) and (2), when both ID nodes and OOD nodes coexist as neighbors, enabling an ID node to focus on its neighboring ID nodes and an OOD node to focus on its neighboring OOD nodes allows for energy updates that improve OOD detection**. To this end, **EDBD utilizes the initial energies as a temporary OOD indicator, and regulates aggregation on each node to alleviate the mixing of energies between OOD nodes and ID nodes**.

---

> > ### Author Response · Authors · 2024-12-02
> > **[Gentle Reminder] Kindly Seeking Your Feedback Before the Deadline**
> >
> > Dear **Reviewer t4Yw**,
> >
> > In our rebuttal, we provided detailed, point-by-point responses to all the concerns and questions you raised. With only **22 hours remaining before the deadline**, we are eager to confirm whether our responses have sufficiently addressed your concerns. We kindly ask you to take a moment to review our rebuttal and share any additional feedback. If there are any remaining questions or concerns, please rest assured that we are prepared to respond promptly.
> >
> > Sincerely,\
> > The Authors

---

> > ### Author Response · Authors · 2024-12-03
> > **[Final Reminder] Eager for Your Feedback on our Rebuttal**
> >
> > Dear **Reviewer t4Yw**,
> >
> > With **only one hour remaining before the deadline**, we are eager to confirm whether our responses have adequately addressed your concerns. We kindly request you to take a moment to review our rebuttal and share any feedback.
> >
> > Sincerely,\
> > The Authors

---

> ### Author Response · Authors · 2024-12-01
> **Eager for Your Feedback on Our Rebuttal**
>
> Dear Reviewer t4Yw,
>
> We sincerely thank you for dedicating your time to review our work and for providing professional feedback. Your insights have significantly contributed to improving the quality of our paper. With two days remaining for the discussion period, we are eager to engage further and understand if our responses have satisfactorily addressed your concerns.
>
> In our rebuttal, **we provided point-by-point responses to all your questions and concerns regarding (1) statistical analysis, (2) complexity, (3) hyperparameter sensitivity , and (4) the reasoning behind EDBD**. In summary:
> * For (1), the original manuscript **already included** a statistical analysis evaluating the significance of EDBD’s superior performance in Appendix C.6.
> * For (2), the original manuscript **already included** an in-depth discussion of EDBD’s complexity in terms of both time and memory in Appendix C.1. In this discussion, **we confirmed EDBD operates well on a large-scale graph** and also **proposed FastEDBD, a lightweight version of EDBD**. In response to your question, **we further compared the Big-O memory complexity with other methods**.
> * For (3), we conducted **a comprehensive analysis of hyperparameter sensitivity**.
> * For (4), we provided **a theoretical analysis offering insights into how EDBD works**.
>
>
> We would greatly appreciate it if you could kindly review our responses. We welcome any additional questions and are happy to provide further clarifications if needed. Thank you again for your valuable feedback and consideration.
>
> Sincerely,\
> The Authors

---

### Official Review · Reviewer_8nsJ · 2024-11-02

**Soundness:** 3
**Presentation:** 3
**Contribution:** 3
**Rating:** 8
**Confidence:** 3

**Summary:**

This paper introduces "spreading OOD detection," for node-level out-of-distribution (OOD) detection on graphs. It addresses limitations in existing methods by modeling the spread of OOD samples through graph structures. The authors propose a new dataset, "Spreading COVID-19," and present a novel approach called Energy Distribution-Based Detector (EDBD) that outperforms existing methods in both spreading and conventional OOD detection tasks.

**Strengths:**

1. The concept of spreading OOD detection is novel and addresses a gap in existing methods.
2. A new dataset is introduced which will help future research
3. The figures used in the paper manage to illustrate the problem and approach well.

**Weaknesses:**

1. It would be interesting to see comparison of EDBD with SOTA in terms of computational complexity and scalability.
2. Some discussion on how the proposed dataset compares to real-world datasets and its characteristics would benefit the paper

**Questions:**

1. Are there any specific graph structures or spreading patterns where EDBD might underperform compared to other methods?
2. How sensitive is the performance of EDBD to the choice of hyperparameters in the energy-aggregation scheme?

---

> ### Author Response · Authors · 2024-11-23
> **Response to Reviewer 8nsJ (1/2)**
>
> We sincerely thank the reviewer for their insightful feedback and recognition of our work's contributions. We provide our responses below.
>
> >**W1.** It would be interesting to see comparison of EDBD with SOTA in terms of computational complexity and scalability.
>
> **A1.** **We have discussed the complexity of EDBD in terms of both time and memory in Appendix C.1 in the manuscript**. In Appendix C.1, we compare the training and inference times of all methods, and also propose FastEDBD, a light version of EDBD. Furthermore, we confirm that EDBD operates well on the OGBN-Arxiv dataset, which contains 169,343 nodes, representing a large-scale graph. Nevertheless, if EDBD encounters scalability issues, we recommend using FastEDBD, which also delivers significant performance gains compared to existing state-of-the-art methods.
>
> As suggested by the reviewer, **we compare the Big-O memory complexity with other methods to consider scalability**. Table 8 in the revised manuscript shows the comparison of Big-O memory complexity. Although EDBD utilize additional matrices $\mathbf{S} \in \mathbb{R}^{N \times N}$ and $\mathbf{C} \in \mathbb{R}^{N \times N}$, $\mathbf{S}$ is a weighted adjacency matrix with non-zero values corresponding to the number of edges, and $\mathbf{C}$ is a diagonal matrix. Thus, **the memory complexity of EDBD is $O(\theta) + O(B(F+C)) + O(N) +O(\mathcal{E})$, which is the same as that of GNNSAFE, the most competitive baseline**. We have added this discussion to Appendix C.1. in the revised manuscript.
>
> >**W2.** Some discussion on how the proposed dataset compares to real-world datasets and its characteristics would benefit the paper.
>
> **A2.** In lines 890–897, we have discussed the differences between our Spreading COVID-19 dataset and existing real-world datasets. **While general infectious disease datasets [1,2,3,4] focus solely on the spreading patterns of infectious diseases, ours includes person-level symptom-based features**. Furthermore, **we provide an ID class for each sample**: normal (no illness), allergies, cold, and flu. This setup enables the formulation of COVID-19 detection as OOD detection, where classifiers trained only on ID data attempt to detect the new disease, COVID-19. To the best of our knowledge, the Spreading COVID-19 dataset is the first COVID-19-related dataset to include both sample-level features and class labels. Based on this discussion with the reviewer, we have supplemented the original explanation regarding the differences compared to existing datasets in the revised manuscript.
>
> [1] Mastrandrea, Rossana, Julie Fournet, and Alain Barrat. "Contact patterns in a high school: a comparison between data collected using wearable sensors, contact diaries and friendship surveys." PloS one 10.9 (2015): e0136497.
> [2] Singh, David E., et al. "Simulation of COVID-19 propagation scenarios in the Madrid metropolitan area." Frontiers in Public Health 9 (2021): 636023.
> [3] Alrasheed, Hend, et al. "COVID-19 spread in Saudi Arabia: modeling, simulation and analysis." International Journal of Environmental Research and Public Health 17.21 (2020): 7744.
> [4] Rafiq, Muhammad, et al. "Numerical simulations on scale-free and random networks for the spread of COVID-19 in Pakistan." Alexandria Engineering Journal 62 (2023): 75-83.

---

> ### Author Response · Authors · 2024-11-23
> **Response to Reviewer 8nsJ (2/2)**
>
> >**Q1.** Are there any specific graph structures or spreading patterns where EDBD might underperform compared to other methods?
>
> **A3.** We have confirmed that EDBD demonstrates superiority across various datasets and epidemic scenarios, outperforming conventional OOD detection methods. However, consider **a graph composed solely of pairs connecting OOD and ID nodes, where ID nodes are exclusively connected to OOD nodes, and vice versa**. In this case, energy-based OOD detection methods that operate at the sample level without considering connectivity may outperform energy-aggregation-based approaches, such as EDBD and GNNSAFE. **Nevertheless, such specific scenarios are extremely rare in real-world applications, thereby leading to EDBD's superior performance across various datasets in both spreading OOD detection and conventional node-level OOD detection**.
>
> >**Q2.** How sensitive is the performance of EDBD to the choice of hyperparameters in the energy-aggregation scheme?
>
> **A4.** We conduct additional experiments to address the reviewer’s concern and **provide a comprehensive analysis of the impact of different hyperparameters, including $\alpha$, $\beta$, $\epsilon$, and $K$**, on the performance of EDBD. Under two settings using SI and SIS models, respectively, we report FPR95-T in spreading OOD detection on the Cora dataset. $\alpha$, $\beta$, $\epsilon$, and $K$ are varied within the ranges of \{0.1,0.2,0.3,0.5\}, \{1/4,1/3,1/2,1\}, \{0.01, 0.05, 0.1, 0.5, 0.75\}, and \{1, 2\}, respectively, which are EDBD’s search ranges.
>
> Figure 7 in the revised manuscript shows the results. Compared to existing state-of-the-art performance, EDBD models consistently exceed it by a considerable margin regardless of the values of $\alpha$, $\beta$, and $K$. While EDBD with $\epsilon =0.75$ commonly shows worse performance compared to the existing state-of-the-art performance, the small value of $\epsilon$ indicates that energy aggregation focuses less on energy similarity and more on the graph structure. This result underscores the importance of considering energy distribution in the energy aggregation process. As stated in Appendix B.2.3 of the manuscript, we determine the hyperparameters of EDBD for each setting using grid search within fixed ranges, based on performance on the validation set. We have added this analysis of hyperparameter sensitivity to Appendix C.7 of the revised manuscript.
>
> **Figure 7:** https://anonymous.4open.science/r/ICLR_2025_6620-B3C2/Figure%207.png

---

> ### Comment · Reviewer_8nsJ · 2024-11-24
>
> I would like to thank the authors for their detailed and thorough response

---

> > ### Author Response · Authors · 2024-11-25
> >
> > We appreciate your consistent positive evaluation of our paper as a good paper. Your insightful feedback has further enhanced its quality.

---

### Official Review · Reviewer_xogV · 2024-11-03

**Soundness:** 2
**Presentation:** 3
**Contribution:** 3
**Rating:** 6
**Confidence:** 4

**Summary:**

This paper introduces a more realistic benchmark for node-level out-of-distribution (OOD) detection on graphs, called “spreading OOD detection”, where  OOD characteristics spread across connected nodes, similar to real-world diffusion processes like disease spreading. The paper dedicates one part to defining this benchmark in detail, using the “Spreading COVID-19” dataset to show its practical value. This benchmark is different from traditional approaches that treat OOD nodes as isolated and do not consider interactions. Another part of the paper is focused on a new method, the Energy Distribution-Based Detector (EDBD), designed to address this benchmark. EDBD uses an energy-aggregation scheme to keep OOD and in-distribution (ID) scores from mixing, helping to identify OOD nodes more accurately.

**Strengths:**

Although the benchmark is presented as realistic, it relies on assumptions in simulations that may be hard to verify, particularly given the use of the LastFM dataset, which may not accurately represent human contact patterns essential for modeling disease spread. The suitability of LastFM’s social connections as a proxy for physical contact is uncertain, as are other factors like spatial constraints and the presence of high-degree nodes that could act as unrealistic “super-spreaders.” To address this concern and enhance the benchmark’s realism, the authors could validate these assumptions against datasets explicitly designed for human contact networks (such as those used in epidemiology studies) or test the model on multiple datasets with diverse structural properties to better reflect real-world contact interactions.

The EDBD method’s performance appears closely tied to assumptions within the benchmark setup, making it difficult to assess its independent technical relevance. The method seems to build on GNNSAFE with the addition of energy similarity and consistency matrices, yet these connections and differences are not fully discussed. To clarify the contribution, the authors could provide a more detailed comparison between EDBD and GNNSAFE, explaining how the added matrices influence performance and to what extent the results depend on benchmark-specific assumptions.

**Weaknesses:**

Although the benchmark is presented as realistic, it relies on assumptions in simulations that may be hard to verify, particularly given the use of the LastFM dataset, which may not accurately represent human contact patterns essential for modeling disease spread. The literature in this area includes foundational works from over a decade ago, which should be discussed, e.g. [1,2,3].
The suitability of LastFM’s social connections as a proxy for physical contact is uncertain, as are other factors like spatial constraints and the presence of high-degree nodes that could act as unrealistic “super-spreaders.” To address this concern and enhance the benchmark’s realism, the authors could validate these assumptions against datasets explicitly designed for human contact networks (such as those used in epidemiology studies) or test the model on multiple datasets with diverse structural properties to better reflect real-world interactions.

The EDBD method’s performance appears closely tied to assumptions within the benchmark setup, making it difficult to assess its technical relevance. The method seems to build on GNNSAFE with the addition of energy similarity and consistency matrices, yet these connections and differences are not fully discussed. To clarify the contribution, the authors could provide a more detailed comparison between EDBD and GNNSAFE, explaining how the added matrices influence performance and to what extent the results depend on benchmark-specific assumptions.

[1] Mossong, J., Hens, N., Jit, M., Beutels, P., Auranen, K., Mikolajczyk, R., ... & Edmunds, W. J. (2008). Social contacts and mixing patterns relevant to the spread of infectious diseases. PLoS medicine, 5(3), e74.
[2] Balcan, D., Colizza, V., Gonçalves, B., Hu, H., Ramasco, J. J., & Vespignani, A. (2009). Multiscale mobility networks and the spatial spreading of infectious diseases. Proceedings of the national academy of sciences, 106(51), 21484-21489.
[3] Pastor-Satorras, R., Castellano, C., Van Mieghem, P., & Vespignani, A. (2015). Epidemic processes in complex networks. Reviews of modern physics, 87(3), 925-979.

**Questions:**

Q1: Since the dataset is central to the benchmark, it is crucial to understand how well it reflects realistic contact-driven networks for virus spreading. Information on whether the dataset’s assumptions have been validated against more realistic epidemic scenarios would also help clarify its suitability for simulating disease dynamics. Could the authors provide detailed information on the construction and structure of the LastFM dataset, including the main mechanisms of edge formation, the presence of high-degree nodes that could act as super-spreaders, and the dataset’s time span in relation to episode durations required to reach 75% node infection?

Q2: Could the authors discuss the sensitivity of the proposed method to variations in the SIS model parameters and its adaptability to different diffusion patterns? Understanding this sensitivity is important for evaluating the method’s robustness in various scenarios. Additionally, could the authors clarify why OOD detection is not approached as a binary classification between ID and OOD, or as a one-class classification in cases where only ID labels are available? Addressing these points would provide valuable insights into the generalizability of the benchmark and the chosen approach for OOD detection.

Q3: Given the high error bars in Tables 2 and 3, how conclusive are the results? Could the authors explain if the similar performance of EDBD and the Energy method suggests that data features, rather than graph structure, drive the performance? Also, since Eq. 4 of the paper is quite similar to Eq. 7 of GNNSAFE, could the authors clarify the exact differences between EDBD and GNNSAFE beyond adding the energy similarity and consistency matrices? For example, does the first row in Table 4 represent GNNSAFE?

Q4: Since hyperparameter tuning can greatly impact performance, could the authors explain how it was done for both EDBD and baseline methods? Was the search for competitors restricted compared to EDBD? For GNNSAFE, was only the supervised loss used, or did the authors include GNNSAFE++ with energy regularization? If only the supervised loss was used, what motivated this choice?

---

> ### Author Response · Authors · 2024-11-23
> **Response to Reviewer xogV (1/5)**
>
> We sincerely thank the reviewer for the detailed feedback, thoughtful questions, and valuable suggestions to improve our work. We provide our responses below.
>
> > **W1.** The literature in this area (human contact patterns) includes foundational works from over a decade ago, which should be discussed, e.g. [1,2,3].
>
> **A1.** While we have discussed relevant studies, including epidemic models, we have overlooked the discussion of human contact. Human contact is a highly important topic in the context of the spread of infectious diseases and plays a key role in transmission pathway identification and disease dynamics modeling. However, **we emphasize that this paper focuses on introducing realistic benchmark setup by simulating the spread of OOD, which can model the spread of brand-new products, contaminants, as well as infectious diseases**. Although we use simple models, such as the SI and SIS models, for OOD spreading to cover these broad scenarios, incorporating human contact modeling would be beneficial for conducting highly realistic simulations, particularly in the context of infectious diseases.
>
> **We conduct additional experiments simulating the spread of COVID-19 on a real-world contact network used in human contact studies**. We utilize a contact network dataset collected in a high school in France [4]. We measure the performance of methods on spreading OOD detection by replacing only the graph structure in the Spreading COVID-19 dataset with this network. Table 13 in the revised manuscript presents the results when the SI and SIS models are used, respectively. As shown in the table, **our EDBD consistently outperforms state-of-the-art methods across all metrics and epidemic models**.
>
> **We have added this discussion, along with the references [1, 2, 3] suggested by the reviewer and [4], to Appendix F of the revised manuscript**. We have included the experimental results on the contact network dataset in Appendix C.8 in the revised manuscript.
>
> **Table 13**: https://anonymous.4open.science/r/ICLR_2025_6620-B3C2/Table%2013.png
>
> [1] Mossong, Joël, et al. "Social contacts and mixing patterns relevant to the spread of infectious diseases." PLoS medicine 5.3 (2008): e74.\
> [2] Balcan, D., Colizza, V., Gonçalves, B., Hu, H., Ramasco, J. J., & Vespignani, A. (2009). Multiscale mobility networks and the spatial spreading of infectious diseases. Proceedings of the national academy of sciences, 106(51), 21484-21489.\
> [3] Pastor-Satorras, R., Castellano, C., Van Mieghem, P., & Vespignani, A. (2015). Epidemic processes in complex networks. Reviews of modern physics, 87(3), 925-979.\
> [4] Mastrandrea, Rossana, Julie Fournet, and Alain Barrat. "Contact patterns in a high school: a comparison between data collected using wearable sensors, contact diaries and friendship surveys." PloS one 10.9 (2015): e0136497.

---

> ### Author Response · Authors · 2024-11-23
> **Response to Reviewer xogV (2/5)**
>
> > **W2.** Although the benchmark is presented as realistic, it relies on assumptions in simulations that may be hard to verify, particularly given the use of the LastFM dataset, which may not accurately represent human contact patterns essential for modeling disease spread. The suitability of LastFM’s social connections as a proxy for physical contact is uncertain, as are other factors like spatial constraints and the presence of high-degree nodes that could act as unrealistic “super-spreaders.” To address this concern and enhance the benchmark’s realism, the authors could validate these assumptions against datasets explicitly designed for human contact networks (such as those used in epidemiology studies) or test the model on multiple datasets with diverse structural properties to better reflect real-world interactions.\
> **Q1.** Since the dataset is central to the benchmark, it is crucial to understand how well it reflects realistic contact-driven networks for virus spreading. Information on whether the dataset’s assumptions have been validated against more realistic epidemic scenarios would also help clarify its suitability for simulating disease dynamics. Could the authors provide detailed information on the construction and structure of the LastFM dataset, including the main mechanisms of edge formation, the presence of high-degree nodes that could act as super-spreaders, and the dataset’s time span in relation to episode durations required to reach 75% node infection?
>
> **A2.** The edges of the LastFM Asia dataset are constructed based on mutual follower relationships among users. As stated in ``Justification on Using LastFM Asia for the Graph Structure” in Appendix A of the manuscript, due to the lack of large human contact networks, **we utilize the structure of this dataset** to simulate the spread of COVID-19 in the proposed Spreading COVID-19 dataset, **based on the rationale that online networks share structural characteristics similar to offline networks [5]**. To further validate the use of the graph structure of the LastFM Asia dataset, **we thoroughly compare it with the High School Contact network** [4].
>
> Figure 5 in the revised manuscript compares the node degree distributions of the two real-world networks. Human contact networks often feature super-spreaders with high node degrees. As expected, the High School Contact network contains super-spreaders with node degrees of 50 or more, as shown in the figure. **Similarly, the LastFM Asia network also includes high-degree nodes, with some exceeding degrees of 140, which act as super-spreaders**. Although the network sizes of the two datasets differ (327 nodes for the High School Contact network and 7624 nodes for the LastFM Asia network), both networks exhibit similar characteristics in terms of the presence of super-spreaders.
>
> Furthermore, **we compare the average episode duration**, defined as the number of steps required for 75\% of nodes to become infected. For the SI, SIS, and SEIR (Susceptible-Exposed-Infectious-Recovered) models, the High School Contact network requires 8.3, 10.1, and 18.4 steps, respectively, while the LastFM Asia network requires 10.0, 11.1, and 24.6 steps, respectively. **Considering the difference in network sizes, we can confirm that the human contact network do not lead to faster spreading compared to the LastFM Asia network**.
>
> Although we demonstrate that the LastFM Asia dataset and the human contact network share similar properties in terms of graph structure, **we further simulate COVID-19 spreading on the High School Contact network and validate the effectiveness of EDBD in those scenarios** (in Appendix C.8 in the revised manuscript).
>
> **Figure 5**: https://anonymous.4open.science/r/ICLR_2025_6620-B3C2/Figure%205.png
>
>
> [5] Dunbar, Robin IM, et al. "The structure of online social networks mirrors those in the offline world." Social networks 43 (2015): 39-47.

---

> ### Author Response · Authors · 2024-11-23
> **Response to Reviewer xogV (3/5)**
>
> > **W3.** The EDBD method’s performance appears closely tied to assumptions within the benchmark setup, making it difficult to assess its independent technical relevance. The method seems to build on GNNSAFE with the addition of energy similarity and consistency matrices, yet these connections and differences are not fully discussed. To clarify the contribution, the authors could provide a more detailed comparison between EDBD and GNNSAFE, explaining how the added matrices influence performance and to what extent the results depend on benchmark-specific assumptions.\
> **Q3-1.** Also, since Eq. 4 of the paper is quite similar to Eq. 7 of GNNSAFE, could the authors clarify the exact differences between EDBD and GNNSAFE beyond adding the energy similarity and consistency matrices? For example, does the first row in Table 4 represent GNNSAFE?
>
> **A3.** EDBD aims to effectively update OOD scores through energy aggregation on a network where ID and OOD nodes coexist. **We clarify that EDBD is not exclusively designed for the proposed spreading OOD detection**. Instead of relying on benchmark-specific assumptions, **EDBD shows its effectiveness in both conventional node-level OOD detection and spreading OOD detection** (as demonstrated in Table 1,2, and 3 of the manuscript).
>
> As the reviewer pointed out, the energy update equations of GNNSAFE [6] and EDBD may appear similar since both update energies (OOD scores) through neighborhood aggregation in a graph. **However, as discussed in lines 281–285, GNNSAFE relies solely on the graph structure during aggregation, which results in undesired energy mixing between ID and OOD nodes, thereby hindering OOD detection**. In this discussion period, **we have theoretically demonstrated that this undesired energy mixing is caused by GNNSAFE in Appendix E of the revised manuscript**. To address this limitation, we introduce the similarity matrix $\mathbf{S}$ and the consistency matrix $\mathbf{C}$ into the energy aggregation process. These matrices, $\mathbf{S}$ and $\mathbf{C}$, are calculated using initial energies, effectively preventing undesired energy mixing.
>
> As the reviewer mentioned, the first rows in Table 4 correspond to the performance of GNNSAFE. **In Sec. 5.4 of the manuscript, including Table 4, we demonstrate how $\mathbf{S}$ and $\mathbf{C}$ individually influence performance**. To provide a clearer illustration of the effects of introducing each matrix to GNNSAFE, we clarify that the first rows in Table 4 represent the performance of GNNSAFE in the revised manuscript.
>
> [6] Wu, Qitian, et al. "Energy-based Out-of-Distribution Detection for Graph Neural Networks." The Eleventh International Conference on Learning Representations.
>
> > **Q2-1.** Could the authors discuss the sensitivity of the proposed method to variations in the SIS model parameters and its adaptability to different diffusion patterns? Understanding this sensitivity is important for evaluating the method’s robustness in various scenarios.
>
> **A4.** As stated in lines 881–889 of the revised manuscript, we determine the parameters of the epidemic models by considering COVID-19 dynamics. Following the reviewer’s suggestion, **we adjust the parameters of the SIS model and compare the performance of the methods** across various SIS models by performing spreading OOD detection on the Spreading COVID-19 dataset. Specifically, we conduct experiments by modifying the original $(\gamma, \delta)$ values of $(0.5,0.1)$ to $(0.7,0.1)$ and $(0.3,0.1)$.  Additionally, we perform experiments by changing the original $(\gamma, \delta)$ values to $(0.5,0.2)$ and $(0.5,0.05)$. Table 14 in the revised manuscript presents the results. As shown in the table, except for AUPR-T under $(\gamma, \delta)=(0.7,0.1)$, **EDBD consistently outperforms state-of-the-art methods and demonstrates robustness against various spreading patterns**. We have included this discussion in the Appendix C.9 of the revised manuscript.
>
> **Table 14**: https://anonymous.4open.science/r/ICLR_2025_6620-B3C2/Table%2014.png

---

> ### Author Response · Authors · 2024-11-23
> **Response to Reviewer xogV (4/5)**
>
> > **Q2-2.** Additionally, could the authors clarify why OOD detection is not approached as a binary classification between ID and OOD, or as a one-class classification in cases where only ID labels are available? Addressing these points would provide valuable insights into the generalizability of the benchmark and the chosen approach for OOD detection.
>
> **A5.** OOD detection aims to identify samples that were not seen during the training phase (*i.e.*, OOD samples) when performing classification into ID (in-distribution) classes. Since it is not always feasible to use samples from all possible classes during training, OOD detection can address a practical scenario. In contrast, **binary classification requires samples from both classes during the training phase**. **By leveraging the OOD detection approach, we can handle completely new samples** (*e.g.*, COVID-19, brand-new products, or computer viruses) **without any prior information**. We have included this insightful discussion in Appendix F of the revised manuscript.
>
> > **Q3-1.** Given the high error bars in Tables 2 and 3, how conclusive are the results?
>
> **A6.** **We have conducted a statistical analysis to evaluate the statistical significance of our EDBD’s superior performance in Appendix C.6 of the manuscript**. Table 12 in the revised manuscript shows the $p$-values comparing EDBD to the runner-up in each setting for all the results in Tables 1, 2, and 3 of the manuscript. As shown in Table 12, the $p$-values for Table 2 and Table 3 are higher compared to those of Table 1. This is because spreading OOD detection tasks involve spreading simulations and feature sampling for each episode, leading to varying OOD spreading patterns and differing training and testing data. These variations significantly affect the difficulty level of each episode, resulting in substantial performance fluctuations and large standard deviations across methods. When the standard deviation is high, statistical tests are less likely to show significant differences between methods, yielding higher $p$-values, even when the mean performance differs between EDBD and the baselines.
>
> Nevertheless, **while the runner-up methods vary depending on the setting, EDBD consistently outperforms them across all settings and metrics, except for AUPR-T in spreading OOD detection on the LastFM Asia dataset using the SIS model**. Furthermore, as shown in Appendix C.9, EDBD consistently demonstrates its superiority when using SIS models with different combinations of parameters. **This consistency underscores EDBD’s robustness and generalizability** across various graph OOD-related tasks. If any additional statistical analyses are required, we are prepared to conduct them promptly.
>
> > **Q3-2.** Could the authors explain if the similar performance of EDBD and the Energy method suggests that data features, rather than graph structure, drive the performance?
>
> **A7.** In Table 2, EDBD and the Energy method shows similar performance; however, **in Table 3, EDBD demonstrates significant performance gains compared to the Energy method** (e.g., 7.80%p and 7.76% on Cora and LastFM Asia, respectively, in terms of FPR95-T when SI models are used). For the Spreading COVID-19 dataset corresponding to Table 2, as the reviewer pointed out, utilizing the graph structure for OOD detection (*i.e.,* EDBD) leads to slight performance gains compared to not using it (*i.e.,* Energy), suggesting that **leveraging features is more critical**. Nevertheless, **EDBD updating energies based on the graph structure still achieves consistent performance gains** across various spreading settings and all metrics in this dataset.

---

> ### Author Response · Authors · 2024-11-23
> **Response to Reviewer xogV (5/5)**
>
> >**Q4.** Since hyperparameter tuning can greatly impact performance, could the authors explain how it was done for both EDBD and baseline methods? Was the search for competitors restricted compared to EDBD? For GNNSAFE, was only the supervised loss used, or did the authors include GNNSAFE++ with energy regularization? If only the supervised loss was used, what motivated this choice?
>
> **A8.** **We have clarified hyperparameter tuning for both EDBD and baseline methods in Appendix B.2.3 and Appendix B.2.4**. **For EDBD, we employ a grid search for hyperparameter tuning $(\alpha, \beta, \epsilon, K)$ are selected from the following ranges**: { ($\alpha, \beta, \epsilon, K) | \alpha \in $ { 0.1, 0.2, 0.3, 0.5 }$, \beta \in$ { $1, \frac{1}{2}, \frac{1}{3}, \frac{1}{4}$ }$, \epsilon \in$ { 0.01, 0.05 , 0.1 , 0.5, 0.75} , $K \in$ {1, 2} } on validation sets. For baseline methods except for OODGAT, we utilize the implementations provided in the GitHub repository for GNNSAFE. For OODGAT, we utilize the official implementation provided by the authors. **Across all the baselines, we adhere to the hyperparameter tuning strategies and settings described in their respective papers**.
>
> **GNNSAFE++ incorporates energy regularization into training under a distinct setting where auxiliary OOD data**, in addition to testing OOD data, **are provied during training**. However, in real-world scenarios, such as infectious diseases, computer viruses, or brand-new products—which are typical targets of spreading OOD detection—the features of new OOD samples often differ entirely from existing ones. Therefore, **we consistently conduct experiments under the setting where no auxiliary data is available for training**.
>
> It is also noteworthy that the **energy regularization used in GNNSAFE++ was originally introduced by [6]**, which proposed energy-based OOD detection, rather than being specific to GNNSAFE. **This regularization** aims to obtain better energy scores and serves as a general method to **boost the performance of all energy-based OOD detection methods, including ENERGY, GNNSAFE, and EDBD**, rather than improving GNNSAFE alone.
>
> [5] Wu, Qitian, et al. "Energy-based Out-of-Distribution Detection for Graph Neural Networks." The Eleventh International Conference on Learning Representations.\
> [6] Liu, Weitang, et al. "Energy-based out-of-distribution detection." Advances in neural information processing systems 33 (2020): 21464-21475.

---

> > ### Comment · Reviewer_xogV · 2024-11-26
> >
> > Thank you to the authors for their detailed and thorough responses. I have updated my ratings to better reflect the contributions -even if limited- I recognize this paper provides.

---

> > > ### Author Response · Authors · 2024-11-27
> > >
> > > Thank you for recognizing the contributions of our work and for updating your rating toward acceptance. We sincerely appreciate your thoughtful review and the opportunity to address your concerns, which has significantly improved the quality of our paper.
> > >
> > > If you have any remaining concerns, please let us know; we are fully prepared to address them immediately.

---

### Official Review · Reviewer_cFt1 · 2024-11-04

**Soundness:** 4
**Presentation:** 3
**Contribution:** 3
**Rating:** 8
**Confidence:** 2

**Summary:**

This research addresses a significant gap in node-level out-of-distribution (OOD) detection by introducing a novel task that considers the interactions among nodes in a graph. Previous methods have relied on unrealistic benchmarks that randomly select OOD nodes, which do not reflect real-world scenarios where OOD properties can spread among neighboring nodes. The key contributions of this paper include:
1. Introduction of Spreading OOD Detection: This new task models how newly emerged OOD nodes can influence their neighbors, simulating more realistic scenarios of OOD detection.
2. Realistic Benchmark Creation: The authors develop benchmarks using epidemic spreading models to simulate the dynamics of OOD node interactions within graphs.
3. “Spreading COVID-19” Dataset: A specific dataset is introduced to illustrate the practical applications of spreading OOD detection, showcasing its relevance to real-world situations.
4. Energy Distribution-Based Detector (EDBD): The paper presents a new detection approach that employs an energy-aggregation scheme aimed at reducing the overlap in OOD scores between in-distribution (ID) and OOD nodes, enhancing detection accuracy.
5.Extensive Experimental Validation: Results from comprehensive experiments demonstrate that the proposed EDBD approach outperforms existing state-of-the-art methods in both spreading OOD detection and traditional node-level OOD detection tasks across seven benchmark datasets.
Overall, this work advances the field of OOD detection by incorporating the complexities of node interactions in graph structures, providing both theoretical and practical contributions.

**Strengths:**

This research addresses a significant gap in node-level out-of-distribution (OOD) detection by introducing a novel task that considers the interactions among nodes in a graph. Previous methods have relied on unrealistic benchmarks that randomly select OOD nodes, which do not reflect real-world scenarios where OOD properties can spread among neighboring nodes. The key contributions of this paper include:
1. Introduction of Spreading OOD Detection: This new task models how newly emerged OOD nodes can influence their neighbors, simulating more realistic scenarios of OOD detection.
2. Realistic Benchmark Creation: The authors develop benchmarks using epidemic spreading models to simulate the dynamics of OOD node interactions within graphs.
3. “Spreading COVID-19” Dataset: A specific dataset is introduced to illustrate the practical applications of spreading OOD detection, showcasing its relevance to real-world situations.
4. Energy Distribution-Based Detector (EDBD): The paper presents a new detection approach that employs an energy-aggregation scheme aimed at reducing the overlap in OOD scores between in-distribution (ID) and OOD nodes, enhancing detection accuracy.
5.Extensive Experimental Validation: Results from comprehensive experiments demonstrate that the proposed EDBD approach outperforms existing state-of-the-art methods in both spreading OOD detection and traditional node-level OOD detection tasks across seven benchmark datasets.
Overall, this work advances the field of OOD detection by incorporating the complexities of node interactions in graph structures, providing both theoretical and practical contributions.

**Weaknesses:**

Theoretical Justification for EDBD: While the energy-aggregation scheme is introduced as a key component of the EDBD, the paper would benefit from a more rigorous theoretical foundation explaining why this method effectively mitigates the mixing of OOD scores. Providing mathematical justifications or theoretical comparisons with other aggregation methods could strengthen the credibility of this contribution.

Sensitivity Analysis: Conducting a sensitivity analysis on the parameters of the EDBD model could provide valuable insights into how different configurations affect performance. Understanding how the model behaves under various conditions would help in fine-tuning the method for practical applications and could guide users in optimizing the approach for specific contexts.

**Questions:**

Theoretical Foundations of EDBD:
Question: What theoretical framework supports the effectiveness of the energy-aggregation scheme in EDBD?
Suggestion: A more detailed theoretical justification for why this approach mitigates the mixing of OOD scores would strengthen your argument and improve the method's credibility.

---

> ### Author Response · Authors · 2024-11-23
> **Response to Reviewer cFt1**
>
> We greatly appreciate your insightful feedback to improve our work. We provide our responses below.
>
> >**W1.** Theoretical Justification for EDBD: While the energy-aggregation scheme is introduced as a key component of the EDBD, the paper would benefit from a more rigorous theoretical foundation explaining why this method effectively mitigates the mixing of OOD scores. Providing mathematical justifications or theoretical comparisons with other aggregation methods could strengthen the credibility of this contribution.\
> **Q1.** Theoretical Foundations of EDBD: Question: What theoretical framework supports the effectiveness of the energy-aggregation scheme in EDBD? Suggestion: A more detailed theoretical justification for why this approach mitigates the mixing of OOD scores would strengthen your argument and improve the method's credibility.
>
> **A1.** Following the reviewer’s suggestion, **we have added to a theoretical analysis in Appendix E of the revised manuscript to explain why the existing energy aggregation method, GNNSAFE [1], leads to this undesired mixing of energies**.
> The key concept of EDBD is to enable the energies (OOD scores) to control their own update process through energy aggregation. In contrast, GNNSAFE, which performs energy aggregation based solely on the graph structure, cannot prevent the undesired mixing of energies between ID (in-distribution) and OOD (out-of-distribution) nodes. It is important to note that the goal of energy aggregation is to enhance the separation between the energies of ID nodes and those of OOD nodes.
> Since energies are positive values, we first assume that the energies of ID nodes and OOD nodes follow gamma distributions. Based on this assumption, **We theoretically demonstrate two points**: **(1)** an ID node aggregating energies from surrounding ID nodes and an OOD node aggregating energies from surrounding OOD nodes are beneficial for updating energies and **(2)** when both ID nodes and OOD nodes coexist in neighbors, simple energy aggregation that solely relies on the graph structure (*i.e.,* GNNSAFE) leads to undesired mixing of energies. **Combining (1) and (2), when both ID nodes and OOD nodes coexist as neighbors, enabling an ID node to focus on its neighboring ID nodes and an OOD node to focus on its neighboring OOD nodes allows for energy updates that improve OOD detection**. To this end, **EDBD utilizes the initial energies as a temporary OOD indicator, and regulates aggregation on each node to alleviate the mixing of energies between OOD nodes and ID nodes**.
>
> The reviewer's suggestion allows us to establish a theoretical basis for why GNNSAFE leads to the undesired mixing of energies, which has significantly enhanced our paper. We have incorporated this important theoretical justification into Appendix E of the revised manuscript.
>
> >**W2.** Sensitivity Analysis: Conducting a sensitivity analysis on the parameters of the EDBD model could provide valuable insights into how different configurations affect performance. Understanding how the model behaves under various conditions would help in fine-tuning the method for practical applications and could guide users in optimizing the approach for specific contexts.
>
> **A2.** We conduct additional experiments to address the reviewer’s concern and **provide a comprehensive analysis of the impact of different hyperparameters, including $\alpha$, $\beta$, $\epsilon$, and $K$**, on the performance of EDBD. Under two settings using SI and SIS models, respectively, we report FPR95-T in spreading OOD detection on the Cora dataset. $\alpha$, $\beta$, $\epsilon$, and $K$ are varied within the ranges of \{0.1,0.2,0.3,0.5\}, \{1/4,1/3,1/2,1\}, \{0.01, 0.05, 0.1, 0.5, 0.75\}, and \{1, 2\}, respectively, which are EDBD’s search ranges.
>
> Figure 7 in the revised manuscript shows the results. Compared to existing state-of-the-art performance, EDBD models consistently exceed it by a considerable margin regardless of the values of $\alpha$, $\beta$, and $K$. While EDBD with $\epsilon =0.75$ commonly shows worse performance compared to the existing state-of-the-art performance, the small value of $\epsilon$ indicates that energy aggregation focuses less on energy similarity and more on the graph structure. This result underscores the importance of considering energy distribution in the energy aggregation process. As stated in Appendix B.2.3 of the manuscript, we determine the hyperparameters of EDBD for each setting using grid search within fixed ranges, based on performance on the validation set. We have added this analysis of hyperparameter sensitivity to Appendix C.7 of the revised manuscript.
>
> **Figure 7:** https://anonymous.4open.science/r/ICLR_2025_6620-B3C2/Figure%207.png
>
> [1] Wu, Qitian, et al. "Energy-based Out-of-Distribution Detection for Graph Neural Networks." The Eleventh International Conference on Learning Representations.

---

> > ### Comment · Reviewer_cFt1 · 2024-11-26
> >
> > Thanks for your detailed response. I have no more issues.

---

> > > ### Author Response · Authors · 2024-11-26
> > >
> > > Thank you for evaluating our work as a good paper and for providing insightful feedback that further improved it. We greatly appreciate your support and are pleased that our responses addressed all your concerns!

---

### Author Response · Authors · 2024-11-23
**General Response**

In this paper, we introduce spreading OOD detection, a novel benchmark setup for node-level OOD detection that incorporates interactions among OOD samples. We also establish a new dataset, Spreading COVID-19, to demonstrate the applicability of spreading OOD detection in real-world scenarios. Additionally, we propose a novel scheme called the Energy Distribution-Based Detector (EDBD), which achieves outstanding performance in both spreading OOD detection and conventional node-level OOD detection tasks.

**We express our profound gratitude to all reviewers for dedicating their time and effort to thoroughly evaluate our manuscript. The constructive feedback provided by the reviewers has greatly enhanced the quality of our paper**. To address the reviewers’ concerns, **we provide point-by-point responses to all the questions and concerns raised by the reviewers**. Below is a summary of the key updates made to our paper:
* Theoretical Analysis (in Appendix. E)
* Hyperparameter Sensitivity (in Appendix. C.7)
* Comparison between LastFM Asia and a Human Contact Network (in Appendix. A)
* Spreading Covid on a Contact Network (in Appendix. C.8)
* SIS Models with Various Parameter Combinations (in Appendix. C.9)
* Spreading OOD Detection for an Urgent Contagion, RSV (in Appendix. C.10)
* Incomplete Graph Structures (Appendix. C.11)
* Spreading OOD Detection using SEIR models (Appendix. C.12)
* Discussion on "Why are human contact studies not considered?” (in Appendix. F)
* Discussion on "Why OOD detection is not approached as a binary classification?” (in Appendix. F)
* Discussion on "OOD Detection vs Outlier Detection” (in Appendix. F)

**For the revised and updated parts in the manuscript, we have marked the changes in blue**. We hope that the responses provided below address all the reviewers’ concerns, and **please let us know if the raised concerns are addressed**. We are happy to answer further questions and would be delighted to provide additional clarifications.

---

### Author Response · Authors · 2024-12-04
**Final Summary**

Dear Senior Area Chairs, Area Chairs, and Reviewers,

We sincerely appreciate the hard work and dedication of the chairs and the reviewers. During the discussion period, the reviewers provided thoughtful and professional feedback. **We have addressed all of the reviewer's concerns**, which allowed us to greatly improve the quality of the paper by incorporating their valuable suggestions. Specifically, in response to the reviewers' suggestion, we have included a theoretical analysis in Appendix E of the revised manuscript. **Except for Reviewer peqA, all reviewers have expressed supportive opinions for acceptance with high scores.**

**While we have thoroughly addressed Reviewer peqA's concerns, we have not received any response to our request for clarification on Reviewer peqA's decision over the past eight days, despite repeated attempts to seek a response from Reviewer peqA.** To facilitate the discussions among the chairs and reviewers, we would like to summarize (1) the contributions of our work and (2) our rebuttal addressing all of Reviewer peqA's concerns as a final note. We hope these points will be taken into consideration when the final decision is made.

***

## Contributions

1.  We introduce **spreading out-of-distribution (OOD) detection, a realistic benchmark for node-level OOD detection** that accounts for interactions among nodes.

2. We establish **a new dataset, Spreading COVID-19**, to demonstrate the applicability of spreading OOD detection in real-world problems.

3. We propose **a novel scheme**, Energy Distribution-Based Detector (EDBD), which **outperforms state-of-the-art methods in both spreading OOD detection and conventional node-level OOD detection tasks**.

***

## Reviewer peqA's Concerns

**More complex epidemic model**

* We conducted **additional experiments using the SEIR model suggested by Reviewer peqA** and verified EDBD's superiority in Appendix C.12 of the revised manuscript.

**More contemporary application than COVID-19**

* We conducted **additional experiments on an urgent contagion, Respiratory Syncytial Virus (RSV)**, and validated EDBD's effectiveness in Appendix C.10 of the revised manuscript.

**Validation of EDBD on graphs with incomplete structures**

* We conducted **additional experiments under edge-missing settings** and confirmed that EDBD maintains reasonable performance in Appendix C.11 of the revised manuscript.

**Lack of training on actual epidemic or contagion data in the context of the Spreading COVID-19 dataset**

* We conducted experiments on the **COVID-19 dataset using various epidemic models, which are widely used in epidemic spreading studies**, and confirmed the consistent superiority of EDBD, as demonstrated in Tables 2, 3, 14, 15, and 17 of the revised manuscript.

**Spreading OOD detection vs. outlier detection/node classification**

* We provided **a clear explanation of the fundamental differences** in Appendix F of the revised manuscript.

***

We strongly believe that the ICLR Senior Area Chairs, Area Chairs, and other reviewers will agree with our perspective upon reviewing our rebuttal to Reviewer peqA. **We hope that Reviewer peqA will take the time to review our clarifications and request even after the discussion period has concluded**. While we respect Reviewer peqA's evaluation, we trust that our rebuttal to Reviewer peqA and the lack of response from Reviewer peqA will be carefully considered in the final decision-making process.


Once again, we extend our sincerest thanks to all the chairs and reviewers for their efforts.

Sincerely,\
The Authors

---

### Meta-Review · Area_Chair_ohLg · 2024-12-17

**Metareview:**

This paper introduces "spreading OOD detection," a novel task in graph-based out-of-distribution (OOD) detection that accounts for the influence of OOD nodes on their neighbors. This addresses a significant limitation of existing methods which typically assume OOD nodes are isolated, an unrealistic assumption in many real-world scenarios where OOD properties can propagate through the graph structure. The authors propose a new benchmark generation scheme based on epidemic spreading models and introduce a "Spreading COVID-19" dataset to demonstrate its practical relevance. Furthermore, they propose the Energy Distribution-Based Detector (EDBD), a new method that outperforms existing approaches on both traditional and spreading OOD detection tasks.

Reviewers appreciate the introduction of the spreading OOD detection task and the development of more realistic benchmarks. They acknowledge the practical value of the "Spreading COVID-19" dataset and the potential of this work to inspire further research in this direction. The proposed EDBD method also demonstrates promising performance improvements in the conducted experiments.

However, reviewers also raise a concern:

- Benchmark Dependence: The EDBD method's strong performance appears to be closely tied to the specific characteristics of the proposed benchmark. It is unclear how well the method would generalize to other scenarios or if it offers significant advantages beyond being well-suited to the authors' benchmark.

Recommendation:

Despite this concern, the paper makes a valuable contribution by introducing a new and relevant task in OOD detection on graphs. The creation of realistic benchmarks and the "Spreading COVID-19" dataset are valuable resources for the community. I recommend accepting this paper as a poster.

**Additional Comments On Reviewer Discussion:**

The discussions during the rebuttal period were nice and interesting and helped in improving the submission

---

### Decision · Program_Chairs · 2025-01-22

Accept (Poster)